# The ubiquitin-like modifier FAT10 interferes with SUMO activation

Annette Aichem[1,2]*, Carolin Sailer[3], Stella Ryu[1,2], Nicola Catone[1], Nicolas Stankovic-Valentin[4], Gunter Schmidtke[2], Frauke Melchior[4], Florian Stengel [3] & Marcus Groettrup [1,2]

The covalent attachment of the cytokine-inducible ubiquitin-like modifier HLA-F adjacent transcript 10 (FAT10) to hundreds of substrate proteins leads to their rapid degradation by the 26 S proteasome independently of ubiquitylation. Here, we identify another function of FAT10, showing that it interferes with the activation of SUMO1/2/3 in vitro and down-regulates SUMO conjugation and the SUMO-dependent formation of promyelocytic leukemia protein (PML) bodies in cells. Mechanistically, we show that FAT10 directly binds to and impedes the activity of the heterodimeric SUMO E1 activating enzyme AOS1/UBA2 by competing very efficiently with SUMO for activation and thioester formation. Nevertheless, activation of FAT10 by AOS1/UBA2 does not lead to covalent conjugation of FAT10 with substrate proteins which relies on its cognate E1 enzyme UBA6. Hence, we report that one ubiquitin-like modifier (FAT10) inhibits the conjugation and function of another ubiquitin-like modifier (SUMO) by impairing its activation.

---

[1] Biotechnology Institute Thurgau at the University of Konstanz, CH-8280 Kreuzlingen, Switzerland. [2] Department of Biology, Division of Immunology, University of Konstanz, D-78457 Konstanz, Germany. [3] Department of Biology, University of Konstanz, D-78457 Konstanz, Germany. [4] Zentrum für Molekulare Biologie der Universität Heidelberg, DKFZ-ZMBH Alliance, D-69120 Heidelberg, Germany. *email: Annette.Aichem@bitg.ch

Modification of proteins with ubiquitin (Ub) or ubiquitin-like (UBL) modifiers regulates virtually all cell biological pathways in eukaryotic cells[1]. To exert their functions, UBL modifiers get covalently attached to their substrates by modifier-specific enzymatic cascades involving an E1 activating enzyme, an E2 conjugating enzyme, and an E3 ligase, which enables covalent attachment of the UBL modifier to its substrate. In most cases, this is a reversible process because the modifier can be removed by UBL-specific proteases[2]. Modification with Small-Ubiquitin-like Modifier (SUMO), also called SUMOylation, is performed with one of several known SUMO isoforms. In higher eukaryotes, there are at least three SUMO isoforms described (SUMO-1, -2, and -3). Reversible modification with SUMO is a highly dynamic process. Both, in the absence or presence of cellular stress, SUMO-1 is mostly found conjugated to substrates. The almost identical isoforms SUMO-2 and -3, named hereafter SUMO-2/3, are found mostly unconjugated in normal growth conditions and shift to predominantly conjugated form under a variety of cellular stresses[3]. Covalent modification with SUMO exerts many different functions as e.g. in signaling, regulation of transcription[4], in cellular stress responses e.g. upon heat shock[5] or under oxidative stress[6]. The SUMO E1 activating enzyme is a heterodimer comprised of SUMO-activating enzyme 1 (SAE1/AOS1) and 2 (SAE2/UBA2) with the active-site cysteine C173 being located on UBA2[7,8]. Similar to Ub, SUMO is activated at its C-terminal diglycine in an ATP/Mg dependent step at the AOS1/UBA2 adenylation domain and then transferred onto the active-site cysteine on UBA2. The formation of the SUMO adenylate thereby induces an active-site remodeling, which is necessary to bind SUMO via thioester linkage to the UBA2 catalytic cysteine[8,9]. The SUMOylation cascade further depends on a single E2 conjugating enzyme, called UBC9, and a few E3 ligases.

Many substrates of SUMO modification are transcription factors as e.g. members of the activator protein 1 (AP-1) family. One type of AP-1 complexes consist of Fos and Jun (c-Jun, JunB, JunD) heterodimers which bind to specific DNA regulatory elements. The regulation of AP-1 activity is complex and includes post-translational modifications such as phosphorylation or modifications with ubiquitin-like modifiers[10–12]. The transcriptional activity of JunB is regulated by several post-translational modifications such as c-Jun N-terminal kinase (JNK)-mediated phosphorylation that causes an enhanced transcriptional activation of the interleukin 4 gene[13]. Phosphorylation-dependent covalent modification of JunB with K48 Ub chains targets JunB for proteasomal degradation. In addition, JunB gets post-translationally modified with SUMO on three SUMOylation consensus sites with lysine 237 being the primary site of SUMO-1 or SUMO-2/3 conjugation. SUMOylation thereby stimulates the transcriptional activity of JunB[11].

Because of the described functions of JunB in the immune system we were wondering, if JunB might be a conjugation substrate of the ubiquitin-like modifier FAT10 (also designated ubiquitin D (UBD)). The fat10 gene was identified in 1996 by sequencing of the human MHC class I locus[14]. FAT10 is expressed in the immune system and its expression is synergistically upregulated by the pro-inflammatory cytokines interferon (IFN)-γ and tumor necrosis factor (TNF)-α[15,16]. Moreover, FAT10 expression is highly upregulated in many different cancer types such as hepatocellular carcinoma (HCC), colon or breast cancer where it enhances cell migration, invasion and metastasis formation[17–21]. FAT10 is also upregulated during the maturation of dendritic cells and epithelial cells in the medulla of the thymus where it affects T cell selection[22]. Moreover, FAT10 is conjugated to autophagy-targeted Salmonella typhimurium bacteria in mice and promotes resistance to Salmonella[23]. FAT10 consists of two UBL domains, which are connected by a flexible linker[14,24]. A

recent proteomic analysis has identified hundreds of FAT10 conjugation substrates and non-covalently interacting proteins[25]. Covalent conjugation of FAT10 to specific substrates is performed by an enzymatic cascade involving the Ub and FAT10-specific E1 enzyme UBA6[26–28], the likewise bi-specific E2 enzyme UBA6-specific enzyme 1 (USE1)[27,29] and putative E3 ligases, awaiting their identification. While covalent conjugation of FAT10 to its substrate proteins such as p62, USE1, UBE1, p53, or PCNA[25,29–32] leads to their direct, ubiquitin-independent degradation by the 26 S proteasome[33,34], the function of non-covalent interaction with FAT10 is still poorly investigated.

Here we identify the transcription factor JunB as a conjugation substrate of FAT10 and show that in presence of FAT10, SUMOylation of JunB is almost completely abolished. Moreover, we find that even bulk SUMO conjugation is largely down-regulated by FAT10, which is due to a direct inhibition of the SUMO E1 activating enzyme AOS1/UBA2 by FAT10.

## Results

**JunB becomes FAT10ylated at a SUMOylation consensus site.** Both, FAT10 and the transcription factor JunB have important roles in the regulation of the immune system. Therefore, we were wondering if JunB might be a conjugation substrate of FAT10. To test this notion we expressed JunB with a C-terminal FLAG-tag together with HA-tagged FAT10 in HEK293 cells and performed an anti-HA immunoprecipitation combined with western blot analysis. As shown in Fig. 1a, expression of both proteins resulted in the formation of a single JunB-FLAG-HA-FAT10 conjugate (Fig. 1a, lane 8), which accumulated in presence of the proteasome inhibitor MG132 (Fig. 1a, lane 9), pointing to a mono-FAT10ylation of JunB and a targeting of the conjugate to degradation by the 26 S proteasome. The conjugate was completely absent when instead of wild-type FAT10 the conjugation incompetent diglycine mutant HA-FAT10-AV was expressed (Fig. 1a, lanes 10 and 11). Next to this covalent modification, a non-covalent interaction of JunB-FLAG and HA-FAT10 or HA-FAT10-AV was observed, which was more pronounced under proteasome inhibition with MG132, since here, both, wild-type HA-FAT10 as well as HA-FAT10-AV accumulated. To exclude overexpression effects, we treated human peripheral blood lymphocytes (PBLs) for 48 h with IFN-γ and TNF-α to induce expression of endogenous FAT10 and performed an immunoprecipitation for FAT10 and western blot analysis for JunB (Fig. 1b). Also under endogenous conditions, a stable JunB-FAT10 conjugate was detectable upon immunoprecipitation using a FAT10-reactive antibody but not with an unspecific iso-type control antibody. The conjugate was even already visible without cytokine treatment since the cells expressed decent amounts of FAT10 mRNA as well as FAT10 protein already before cytokine treatment (Fig. 1b, lysate IB: FAT10 and real-time PCR data in the bar graph on the right). Conjugation of FAT10 to its substrates targets them for proteasomal degradation[25,29,30]. Cycloheximide (CHX) chase experiments using HEK293 cells, transiently transfected with expression plasmids for JunB-HA and 6His-3xFLAG-FAT10 (named hereafter as FLAG-FAT10) revealed that this is also true for the JunB-FAT10 conjugate (Fig. 1c). A densitometric analysis confirmed the degradation of the JunB-FAT10 conjugate (Fig. 1d). In a next step we aimed at identifying the lysine residues of JunB, which get FAT10ylated. FLAG-FAT10 was expressed in HEK293 cells together with HA-tagged JunB mutants, in which the lysines of the three SUMOylation consensus sites K237R, K267R, K301R were mutated singly or all together (K3R) to arginines[11]. Consecutive anti-HA immunoprecipitation and anti-FLAG western blotting revealed that the lysines of all three SUMOylation consensus sites

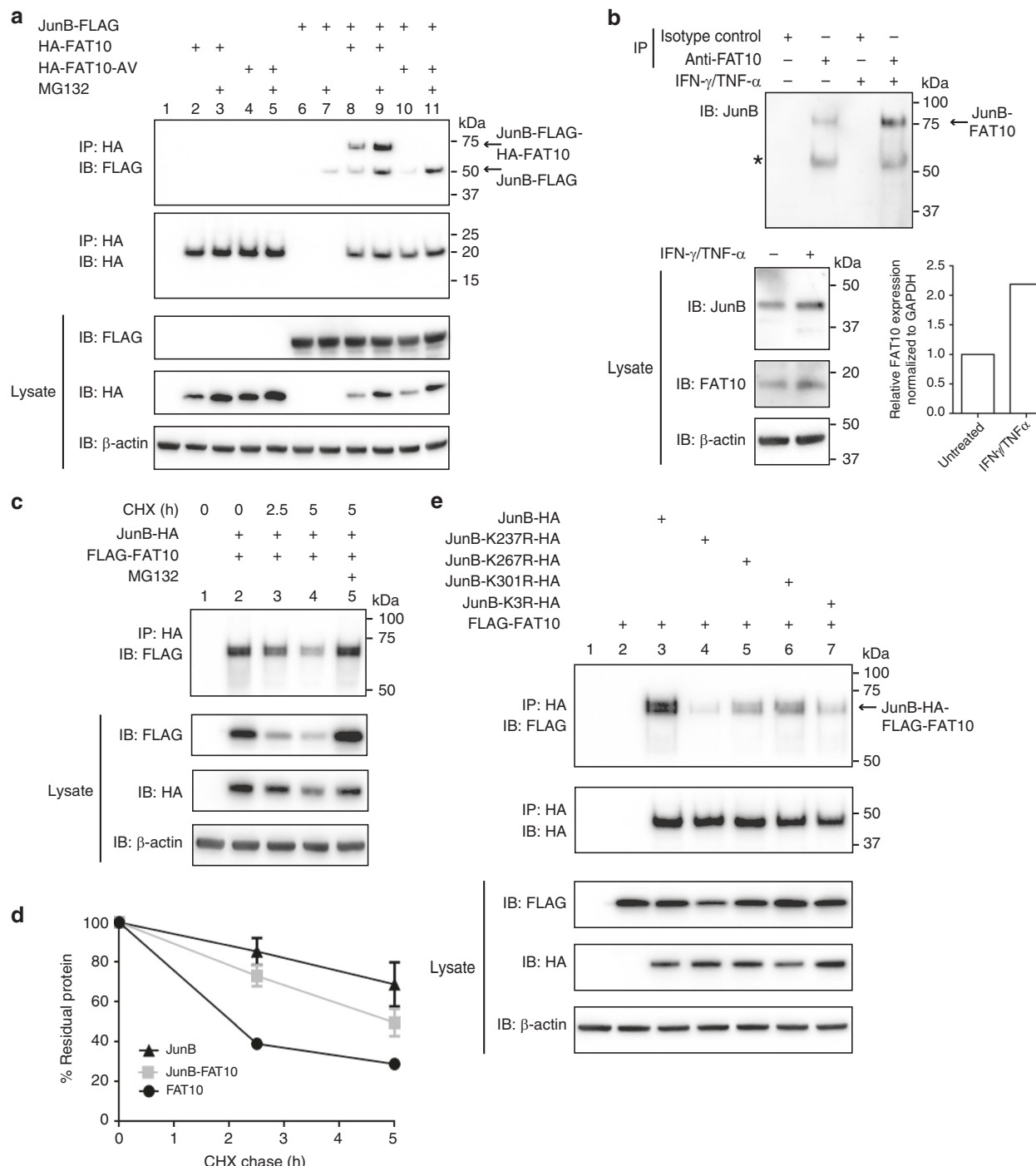

were involved in FAT10ylation since with all JunB mutants, the formation of the JunB-FAT10 conjugate was markedly diminished (Fig. 1e, lane 3 versus lanes 4–7).

**JunB SUMOylation is diminished in the presence of FAT10.** Since our data revealed that both modifiers were conjugated to the same lysines within JunB, we were wondering if FAT10ylation and SUMOylation of JunB was regulated by a putative competitive mechanism. In a first approach, FAT10ylation and SUMOylation of JunB was investigated under in vitro conditions with recombinant proteins. A clear JunB-FAT10 conjugate was formed under these conditions when recombinant JunB with a

C-terminal HA-6His tag (JunB-HA-His) was incubated with recombinant FLAG-tagged UBA6 and untagged FAT10 in in vitro buffer, supplemented with ATP (Fig. 2a, IB: HA, lane 3). In accordance with what we have published recently for two other FAT10 substrates, namely UBE1 and OTUB1[30,35], only the FAT10 E1 activating enzyme UBA6 was necessary to achieve in vitro FAT10ylation of JunB. Likewise, JunB was SUMOylated when it was incubated with AOS1/UBA2, UBC9, and either SUMO-1 (Fig. 2a, IB: HA, lane 5), or SUMO-2 (Fig. 2a, IB: HA, lane 9). In vitro SUMOylation with SUMO-1 or SUMO-2/3 was likewise clearly visible in a western blot using SUMO-reactive antibodies (Fig. 2a bottom, lane 5 and 9). In presence of both, the SUMOylation system (AOS1/UBA2, UBC9 and SUMO-1) and

**Fig. 1** JunB becomes FAT10ylated at a SUMOylation consensus site. **a** JunB-FAT10ylation under overexpressing conditions in HEK293 cells. Cells were transiently transfected with expression plasmids for the proteins indicated and an immunoprecipitation (IP) against HA-FAT10 combined with western blot (IB) analysis was performed. 10 μM  proteasome inhibitor MG132 was added for a total of 6 h before harvesting, where indicated. β-actin was used as loading control. Shown is one experiment out of three experiments with similar outcomes. **b** Conjugation of FAT10 to JunB under endogenous conditions in human PBLs. Endogenous FAT10 expression was induced by IFN-γ/TNF-α treatment for 48 h and JunB-FAT10 conjugate formation was analyzed in a combined immunoprecipitation/western blot analysis. Shown is one experiment out of three experiments with similar outcomes. The asterisk marks an unspecific background band. Mouse IgG1 was used as isotype control for the IP. Bar graph represents relative expression levels of FAT10 mRNA, normalized to the housekeeping gene GAPDH as measured by real-time PCR. Ct-levels of untreated cells were 22.4, and 21.7 for IFN-γ/TNF-α treated cells. **c** The degradation rate of the JunB-HA-FLAG-FAT10 conjugate in transiently transfected HEK293 cells was monitored by cycloheximide (CHX) chase experiments. Cells were treated for 2.5 or 5 h with 50 μg/ml CHX to inhibit de novo protein synthesis. Where indicated, cells were additionally treated with 10 μM proteasome inhibitor MG132 for a total of 6 h. Immunoprecipitation and western blot analysis was performed as described in **a**. **d** Densitometric analysis of ECL signals of CHX chase experiments as shown in **c**. ECL levels were normalized to the ECL signals of β-actin and the level of 0 h CHX was set to unity. Values are shown for five independent experiments with similar outcomes as means ± s.e.m. **e** FAT10 becomes conjugated to a lysine within one of the three reported SUMOylation consensus sites in JunB. Western blot analysis of JunB-HA-FLAG-FAT10 conjugates in transiently transfected HEK293 cells upon expression of wild-type JunB or of JunB mutants, in which single lysines within three different SUMOylation consensus sites or all three (JunB-K3R-HA) were mutated to arginines. Immunoprecipitation and western blot analysis was performed as described in **a**. Shown is one experiment out of three experiments with similar outcomes. Source data are provided as a Source Data file

the FAT10ylation system (FLAG-UBA6 and FAT10), again a JunB-FAT10 conjugate was detectable; however, SUMOylation of JunB was almost completely abolished (Fig. 2a, lane 6). However, a competition between SUMO and FAT10 for being conjugated to the same lysine could be excluded since JunB SUMOylation was also completely abolished in the presence of FAT10 but in the absence of UBA6 (Fig. 2a, lane 7). Moreover, even the non-conjugatable FAT10-AV mutant was as potent as wild-type FAT10 in inhibiting SUMO-1 or -2 conjugation to JunB (Fig. 2a, lane 8 and 10). This led to the conclusion that in presence of FAT10, JunB SUMOylation must be inhibited by a mechanism that is independent of the UBA6-dependent conjugation of FAT10.

To confirm these data *in cellulo*, a stably FLAG-FAT10 expressing HEK293T transfectant[23] was transiently transfected with an expression plasmid for JunB-HA or with the mutant JunB-K3R-HA, as negative control (Fig. 2b). Upon immunoprecipitation against the HA-tag of JunB and a western blot against endogenous SUMO-2/3 a SUMO conjugate pattern of JunB was observed, which was almost completely absent when instead of wild-type JunB the JunB-K3R-HA mutant was expressed as control (Fig. 2b, lane 2 and 3). When FLAG-FAT10 was co-expressed, SUMOylation of both JunB variants was strongly diminished (Fig. 2b, lane 5 and 6), showing FAT10´s inhibitory effect also under *in cellulo* conditions.

**FAT10 leads to a general downregulation of SUMO conjugation.** So far, we have shown that the SUMOylation of JunB was blunted in the presence of FAT10. To test this function of FAT10 also for additional substrates, SUMOylation of RanGAP (Ran-GAP1-tail) was investigated using an in vitro Fluorescence Resonance Energy Transfer (FRET)-based SUMOylation assay, as described earlier (ref. [36] and drawing in Fig. 3a). Recombinant SUMO-1 fused to yellow fluorescence protein (YFP) was incubated with recombinant RanGAP1-tail, fused to cyan fluorescence protein (CFP) in presence of ATP, AOS1/UBA2, and UBC9. The FRET signal, representing the increase in the amount of SUMOylated RanGAP1-tail, was measured over 1 hour (Fig. 3a, left panel, red line). The addition of increasing amounts of unlabeled SUMO-1 (0.35–5.4 μM SUMO-1) to the reaction led to loss of the FRET signal in a concentration-dependent manner. Addition of the same increasing concentrations of FAT10 to the reaction reduced the FRET signal in a similar way as addition of SUMO-1 did (Fig. 3a, middle panel), whereas the same increasing concentrations of ubiquitin had no effect on RanGAP

SUMOylation (Fig. 3a, right panel). As already observed for JunB SUMOylation in Fig. 2, the FAT10 diglycine mutant FAT10-AV also inhibited RanGAP SUMOylation, yet even more efficiently (Supplementary Fig. 1).

Assuming that a FAT10-mediated inhibition of SUMO conjugation must exert an impact on the in cellulo formation of SUMO conjugates we investigated if FAT10 might inhibit not only SUMOylation of single substrates but also of bulk SUMO conjugates. To this aim we used a stably FLAG-FAT10 expressing HEK293T transfectant, expressing moderate FLAG-FAT10 amounts similar to the level of endogenous FAT10 upon induction with cytokines. By western blotting, SUMO-1 or SUMO-2/3 conjugates, induced by heat shock, were monitored. Stable FLAG-FAT10 expression in HEK293T cells resulted in a marked decrease in overall SUMO-1 conjugates (Fig. 3b, lane 1, 2 versus 3, 4). Heat shock-induced SUMO-2 conjugation was likewise heavily decreased in FLAG-FAT10 expressing cells (Fig. 3c, lane 4 versus 2). Moreover, expression of the mutant FLAG-FAT10-AV in cells was just as efficient as its wild-type FLAG-FAT10 counterpart (Fig. 3d). To ensure that the down-regulation of SUMO-1 or -2/3 conjugation was not due to a negative influence of FAT10 onto *SUMO* mRNA expression, real-time PCR analysis was performed. In contrast to the western blot results, HEK293T-FLAG-FAT10 cells showed even a slight increase in *SUMO-1* and *-2/3* mRNA expression, as shown for one representative experiment in Supplementary Fig. 2a and b.

We next investigated the impact of FAT10 expression on the SUMOylation of an endogenous SUMO substrate, namely of promyelocytic leukemia protein (PML), in HEK293T wild-type or HEK293T-FLAG-FAT10 cells. PML bodies assemble through polymerization of several single PML proteins, forming a spherical structure. Polymerized PML undergoes a massive SUMOylation, which is a prerequisite for the formation of PML nuclear bodies (NBs)[37]. We therefore investigated by confocal microscopy, if the amount of SUMOylated PML bodies was reduced in presence of FAT10. While a clear colocalization of PML and SUMO-1 was detectable in HEK293T wild-type cells (Supplementary Fig. 3a, upper panels), this colocalization was reduced in cells, stably expressing FLAG-FAT10 (Supplementary Fig. 3a, lower panels). Nuclear localization of both, PML and SUMO-1, was verified by co-staining with the dye nuclear green DCS1 (Supplementary Fig. 3b, c). The significance of this downregulation was calculated by quantification of the Pearson`s coefficient, giving the probability of colocalization of PML and SUMO-1 in 105 wild-type or FLAG-FAT10 expressing cells (Supplementary Fig. 3a, right panel). We additionally calculated

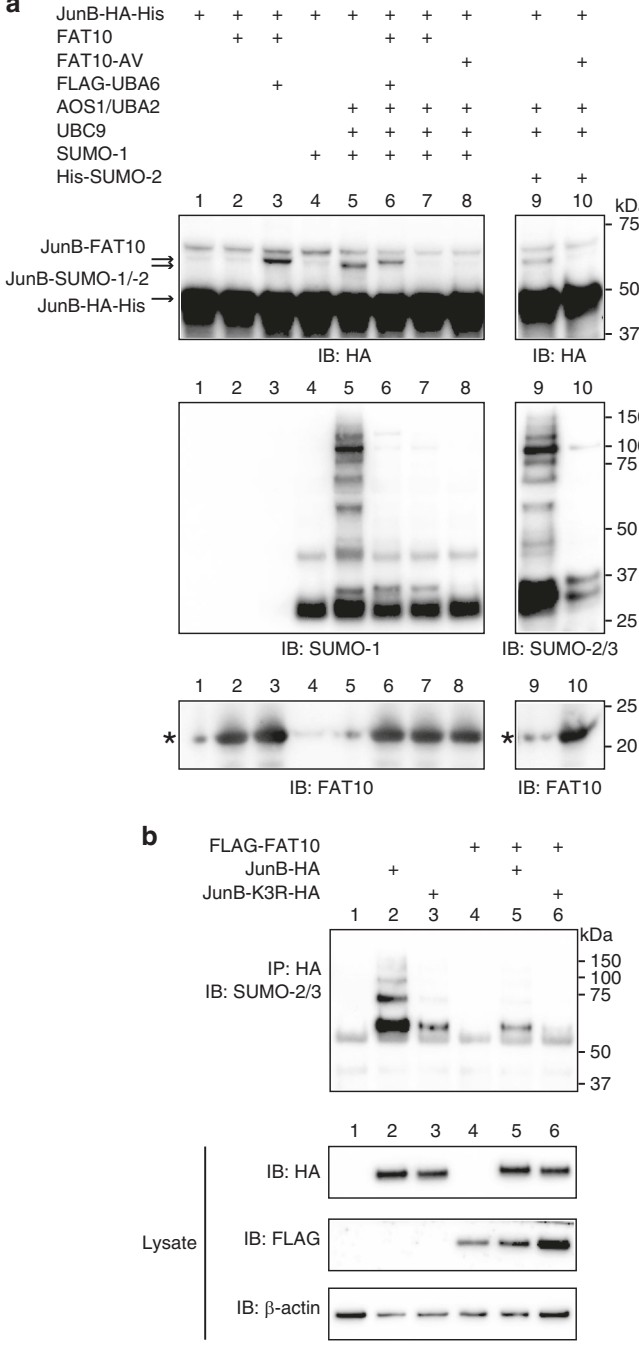

**Fig. 2** JunB SUMOylation is diminished in the presence of FAT10. **a** Western blot showing in vitro FAT10ylation and SUMOylation of JunB. SUMO-1 or -2 conjugation to JunB is completely inhibited in the presence of FAT10 or FAT10-AV. Recombinant proteins were incubated at 30 °C for 50 minutes in in vitro buffer, supplemented with ATP. Reactions were stopped by addition of gel sample buffer, supplemented with 10% 2-mercaptoethanol (2-ME) and boiled. Shown is one experiment out of three experiments with similar outcomes. The recombinant protein amounts used can be found in the methods section. Asterisks mark unspecific background signals in lanes 1, 4, 5 and 9. **b** In cellulo SUMOylation of JunB is diminished in the presence of FAT10. Stably FLAG-FAT10 expressing HEK293T cells were transiently transfected with HA-tagged JunB or JunB-K3R and an anti-HA immunoprecipitation combined with western blot analysis was performed. Shown is one experiment out of three experiments with similar outcomes. Source data are provided as a Source Data file

the amount of PML bodies per cell (Fig. 3e, f). Stable FLAG-FAT10 expressing cells hereby contained significantly less PML bodies per cell than their wild-type counterpart (Fig. 3e). When expression of endogenous FAT10 was induced with IFN-γ/TNF-α in HEK293 cells, an increase in the number of PML bodies was detected. However, HEK293 cells in which FAT10 had been knocked out by CRISPR/Cas9 (FAT10-ko)[38] contained significantly more PML bodies per cells upon IFN-γ/TNF-α treatment, most probably due to the missing endogenous FAT10-mediated inhibition of SUMO activation in these cells (Fig. 3f). Likewise, SUMOylation of PML was increased in HEK293 FAT10-knockout cells upon treatment with IFN-γ/TNF-α and As₂O₃ as compared to HEK293 wild-type (293) cells as monitored by western blot analysis of crude cell lysates (Fig. 3g, lanes 3 and 6). Taken together, the stable expression of FLAG-tagged FAT10 in HEK293T or HEK293 cells led to a massive downregulation of bulk SUMO conjugates but also of single endogenous substrates such as PML.

**Endogenous FAT10 inhibits SUMO conjugation**. Next we examined if the inhibition of SUMO activation could also be observed in presence of endogenous FAT10 in the HCC cell lines HepG2 and HepG3, treated or not with IFN-γ/TNF-α. Confirming our results obtained with FLAG-FAT10 expressing HEK293 and HEK293T cells, the presence of endogenous FAT10 also resulted in reduced amounts of SUMO-1 conjugates in both, HepG2 and HepG3 cells (Fig. 4a). Additionally, SUMO-2 conjugation, induced by treating the cells for 30 min with a 43 °C heat shock, was also clearly diminished in both cell lines in presence of endogenous FAT10 (Fig. 4b, lanes 3 and 4). A similar result was obtained when the same experiments were performed in additional cancer cells lines such as HCT116 or MCF-7 (Supplementary Fig. 4).

To exclude any impact of IFN-γ or TNF-α other than FAT10 induction on SUMO conjugation, we reconstituted HEK293 FAT10-knockout cells either with untagged FAT10 or treated the cells with IFN-γ/TNF-α to investigate a putative impact of the cytokines on SUMO conjugation. Upon expression of untagged FAT10, SUMO-1 conjugate formation was likewise reduced in these cells as seen before in FLAG-FAT10 expressing cells (Fig. 4c, lane 2 versus 1). As expected, treatment of FAT10-knockout cells with IFN-γ/TNF-α did not induce endogenous FAT10 expression and in addition, no impact on the SUMO-1 conjugate pattern was observed (Fig. 4c, lane 3). Treatment of the cells with IFN-γ/TNF-α combined with FAT10 reconstitution by transient transfection at the same time led again to a reduction in SUMO-1 conjugates (Fig. 4c, lane 4), confirming that the observed downregulation of SUMO-1 conjugation was due to the presence of FAT10 and not due to other IFN-γ/TNF-α-dependent effects. Regarding SUMO-2 conjugation upon heat shock, the obtained results could be confirmed also in the FAT10-knockout cells. While heat shock induced SUMO-2 conjugation (Fig. 4d, lane 5 versus 1), reconstitution with FAT10 led to a reduction of SUMO-2 conjugates (Fig. 4d, lanes 6, 8) and IFN-γ/TNF-α alone had again no inhibitory effect (Fig. 4d, lane 7 versus 5). Beyond that, mRNA expression levels of *SUMO-1*, *SUMO-2*, *UBA2*, *AOS1*, and *UBC9* in HepG2 and HepG3 cells did not significantly change upon IFN-γ/TNF-α treatment (Supplementary Fig. 2c–g), thus excluding a regulation of SUMO conjugation on the transcriptional level upon IFN-γ/TNF-α treatment. In summary, downregulation of SUMO conjugation was confirmed with endogenous FAT10 in HCC cell lines as well as in HEK293 cells.

**FAT10 inhibits SUMO activation by AOS1/UBA2**. While searching for the step of the SUMO conjugation pathway at which FAT10 interferes with SUMO conjugation we investigated

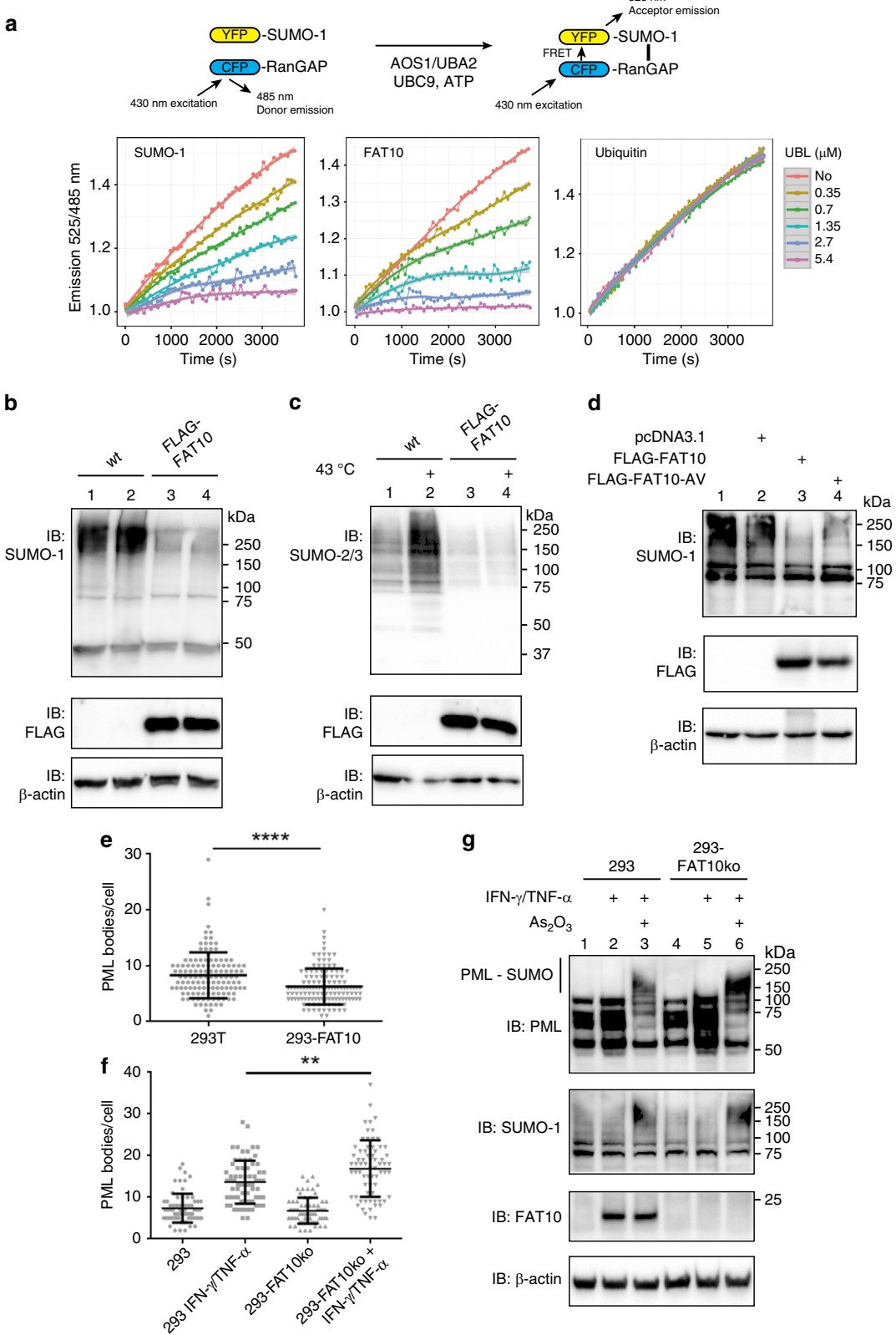

a putative impact of FAT10 on the activity of the SUMO E1 activating enzyme AOS1/UBA2. A co-immunoprecipitation of recombinant FAT10 and AOS1/UBA2 revealed that FAT10 directly interacted with AOS1/UBA2 in absence (Fig. 5a, lane 5) or presence of ATP (Supplementary Fig. 5). Likewise, a direct

interaction of the FAT10 diglycine mutants FAT10-AV or FAT10ΔGG with AOS1/UBA2 was detectable (Fig. 5a, lane 6 and 7). We then investigated the impact of FAT10 on SUMO-1 activation by AOS1/UBA2 under in vitro conditions with recombinant proteins. SUMO-1 activation was monitored by

**Fig. 3** FAT10 leads to a general downregulation of SUMO conjugation. **a** A FRET-based in vitro SUMOylation assay with the SUMO substrate RanGAP1-tail (RanGAP), fused to CFP, and SUMO-1 fused to YFP. All modifiers were added in the same increasing concentrations, as indicated, and the FRET signal was measured over a time course of 1 h. **b**, **c** Western blot analyses of SUMO-1 or -2/3 conjugates in crude cell lysates prepared from HEK293T wild-type cells (wt) or stably FLAG-FAT10 expressing HEK293T cells (FLAG-FAT10) in presence of 10 mM NEM using the indicated antibodies. **c** Before harvesting, cells were treated for 30 minutes with a 43 °C heat shock to induce SUMO-2 conjugation. β-actin was used as loading control. Shown is one experiment out of three experiments with similar outcomes. **d** Crude cell lysates of HEK293 cells, transiently transfected either with empty plasmid pcDNA3.1 or with expression plasmids for the FAT10 variants, as indicated. Preparation and analysis was performed as described in **b** and **c**. Shown is one experiment out of two experiments with similar outcomes. **e** Quantification of the number of PML bodies per cell in 293 T and 293-FAT10 cells ($n = 129$ cells each from three independent experiments). **f** Quantification of the number of PML bodies per cell in HEK293 (293) or HEK293-FAT10-knockout cells (293-FAT10ko), treated or not for 24 h with IFN-γ/TNF-α ($n = 69$ cells each from two independent experiments). **e**, **f** Significance was calculated using an unpaired, non-parametric Mann-Whitney test. A two-tailed P-value of <0.0001 was considered to be highly statistically significant. **e** $p < 0.0001$ (****); **f** $p < 0.025$ (**). **g** $As_2O_3$-induced SUMOylation of PML in HEK293 wild-type (293) and HEK293-FAT10-knockout (293-FAT10ko) cells, both stimulated or not for 24 h with IFN-γ/TNF-α. Proteins from crude lysates were separated on 4–12% Bis/Tris gradient gels (NuPage) and subjected to western blot analysis using PML, SUMO-1 or FAT10-reactive antibodies. β-actin was used as loading control. Shown is one representative experiment out of three experiments with similar outcomes. Source data are provided as a Source Data file

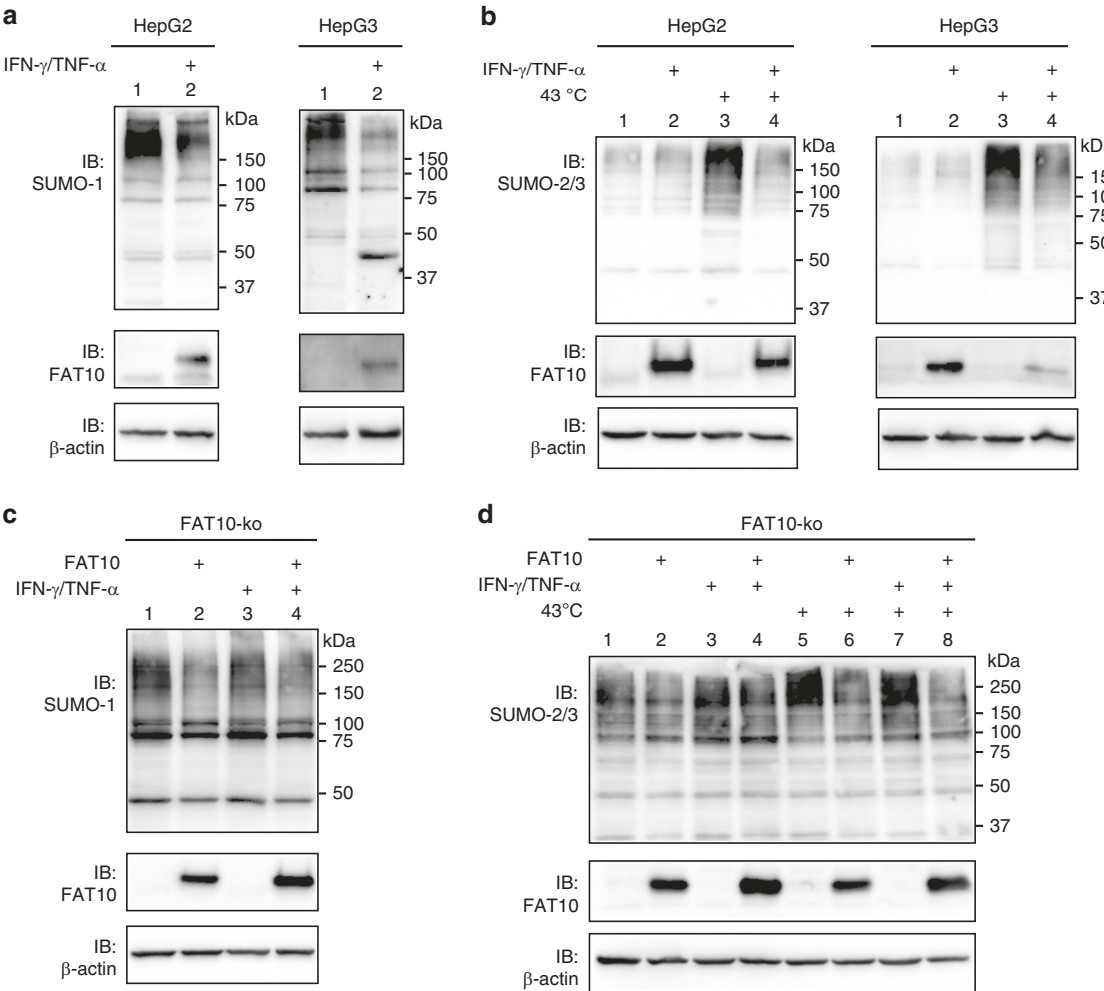

**Fig. 4** Endogenous FAT10 inhibits SUMO conjugation. **a**, **b** Western blot analyses of SUMO-1 or SUMO-2/3 conjugates in crude cells lysates of hepatocellular carcinoma cell lines HepG2 and HepG3, prepared in presence of 10 mM NEM. Endogenous FAT10 expression was induced with IFN-γ/TNF-α for 24 h, as indicated. **b** Cells were additionally treated with a 43 °C heat shock before harvesting, where indicated. **c**, **d** Endogenous FAT10 inhibits SUMO-1 conjugation. HEK293 FAT10-knockout cells (FAT10-ko) were treated and analyzed as described in **a** and **b**. Where indicated, cells were additionally transiently transfected with an expression construct for untagged FAT10. Each panel shows one experiment out of three experiments with similar outcomes. Source data are provided as a Source Data file

Western blotting under non-reducing and reducing (4% 2-ME) conditions with a UBA2-reactive antibody, since the active-site cysteine of AOS1/UBA2 (C173) had been allocated to UBA2. In the presence of SUMO-1 and ATP, a clear shift of the UBA2 signal was observed, which we attributed to a thioester-bound SUMO-1 at the active-site of UBA2 (Fig. 5b, IB: UBA2 (non-red.), lane 2). The described auto-SUMOylation of UBA2 upon SUMO-1 activation[39,40] was also clearly visible under

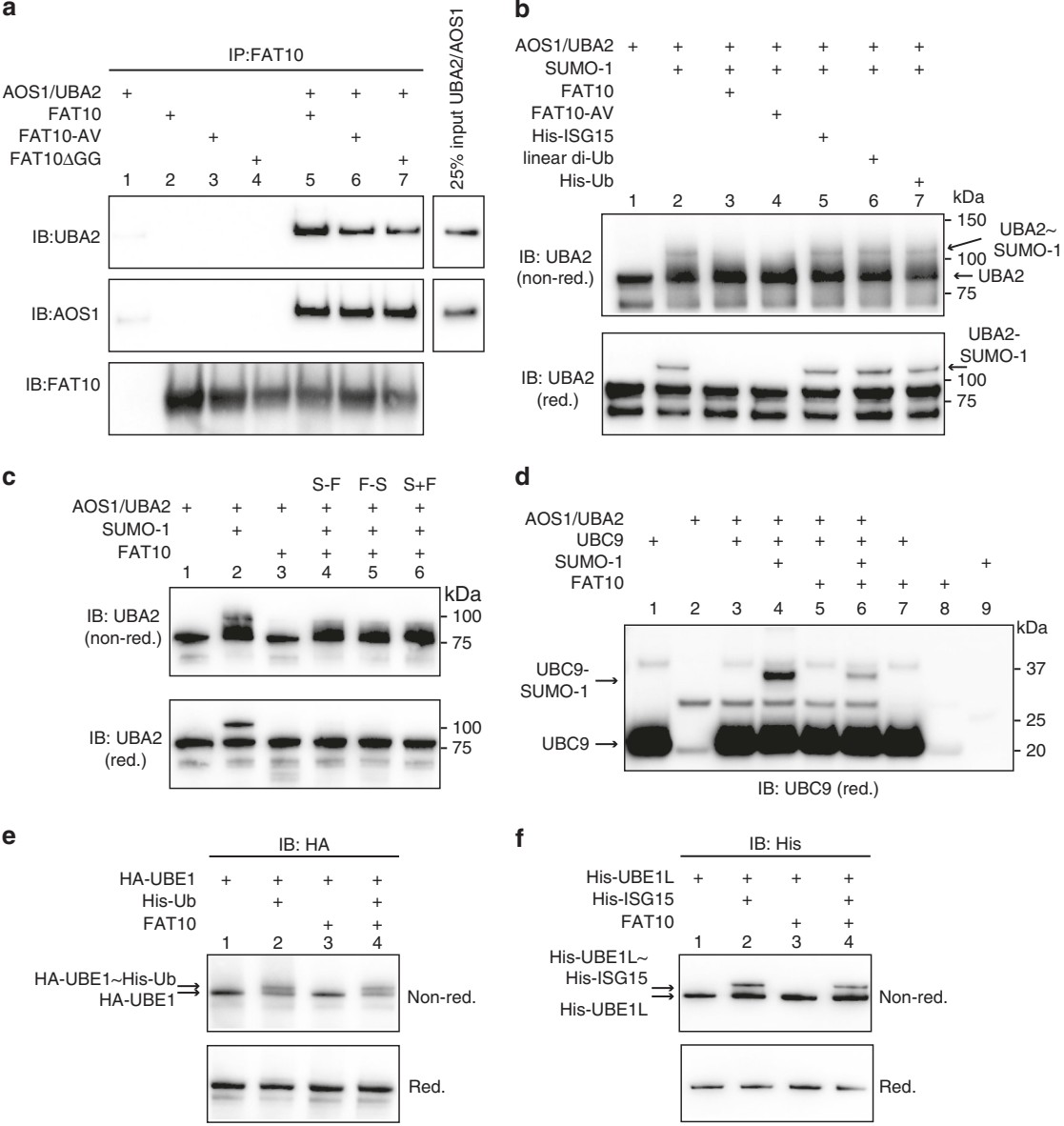

**Fig. 5** FAT10 inhibits SUMO activation by AOS1/UBA2. **a** In vitro co-immunoprecipitation experiment using recombinant AOS1/UBA2 and FAT10 variants. Immunoprecipitation was performed with a monoclonal FAT10-reactive antibody (clone 4F1 and Ref. [29]) and subsequent western blot analysis with the antibodies indicated. **b** Analysis of in vitro SUMO-1 activation by AOS1/UBA2 in presence or absence of FAT10 and other UBL modifiers, as indicated. Proteins were incubated in in vitro buffer, supplemented with ATP for 30 min at 30 °C. Samples were analyzed under non-reducing (non-red.) or reducing (red.) (4% 2-ME) conditions on a western blot using an UBA2-reactive antibody. Shown is one experiment out of three experiments with similar outcomes. **c** In vitro SUMO-1 activation in presence or absence of FAT10, as described in **b**. S-F: pre-incubation of SUMO-1 and AOS1/UBA2 for 30 min at 30 °C before addition of FAT10; F-S: pre-incubation of FAT10 and AOS1/UBA2 for 30 min at 30° before addition of SUMO-1; S + F: addition of both modifiers at the same time. Samples were analyzed under non-reducing (non-red.) or reducing (red.) (4% 2-ME) conditions on a western blot with an UBA2-reactive antibody. **d** Analysis of UBC9 auto-SUMOylation in presence or absence of FAT10 under in vitro conditions, as described in **b** and **c**. A western blot using an UBC9-reactive antibody was performed under reducing conditions (10% 2-ME) to visualize auto-SUMOylation of UBC9. **e** FAT10 does not interfere with Ub activation. In vitro activation of His-tagged Ub by its cognate E1 enzyme UBE1, in the presence or absence of FAT10. Proteins were incubated in in vitro buffer for 30 min at 37 °C and reactions were stopped by addition of gel sample buffer, supplemented (red.) or not (non-red.) with 4% 2-ME and boiled. Activation was analyzed on 4–12% gradient gels (NuPage). **f** FAT10 does not interfere with ISG15 activation. In vitro activation of His-tagged ISG15 (His-ISG15) by its cognate E1 enzyme UBE1L in the presence or absence of FAT10 as described in **e**. The recombinant protein amounts used in each experiment can be found in the methods section. Each panel shows one experiment out of three experiments with similar outcomes. Source data are provided as a Source Data file

reducing conditions (Fig. 5b, IB: UBA2 (red.), lane 2). However, in presence of equimolar concentrations of wild-type FAT10 or of FAT10-AV, SUMO-1 activation as well as the subsequent auto-SUMOylation was almost completely inhibited (Fig. 5b, lane 3 and 4), showing that FAT10 directly inhibited SUMO activation.

The FAT10ΔGG mutant was as potent as wild-type FAT10 or the FAT10-AV variant (Supplementary Fig. 6). Moreover, this effect was specific for FAT10 since in the presence of ISG15, linear di-Ub or Ub, SUMO-1 activation was not diminished (Fig. 5b, lanes 5–7).

Of note, the order of addition of FAT10 and SUMO-1 to the reaction mixture did not make any difference with respect to the inhibition of SUMO activation. Whether AOS1/UBA2 was pre-incubated for 30 min with SUMO-1 before adding FAT10 (S-F), or vice versa (F-S) or whether both modifiers were added at the same time (S + F), SUMO-1 activation was always completely abolished (Fig. 5c). We further observed that the auto-SUMOylation of the SUMO E2 enzyme UBC9 was likewise diminished in presence of FAT10 (Fig. 5d, lane 4 and 6). This might be explained by the reduced amount of activated SUMO in presence of FAT10 and consequently reduced transfer onto UBC9, but also by the inhibition of auto-SUMOylation of AOS1/UBA2, which was suggested to enhance the interaction with UBC9[39,40]. Moreover, the inhibitory action of FAT10 was specific for SUMO activation since the presence of FAT10 had no negative impact on the activation of Ub by its cognate E1 activating enzyme UBE1 (Fig. 5e, lane 2 and 4), or on the activation of ISG15 by its E1 activating enzyme UBE1L (Fig. 5f, lane 2 and 4).

**FAT10 gets unproductively activated by AOS1/UBA2.** Our data revealed that FAT10 directly interacts with and inhibits AOS1/UBA2. Therefore, we tested if FAT10 itself might be activated by AOS1/UBA2, thereby blocking access of SUMO to AOS1/UBA2. To test this idea, in vitro activation assays were performed showing a clear ATP-dependent activation of SUMO-1 under non-reducing and reducing conditions, as control (Fig. 6a, lane 2 and 3, and 6b, lane 2). Surprisingly, also FAT10 was activated in an ATP-dependent manner under these experimental conditions (Fig. 6a, lane 4 and 5, and 6b, lane 3), while the FAT10 diglycine mutants FAT10-AV and FAT10ΔGG were not activated (Fig. 6b, lane 4 and 5). The UBA2-FAT10 conjugate was completely absent under reducing conditions (Fig. 6a, IB: UBA2 (red.), 4% 2-ME), pointing to a thioester linkage. However, although FAT10 was obviously activated by AOS1/UBA2, it was not transferred onto its cognate E2 conjugating enzyme USE1 (Fig. 6a, IB: 6His, lane 7), whereas FAT10 activated by its cognate E1 activating enzyme UBA6 was handed over to USE1 under the same experimental conditions (Fig. 6a, IB: 6His, lane 9). Activation of FAT10 by AOS1/UBA2 was further confirmed by a luminescence-based ATP consuming assay, where the remaining ATP in the in vitro SUMO or FAT10 activation assay was measured. While a reduction in ATP was apparent in presence of SUMO-1, the reduction in ATP was even more pronounced when FAT10 instead of SUMO-1, or when both modifiers were added to the reaction at the same time (Fig. 6c).

As a further confirmation for thioester formation, the same assays were performed using the active-site cysteine mutant of AOS1/UBA2, AOS1/UBA2-C173A. As compared to the wild-type AOS1/UBA2 (Fig. 6d, lane 2) SUMO-1 was not activated by the AOS1/UBA2-C173A mutant, as expected (Fig. 6d, lane 4). The same was true for FAT10, which again was ATP-dependently activated by the wild-type but not by the mutant AOS1/UBA2 (Fig. 6d, lane 6 and 8) confirming that FAT10 indeed was activated and bound to the AOS1/UBA2 active-site cysteine. As we had shown above, FAT10 activated by AOS1/UBA2 could not be transferred onto the FAT10-specific E2 conjugating enzyme USE1; however, a partial transfer onto the SUMO E2 conjugating enzyme UBC9 was detectable (Fig. 6d, lane 10, IB: UBC9 non-red.) Different to SUMO-1, FAT10 was exclusively thioester-linked to UBC9 and did not resist reducing conditions, in contrast to the auto-SUMOylated form of UBC9 (Fig. 6d, IB: UBC9 red., lane 10 and 11).

In summary we could show, that the SUMO E1 AOS1/UBA2 can indeed activate FAT10. However, transfer of activated FAT10 onto an E2 conjugating enzyme was only detectable in presence of UBC9, but not in presence of the FAT10-specific E2 USE1. This posed the question whether AOS1/UBA2 might be able to replace the FAT10 E1 activating enzyme UBA6 and if FAT10 might be transferred onto substrates, using the SUMOylation pathway. We created a HEK293 UBA6 (UBA6-ko) knockout cell line with CRISPR-Cas9 technology and transfected these cells with a FLAG-FAT10 expression construct. FAT10 conjugates were clearly detectable in wild-type HEK293 cells (Fig. 7a, lane 2) but not in UBA6-ko cells (Fig. 7a, lane 3 and 4), thus excluding a redundancy of the two E1 enzymes for FAT10 conjugation. Even overexpression of AOS1/UBA2 in UBA6-ko cells didn´t rescue endogenous FAT10 conjugate formation (Fig. 7b, lane 4). This further confirmed that AOS1/UBA2 does not serve as a second E1 activating enzyme for FAT10. The same was true in case of in vitro FAT10ylation of JunB. While FAT10 was conjugated to JunB in presence of UBA6 and ATP (Fig. 7c, lane 7), no conjugation of FAT10 to JunB was visible in presence of AOS1/UBA2, UBC9 and ATP (Fig. 7c, lane 8), although JunB SUMOylation was detectable under same experimental setup (Fig. 7c, lane 10).

**FAT10 non-covalently interacts mainly with AOS1.** Next, we investigated the mechanism how FAT10 was able to inhibit SUMO activation. The results shown above suggest a blockage of the SUMO conjugation pathway through an unproductive FAT10 activation by AOS1/UBA2. However, this does not explain the results obtained with the FAT10 diglycine mutants FAT10-AV and FAT10ΔGG. Since both mutants were at least as potent as wild-type FAT10 in inhibiting SUMO activation (Fig. 5b and Supplementary Fig. 6), we hypothesized that the main pathway of inhibition must be mediated by a non-covalent interaction of FAT10 with the adenylation domain of AOS1/UBA2. To test this idea, we performed chemical protein crosslinking coupled to mass spectrometry (XL-MS) experiments. The analysis revealed a large number of high-confidence crosslinks both within the different subunits of AOS1/UBA2 but also between FAT10 and AOS1 and, less pronounced, FAT10 and UBA2, suggesting that FAT10 does indeed likely mainly interact with AOS1. This corroborated our in vitro co-immunoprecipitation data in Fig. 5a, showing that FAT10 directly interacts with both SUMO E1 subunits, however, as shown here, apparently to a higher extent with AOS1 (Fig. 7d and Supplementary Data 1). This finding was further confirmed by determination of the affinity constant of the FAT10-AOS1/UBA2 interaction using the Octet system (Supplementary Fig. 7). The measured $K_d$ value of 1.8 μM for the interaction of FAT10 and AOS1/UBA2 is similar to the previously published $K_m$ value for SUMO-1 and AOS1/UBA2 of 0.74508 ± 0.1105 μM[41], which is in good agreement with the FRET-based inhibition assay shown in Fig. 3a.

The same experiment performed with the FAT10 diglycine mutant FAT10-AV resulted in the identification of approximately the same crosslinks as seen for wild-type FAT10, suggesting that the covalent interaction of FAT10 with AOS1/UBA2 was not mandatory to exert the inhibitory function (Fig. 7e and Supplementary Data 2). Moreover, a similar ATP consumption was measured in in vitro activation assays with AOS1/UBA2 when instead of wild-type FAT10, FAT10-AV was added to the reaction, consistent with an adenylation of the mutant, albeit it can`t be thioester linked to the catalytic cysteine of UBA2 (Fig. 6c).

Upon adenylation of SUMO, the UBA2 Cys domain undergoes a conformational change and rotates the active-site cysteine by 130° towards the adenylated SUMO. This step is a prerequisite for the formation of a thioester bond between the SUMO C-terminal

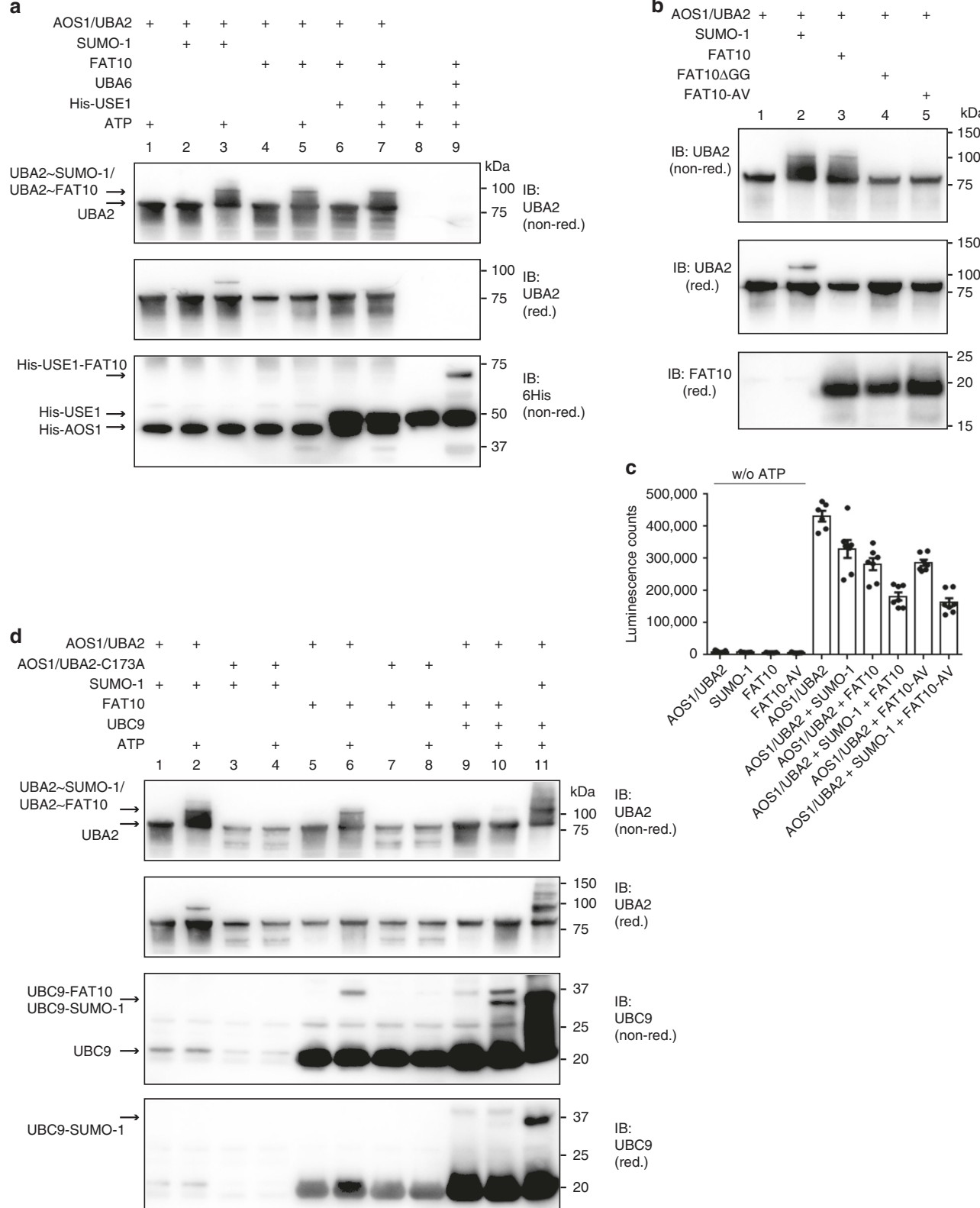

glycine and the active-site cysteine and thus a prerequisite for SUMO conjugation[8]. We hypothesized that a non-covalent interaction of FAT10 with AOS1/UBA2 might sterically block this rotation and thus inhibit SUMO activation. To test this hypothesis, quantitative crosslinking experiments were performed and the intramolecular crosslinks between UBA2 and AOS1 in presence of SUMO-1 alone, or of SUMO-1 and FAT10 together

were investigated (Supplementary Fig. 8b–d and Supplementary Data 3 and 4). For this purpose, the minimal molar excess of FAT10 to AOS1/UBA2 was determined, which was necessary to lead to a complete inhibition of SUMO-1 activation. The in vitro experiments revealed that a molar excess of 4:1 (FAT10 : AOS1/UBA2) in presence of a molar excess of 4:1 (SUMO-1 : AOS1/UBA2) was sufficient to almost completely inhibit SUMO-1

**Fig. 6** FAT10 can be activated by AOS1/UBA2. **a** In vitro experiment showing SUMO-1 or FAT10 activation by AOS1/UBA2. Activation was performed in presence or absence of ATP, as indicated, under in vitro conditions with recombinant proteins for 30 min at 37 °C. Proteins were separated on 4–12% gradient gels (NuPage) and activation was visualized by western blot analysis under non-reducing (non-red.) or reducing (red.) (4% 2-ME) conditions using the antibodies indicated. **b** Same as in **a** but additionally with the FAT10 diglycine mutants FAT10ΔGG and FAT10-AV. **c** Luminescence-based ATP consumption assay. Shown is the mean of five independent in vitro SUMO or FAT10 activation assays performed as shown in **a**. The residual ATP was measured by an ATP consuming luciferase reaction. The single recombinant proteins in the absence of ATP (w/o ATP) were used as control. **d** Same as in **a** but additionally with the AOS1/UBA2 active-site mutant AOS1/UBA2-C173A and UBC9. The exact recombinant protein amounts used in each experiment can be found in the methods section. Each western blot panel shows one experiment out of three experiments with similar outcomes. Source data are provided as a Source Data file

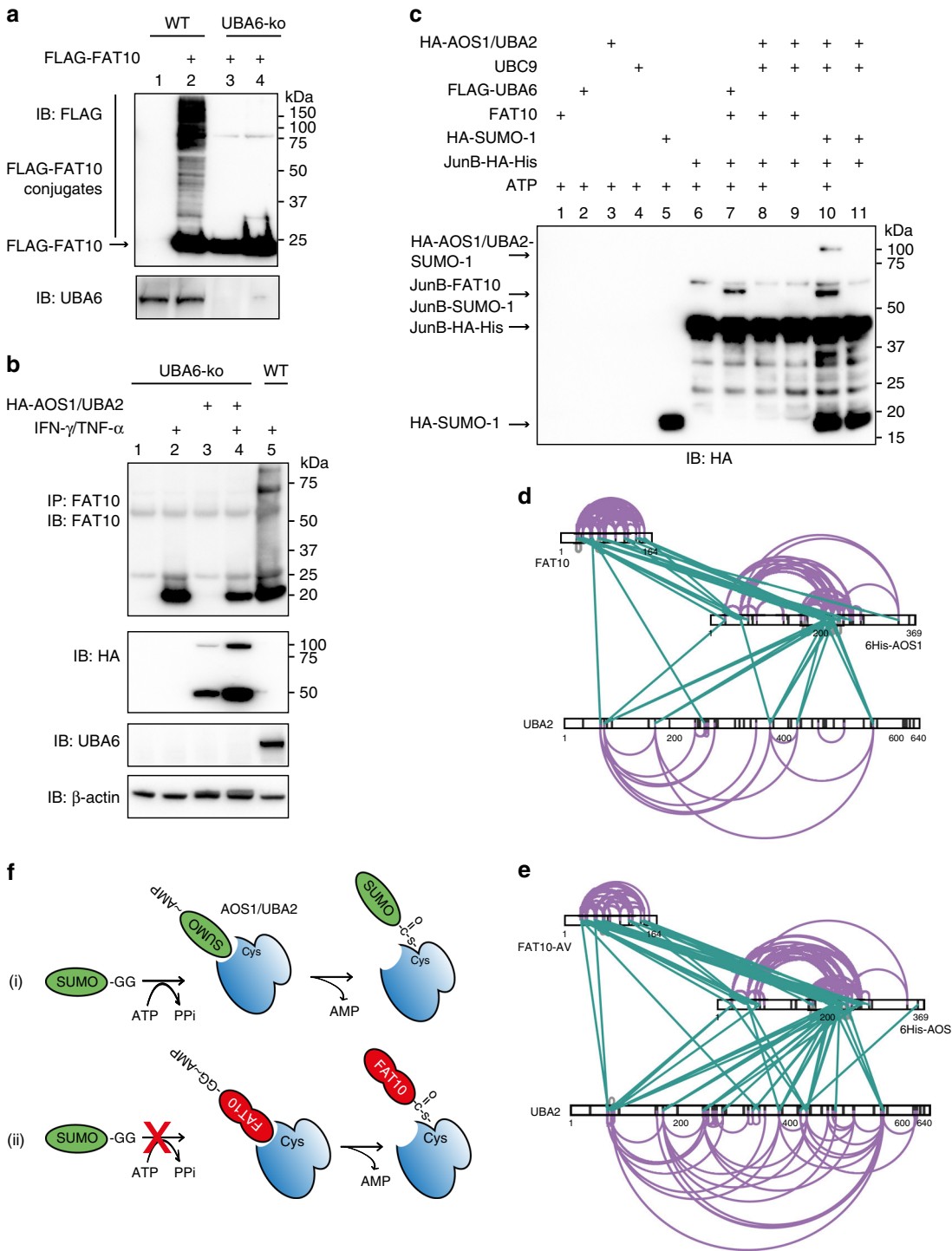

**Fig. 7** FAT10 is unproductively activated by AOS1/UBA2. **a** No FAT10 conjugates are formed in HEK293-UBA6 knockout cells. FAT10 conjugate formation was investigated using crude cell lysates, prepared under denaturing conditions of FLAG-FAT10 expressing HEK293 wild-type (WT), or HEK293-UBA6 knockout (UBA6-ko) cells. FAT10 conjugates were visualized by western blotting using the antibodies indicated. Shown is one experiment out of three experiments with similar outcomes. **b** Reconstitution of UBA6-ko cells with overexpressed AOS1/UBA2 does not rescue FAT10 conjugate formation in these cells. HEK293-UBA6-ko cells were transiently transfected with expression plasmids for SUMO E1 subunits UBA2 and AOS1. Cells were treated with IFN-γ/TNF-α to induce endogenous FAT10 expression. HEK293 wild-type (WT) cells were used as a control. Proteins were separated on 4–12% gradient gels (NuPage) and visualized by western blotting using the antibodies indicated. Shown is one experiment out of two experiments with similar outcomes. **c** No conjugation of FAT10 onto JunB under in vitro conditions in presence of the SUMOylation system instead of the FAT10ylation system after 60 min at 37 °C. In vitro JunB FAT10ylation assay was performed as shown in Fig. 2a. The exact recombinant protein amounts used are listed in the methods section. Shown is one experiment out of three experiments with similar outcomes. **d** Intra- and inter-protein crosslinks of AOS1/UBA2 and FAT10 in the absence of ATP. FAT10 (Sequence offset to UniProt sequence -1 aa due to tag) and the SUMO E1 subunits 6-His-AOS1 (Sequence offset to UniProt sequence: +23 aa) and UBA2 are depicted on the residue level. Lysine residues, as potential targets of the crosslinking agent used in this study, are indicated as black lines. Inter-protein crosslinks are shown in green and intra-protein crosslinks in purple. **e** Intra- and inter-protein crosslinks of AOS1/UBA2 and FAT10-AV in the absence of ATP, as described in **d**. **f** Schematic depiction of the mechanisms how FAT10 inhibits SUMO activation. (i) SUMO activation by AOS1/UBA2 in absence of FAT10. (ii) Inhibition of SUMO activation when FAT10 is non-covalently bound to the adenylation site of AOS1/UBA2, or when FAT10 is thioester bound to the active-site cysteine of the AOS1/UBA2. Source data are provided as a Source Data file

activation after 10 min at 30 °C (Supplementary Fig. 8a). Here, the addition of FAT10 to AOS1/UBA2 and SUMO-1 resulted in no apparent differences in the crosslinking pattern within and between AOS1/UBA2 and SUMO-1 (Supplementary Fig. 8c and d), indicating that FAT10 has no discernible impact on the conformational state of the activated SUMO E1. However, the experiments did confirm that SUMO-1 and FAT10 share at least partially the same binding sites on AOS1, pointing again to a competition of FAT10 and SUMO for binding to the adenylation site (Supplementary Fig. 8c, green lines).

Taken together, we show that FAT10 inhibits SUMOylation at the SUMO E1 activating enzyme AOS1/UBA2. Our results suggest a competitive mechanism in which FAT10 itself is activated by AOS1/UBA2 but not transferred onto substrates leading to a blockage of the SUMO conjugation pathway. We propose a model (Fig. 7f) according to which FAT10, either non-covalently bound to the adenylation site of AOS1/UBA2 or thioester-linked to the catalytic cysteine of AOS1/UBA2, blocks SUMO activation.

## Discussion

While investigating JunB as a substrate of FAT10 we have uncovered a so far unknown mechanism how SUMO conjugation is regulated by FAT10. Our peptide mapping and immunoprecipitation experiments showed that FAT10 directly interacted with both SUMO E1 subunits, UBA2 and AOS1. This direct interaction led to a complete inhibition of SUMO activation under in vitro conditions (Fig. 5), and to a considerable reduction of bulk SUMO-1 or -2/3 conjugates in cells (Figs. 3 and 4). This mechanism is very specific since the other tested UBL modifiers ISG15 and Ub did not interfere with SUMO activation. Moreover, the presence of FAT10 had no influence on Ub or ISG15 activation by their cognate E1 activating enzymes (Fig. 5e, f). These data pose the question how FAT10 is able to perform this inhibition. One possible explanation comes from our in vitro experiments, showing that FAT10 itself can be activated in an ATP-dependent manner by AOS1/UBA2. However, this activation is unproductive, since FAT10 is not transferred onto substrates such as JunB, leaving FAT10 bound to AOS1/UBA2, thus blocking the access of SUMO to its E1 enzyme (Figs. 6 and 7a–c). Nevertheless, thioester formation with AOS1/UBA2 was not essential for inhibition as the FAT10 diglycine mutants FAT10-AV and FAT10ΔGG exerted the same inhibitory effect as wild-type FAT10 (Figs. 3d, 5b and Supplementary Fig. 6). Based on our data, we propose that FAT10´s inhibitory action is mainly due to a non-covalent binding to the adenylation domain of AOS1/

UBA2, which normally SUMO occupies when being activated, and to a minor extent due to binding to the active-site cysteine by forming a thioester linkage. This hypothesis is based on the facts that all FAT10 variants could be adenylated at AOS1/UBA2; however, only the wild-type FAT10 could be thioester linked to the catalytic cysteine of UBA2. The latter was further confirmed by our quantitative crosslinking data showing that the presence of FAT10 had no impact on the conformational state of activated AOS1/UBA2, thus a transfer of activated FAT10 onto the catalytic cysteine is not hindered. Thus, both, covalent and non-covalent interaction of FAT10 with AOS1/UBA2, result in a blockage of SUMO activation and conjugation.

What then are the functional consequences of inhibiting SUMOylation by FAT10? We investigated the impact of FAT10 on single SUMO substrates, namely JunB and PML protein. In case of JunB we could show, that in presence of FAT10, SUMOylation of JunB was completely inhibited under in vitro conditions and strongly downregulated in cells. Next to the identified blockage of AOS1/UBA2, JunB SUMOylation may additionally be inhibited by the conjugation of FAT10 directly to lysines within the SUMOylation consensus sites of JunB, thus blocking SUMOylation and leading to the degradation of JunB by the 26 S proteasome (Fig. 1). Interestingly, a competition between SUMO and FAT10 for being conjugated to the same lysine could be excluded since JunB SUMOylation was completely abolished in the presence of FAT10 but in the absence of UBA6 (Fig. 2a, lane 7). SUMOylation of JunB promotes interleukin 2 and 4 (IL-2 and IL-4) transcription and differentiation of Th2 helper cells[11]. FAT10-mediated degradation and reduced SUMOylation of JunB may hence contribute to the observed elevation of the Th2-derived cytokine IL-10 in the muscle and plasma of FAT10[−/−] mice[42].

FAT10 is highly upregulated under inflammatory conditions in presence of IFN-γ and TNF-α[15,16]. Pro-inflammatory cytokines which cause expression of FAT10 are also expressed in tumor microenvironments and contribute most probably to the high expression of FAT10 in different cancer types[19,20,43,44]. By confocal microscopy we detected significantly reduced numbers of promyelocytic leukemia (PML) bodies in HEK293 cells, stably expressing moderate amounts of FLAG-FAT10 (Fig. 3e). Likewise, more PML bodies and an increase in PML SUMOylation was measured in HEK293 FAT10-knockout cells, treated with IFN-γ /TNF-α, as compared to untreated cells (Fig. 3f, g). PML bodies are important regulators of several cellular processes such as genome maintenance and DNA repair. A FAT10-mediated downregulation of PML SUMOylation might therefore favor the transforming capacities of cells expressing FAT10 and promote tumor formation, which would be in line with the above

mentioned roles of FAT10 in cancer development. In fact, SUMO was described to be indispensable for cancer cell proliferation[45,46] and an siRNA-mediated knockdown of UBA2 was shown to inhibit colorectal cancer cell invasion and migration by down-regulation of the Wnt/β-catenin signaling pathway[47]. Further elucidating the mutual influence of SUMO and FAT10 conjugation systems during cancer development might therefore open new avenues for the development of anti-cancer therapies.

## Methods

**Plasmids.** Plasmids used for transient expression of JunB variants in HEK293 cells were pCMV-myc-DDK-JunB (JunB-FLAG) (Origene), pcDNA3-JunB-HA, pcDNA3-JunB-HA-K237R, pcDNA3-JunB-HA-K267R, pcDNA3-JunB-HA-K301R, and pcDNA3-JunB-HA-K3R [11], kindly provided by Dr. Marc Piechaczyk, Montpellier). For expression of FAT10 variants, the following plasmids were used: pcDNA3.1-HA-FAT10[48], pcDNA3.1-HA-FAT10-AV[29], and pcDNA-His-3xFLAG-FAT10 (FLAG-FAT10)[26]. pcDNA3.1-His/-A (Invitrogen) was used to balance plasmid amounts. For the expression of AOS1/UBA2 in HEK293 cells, pCRUZ-HA-AOS1 and pcDNA3-HA-UBA2 were used. For generation of pcDNA3-HA-UBA2, UBA2 was amplified by PCR from HEK293T cDNA and cloned with Asp718 and XhoI into pcDNA3.1-HA-UBE1L2[28]. Plasmids for recombinant protein expression in E. coli were pET11d-UBA2 and pET28α-His-AOS1 (both previously described[49,50]). pET11a-UBA2-C173A was generated by site-directed mutagenesis of pET11d-UBA2 with primers UBA2 SDM C173A fwd 5′-GAAGGTGTGTTACGAATTGTGGCGCCAGGAAAGGTTCTCTGGG-3′ and UBA2 SDM C173A rev 5′-CCCAGAGAACCTTTCCTGGCGCCACAAT TCGTAACACACCTTC-3′. Recombinant FAT10 or the cysteine-free FAT10 variant FAT10(C0) were expressed from pSUMO-FAT10[51] or pSUMO-FAT10 (C0)[24], respectively. A FAT10 variant with the C-terminal diglycine motif changed from GG to AV was expressed from pSUMO FAT10-AV[38]. For recombinant expression of a FAT10 variant missing the C-terminal diglycine motif, pSUMO-FAT10ΔGG was generated with primers 5′-BsaI-FAT10 5′-CCAGTGGGTCT CAGGTGGTGCTCCCAATGCTTCCTGCCTCTGTGTGC-3′ and 3′-XhoI-dGG-STOP-FAT10-SYC 5′-CCGCTCGAGTTAAATACAATAAGATGCCAGGAAG AGTAAG-3′. To generate the bacterial expression construct pET24α-JunB-HA-6His, JunB was amplified from pCMV6-JunB-myc-DDK (Origene) with primers 5′-BamHI-hJunB 5′-CGCGGATCCATGTGCACTAAAATGGAACAGCCC-3′ and 3′-XhoI-hJunB 5′-CCGCTCGAGGAAGGCGTGTCCCTTGACC-3′ and inserted via restrictions sites BamHI and XhoI into pET24α. The HA-Tag was synthesized with XhoI restriction sites 5′-TCGAGTACCCCTACGACGTGCCC GACTACGCCC-3′ (Microsynth, Balgach, Switzerland) and inserted 3`between JunB and the 6His-tag to create pET24α-JunB-HA-6His.

**Recombinant proteins and in vitro assays.** GST and GST-FAT10 were purified as follows: plasmids pGEX4T-3 or pGEX4T-3-FAT10[28] were transformed into E. coli BL21(DE3) and expression of GST or GST-FAT10 was induced with 0.1 mM Isopropyl-β-D-thiogalactoside (IPTG) for 5 h at 20 °C. Cells were harvested, dissolved in 5 ml PBS/g bacterial pellet, lysed (Cell Disruptor, Constant Systems) and PMSF was added to a final concentration of 1 mM. The cell suspension was cleared by centrifugation at 4 °C for 30 min at 32,000 × g, filtered through a 0.45 µm filter and incubated for 2 hours at 8 °C with 5 ml glutathione agarose (Sigma), pre-equilibrated in PBS. The glutathione agarose was washed two times with PBS and once with 50 mM Tris-HCl, pH 8.0. Elution of GST or GST-FAT10 was performed by incubating the glutathione agarose with 2 ml 50 mM Tris-HCl, pH 8.0, supplemented with 5 mM glutathione. Eluted protein lysates were subsequently desalted using PD-10 columns (GE Healthcare), as described by the manufacturer. Recombinant SUMO E2 (UBC9), MDYKDDDDK-tagged UBA6 (FLAG-UBA6), 6His-UBE1L (His-UBE1L), SUMO-1, 6His-ISG15 (His-ISG15), linear di-ubiquitin (linear di-Ub) and 6His-ubiquitin (His-Ub) were purchased from Enzo Lifesciences (Lausen, Switzerland). 6His-SUMO-2 was purchased from BostonBiochem. Expression plasmids pSUMO-HA-UBE1[30]; pSUMO-FAT10[51]; pSUMO-FAT10 (C0)[24]; pSUMO-FAT10-AV[38] and pSUMO-FAT10ΔGG[38] were used to purify HA-UBE1 and the respective untagged FAT10 variants. Briefly, expression was performed in E. coli BL21(DE3) overnight at 21 °C upon induction of protein expression with 0.4 mM IPTG. Cells were harvested and resuspended in 5 ml of binding buffer (20 mM Tris-HCl, pH 7.5, 150 mM NaCl, 20 mM imidazole, 1 mM TCEP and 1 tablet/100 ml cOmplete™, Mini, EDTA-free Protase Inhibitor Cocktail Tablet (Roche, Rotkreuz, Switzerland))/g bacterial pellet. Cells were lysed with at least two cycles at 2,5 kbar in a cell disrupter (Constant Cell Disrupter TS, Constant Systems Ltd.). Lysates were cleared by two filtration steps through filters with 1.2 µm and 0.45 µm particle pores, respectively, and loaded onto a pre-equilibrated 5 ml HisTrap FF column (GE Healthcare) using AektaExplorer (GE Healthcare) driven by the UNICORN software (GE Healthcare). Unspecifically bound proteins were removed by washing with ten column volumes of binding buffer and 5 column volumes of binding buffer containing 5% of elution buffer (20 mM Tris-HCl, pH 7.5, 150 mM NaCl, 0,5 M imidazole, 1 mM TCEP). Subsequent elution was performed with five column volumes of elution buffer. Imidazole was removed by size exclusion chromatography (HiPrep 26/10 desalting column, GE

Healthcare) preequilibrated in binding buffer. Elution was performed with binding buffer until baseline. The 6xHis-SUMO-tag was removed by incubating the eluted proteins with 4 µg ULP-1–6xHis/mg protein overnight at 4 °C on a roller and subsequent loading onto a 5 ml HisTrap FF column equilibrated in binding buffer. Elution of untagged recombinant HA-UBE1 or FAT10 variants was achieved by collecting the flow through while loading the sample onto the column. A final polishing step was performed by cation exchange chromatography using a HiScreen Capto SP ImpRes (GE Healthcare) preequilibrated in binding buffer (20 mM Tris-HCl, pH 7.5, 150 mM NaCl). Elution of recombinant proteins was achieved by a gradient elution (0–50%) with 20 column volumes of elution buffer (20 mM Tris-HCl, pH 7.5, 2 M NaCl). A final buffer exchange with a HiPrep 26/10 column was performed and recombinant proteins were stored at −80 °C in storage buffer (20 mM Tris-HCl, pH 7.5, 150 mM NaCl, 5% glycerol). Recombinant AOS1/UBA2 was purified as described[49]. In short, pET11d-UBA2 and pET28α-His-AOS1 were transformed into E. coli BL21(DE3) and simultaneous expression of AOS1 and UBA2 was induced with 1 mM IPTG for six hours at 25 °C. Cells were harvested and resuspended in 5 ml lysis buffer (20 mM Tris-HCl, pH 7.5, 300 mM NaCl, 10 mM imidazole, 2 mM TCEP and 1 tablet/100 ml cOmplete™, Mini, EDTA-free Protase Inhibitor Cocktail Tablet (Roche, Rotkreuz, Switzerland))/g bacterial pellet. Upon mechanical lysis as described above, PMSF was added to a final concentration of 1 mM. Lysates were cleared by centrifugation (30,000 × g, 4 °C, 30 min), filtered through a 0.45 µm filter and incubated with 6 ml Ni$^{2+}$ beads (Takara), pre-equilibrated in binding buffer (20 mM Tris-HCl, pH 7.5, 300 mM NaCl, 10 mM imidazole, 1 mM TCEP) for 1 hour at 8 °C with rolling. Beads were subsequently loaded onto a 1/10 EconoColumn (BioRad) and washed with wash buffer (20 mM Tris-HCl, pH 7.5, 300 mM NaCl, 20 mM imidazole, 1 mM TCEP) until baseline. Elution of AOS1/UBA2 was performed with three column volumes of elution buffer (20 mM Tris-HCl, pH 7.5, 300 mM NaCl, 250 mM imidazole, 1 mM TCEP). The eluate was further subjected to size exclusion chromatography (HiLoad 16/60 Superdex 200 prep grade) and anion exchange chromatography (Resource Q 1 ml), desalted using PD10 desalting columns (GE Healthcare), and stored at −80 °C in 40 mM Tris-HCl pH 7.5, 100 mM NaCl, 20 mM MgCl$_2$ and 1 mM TCEP. Recombinant HA-SUMO-1 was purified from E. coli BL21(DE3) transformed with pET23a-HA-SUMO-1. Protein expression was performed upon addition of 0.4 mM IPTG for 20 h at 21 °C. Cells were harvested, resuspended in 5 ml of binding buffer (50 mM Tris-HCl, pH 7.5, 25 mM NaCl, 1 mM TCEP and one tablet / 100 ml cOmplete™, Mini, EDTA-free Protase Inhibitor Cocktail Tablet (Roche, Rotkreuz, Switzerland))/ / g bacterial pellet and lysed by mechanical disruption as described above. PMSF was added to a final concentration of 1 mM before the lysate was cleared by centrifugation and filtered using a 0.45 µm filter. As a first purification step by anion exchange chromatography, the lysate was incubated with 4 ml Q sepharose FF-beads (GE Healthcare), washed with five column volumes of binding buffer, five column volumes of wash buffer (50 mM Tris-HCl, pH 8.8, 25 mM NaCl, 1 mM TCEP) and eluted with elution buffer (50 mM Tris-HCl, pH 7.5, 500 mM NaCl, 1 mM TCEP). Upon a subsequent size exclusion chromatography (HiLoad 16/60 Superdex 75 prep grade), HA-SUMO-1 was stored at −80 °C in 20 mM HEPES pH 7.3, 110 mM KOAc, 2 mM Mg(OAC)$_2$, 150 mM NaCl, 5% glycerol and 1 m TCEP. JunB-HA-His was purified from E. coli BL21 (DE3) transformed with pET24α-JunB-HA-6His. Upon induction of protein expression with 0.4 mM IPTG, cells were grown for 5 hours at 30 °C, harvested and dissolved in 5 ml lysis buffer (20 mM Tris-HCl, pH 7.5, 150 mM NaCl, 20 mM imidazole, 2 mM TCEP, 10% Glycerol, 1 tablet/100 ml cOmplete™, Mini, EDTA-free Protase Inhibitor Cocktail Tablet (Roche, Rotkreuz, Switzerland)) / g bacterial pellet. After mechanical lysis as described above, PMSF was added to a final concentration of 1 mM. The lysate was centrifuged for 30 min at 32,000 × g and filtered through a 0.45 µm filter. Proteins were purified in a first step using Ni$^{2+}$ Immobilized Metal Affinity Chromatography (HisTrap 5 ml, GE Healthcare) preequilibrated in 20 mM Tris-HCl, pH 7.5, 150 mM NaCl, 20 mM imidazole, 1 mM TCEP and eluted with 50% of 20 mM Tris-HCl, pH 7.5, 150 mM NaCl, 500 mM imidazole, 1 mM TCEP, using an Aekta Explorer 10 System. In a second step, the collected fractions were buffer exchanged into storage buffer (20 mM Tris-HCL, pH-7.5, 150 mM NaCl, 1 mM TCEP, 10% Glycerol) by size exclusion chromatography (HiPrep Desalting 26/10, GE Healthcare).

In vitro SUMOylation and FAT10ylation assays were performed with the below mentioned protein amounts for 30–40 min at 30 or 37 °C in 1x in vitro buffer (20 mM Tris-HCl (pH 7.6), 50 mM NaCl, 10 mM MgCl$_2$, 4 mM ATP, 0.1 mM dithiothreitol (DTT), 1x protease inhibitor mixture (cOmplete™, Mini, EDTA-free Protase Inhibitor Cocktail Tablets (Roche, Rotkreuz, Switzerland)). Reactions were stopped by addition of 5x gel sample buffer with (reducing) or without (non-reducing) 4% 2-mercaptoethanol, and boiled. Proteins were separated on 12.5% Laemmli SDS PAGE or on 4–12% NuPage Bis/Tris gradient gels (Invitrogen) and subjected to western blot analysis with the antibodies indicated. Protein amounts used were as follows: Fig. 2a: 7.9 µM JunB-HA-His, 6.2 µM FAT10, 7.4 µM FAT10-AV, 42 nM FLAG-UBA6, 0.2 nM AOS1/UBA2, 1.4 µM UBC9, 8.3 µM SUMO-1, 12.5 µM His-SUMO-2. Figure 5b: 0.2 nM AOS1/UBA2, 8.3 µM SUMO-1, 6.2 µM FAT10, 6.2 µM FAT10-AV, 6.2 µM His-ISG15, 6.2 µM linear di-Ub, 15 µM His-Ub; Fig. 5c: 0.8 µM AOS1/UBA2, 8.3 µM SUMO-1 and 6.2 µM FAT10. Figure 5d: 0.3 µM AOS1/UBA2, 1.4 µM UBC9, 8.3 µM SUMO-1, 5.5 µM FAT10. Figure 5e: 5.5 µM His-Ub, 0.6 µM HA-UBE1, 7.4 µM FAT10. Figure 5f: 1.4 µM 6His-ISG15, 0.22 µM 6His-UBE1L, 7.4 µM FAT10. Figure 6a: 0.34 µM AOS1/UBA2, 8.3 µM SUMO-1, 11 µM FAT10, 0.1 µM FLAG-UBA6, 1.1 µM 6His-USE1; Fig. 6b: 0.34 µM

AOS1/UBA2, 8.3 μM SUMO-1, 11 μM FAT10/-ΔGG/-AV, each; Fig. 6d: same as in Fig. 6a but additionally with 1.5 μM AOS1/UBA2-C173A, 0.7 μM UBC9; Fig. 7c: 0.29 μM AOS1/UBA2, 1.4 μM UBC9, 0.12 μM FLAG-UBA6, 9 μM FAT10, 14 μM HA-SUMO-1, 7.9 μM JunB-HA-His.

In vitro co-immunoprecipitation was performed after incubating 25 nM AOS1/UBA2 with 0.7 μM of each FAT10 variant in a total volume of 540 μl 1x in vitro buffer (in presence or absence of 4 mM ATP), using the monoclonal FAT10-reactive antibody (clone 4F1[29]) bound to protein A sepharose for three hours at 8 °C. Beads were washed twice with NET-TN and NET-T buffer, boiled in 5x loading buffer as described in ref. [52] and subjected to western blot analysis using the antibodies indicated.

**Cell culture, transfection, and endogenous FAT10 expression**. HEK293 (ATCC® CRL-1573™), HEK293T (ATCC® CRL-11268™), HEK293 CRISPR/Cas9 FAT10ko cells[38], and stable FLAG-FAT10 expressing HEK293T cells[23], were cultivated in HyClone™ Iscove`s modified Dulbecco`s Medium (IMDM) (VWR International GmbH, Dietikon, Switzerland), supplemented with 10% fetal calf serum (Gibco/Thermo Fisher Scientific), 1% stable glutamine (100x, 200 mM), and 1% penicillin/streptomycin (100x) (both from Biowest/VWR). HCC cell lines HepG2 (Sigma–Aldrich, ECACC 85011430) and HepG3[53] as well as MCF-7 cells (ATCC® HTB-22™) were cultivated in HyClone™ Dulbecco's High Glucose Modified Eagles Medium (DMEM) (VWR International GmbH, Dietikon, Switzerland) supplemented with 10% fetal calf serum (Gibco/Thermo Fisher Scientific), 1% non-essential amino acids, 1% stable glutamine (100x, 200 mM) and 1% penicillin/streptomycin (100x) (all three from Biowest/VWR). HCT116 cells (ATCC® CCL-247™) were cultivated in RPMI supplemented as described for IMDM. A HEK293 CRISPR/Cas9 UBA6-ko cell line was generated by transfection of HEK293 cells with pCMV-Cas9-GFP containing UBA6-specific gRNA (Sigma–Aldrich). Twenty-four hour after transfection, single GFPhigh cells were sorted using BD FACS Aria™ Ilu (BD Biosciences) and cultivated as described above for HEK293 cells. Cell clones with a successful UBA6 knockout were identified by western blot analysis, using a UBA6-specific polyclonal antibody (Ref. [29] and Enzo Lifesciences, Lausen, Switzerland). Induction of endogenous FAT10 expression was performed as previously described[38,52]. In detail, HEK293 cells were seeded with a density of $1–2 \times 10^6$ cells/cell culture dish (10 cm) and grown for additional 24 h at 37 °C and 5% $CO_2$. Medium was removed and new medium, supplemented with 300 U × mL$^{-1}$ IFN-γ and 600 U × mL$^{-1}$ TNF (both from Peprotech GmbH, Hamburg, Germany) was added for another 24 h before harvesting the cells. Transient transfection of cells was performed with TransIT-LT1 transfection reagent (Mirus) as described before[38,52]. Briefly, HEK293 cells were seeded with a density of $1–2 \times 10^6$ cells/cell culture dish (10 cm). After 24 h, medium was removed and fresh medium was added to the cells. The transfection mixture was prepared as follows: 600 μl of IMDM without additives was incubated with 19 μl TransIT-LT1 transfection reagent (Mirus), vortexed briefly and incubated five minutes at room temperature. A total amount of 6.6 μg of the respective plasmids were added, vortexed and incubated for additional 15 min at room temperature before the mixture was added dropwise to the cells. Cells were harvested 24 h after the transfection. All cells were regularly tested to be negative for Mycoplasma infection using the MycoAlert™ kit (Roche).

**Cell extracts, immunoprecipitation, CHX chase, and antibodies**. Cell extracts using NP-40 lysis buffer for immunoprecipitation were prepared as described previously[38,52]. Briefly, cells were harvested by a standard trypsinization protocol, centrifuged for 4 min at $300 \times g$ and pellets were lysed in 1.2 ml NP-40 lysis buffer (20 mM Tris-HCl, pH 7.6, 50 mM NaCl, 10 mM $MgCl_2$ and 1% NP-40, supplemented with 1x protease inhibitor mix (cOmplete™, Mini, EDTA-free Protease Inhibitor Cocktail Tablets (Roche, Rotkreuz, Switzerland)) for 30 min on ice. Lysates were cleared by centrifugation (30 min, $20,000 \times g$, 4 °C) and subjected to immunoprecipitation using either 25 μl of preequilibrated anti-HA agarose conjugate HA-7 (Sigma) for immunoprecipitation of HA-tagged proteins, or with 30 μl of pre-equilibrated Protein A-Sepharose and 5 μg of the monoclonal FAT10-reactive antibody 4F1 (Enzo Lifesciences and[29]) for immunoprecipitation of endogenous FAT10, for 2 h at 8 °C with rolling. Beads were washed twice with 1 ml of NET-TN wash buffer (50 mM Tris-HCl, pH 8.0, 650 mM NaCl, 5 mM EDTA, 0.5% Triton X-100) and subsequently twice with NET-T buffer (50 mM Tris-HCl, pH 8.0, 150 mM NaCl, 5 mM EDTA, 0.5% Triton X-100). The wash buffer was removed completely and beads were directly boiled in 25 μl of a standard 5x SDS gel sample buffer, supplemented with 4% β-mercaptoethanol. Cycloheximide (CHX) chase experiments were exactly performed as recently outlined[24]. Briefly, before harvesting, cells were treated with a final concentration of 50 μg/ml CHX (stock solution 50 mg/ml in DMSO) for the indicated time points and additionally with 10 μM proteasome inhibitor MG132 (stock solution 10 mM in DMSO), where indicated. Crude cell extracts for visualization of SUMO conjugates were prepared as follows: Cells were grown in 10 cm cell culture dishes to a confluency of 80% (~$2 \times 10^6$ cell) and treated or not with 300 U × mL$^{-1}$ IFNγ and 600 U × mL$^{-1}$ TNF for additional 24 h, as indicated. For visualization of PML SUMOylation, cells were treated additionally with $As_2O_3$, as described previously[54]. In detail, a 280 mM stock solution of $As_2O_3$ was prepared in 1 M NaOH and further diluted in 50 mM NaCl/10 mM Tris-HCl, pH 7.6, to a final stock concentration of 1 mM. Cells were incubated for one hour with a final concentration of 1 μM $As_2O_3$, before

harvesting. Medium was removed and cells were carefully washed with 10 ml/dish of phosphate buffered saline (PBS) supplemented with 10 mM N-ethyl maleimide (NEM). Cells were directly lysed within the cell culture dish by adding 400–500 μl 5x standard gel sample buffer, containing 10 mM NEM. Cell lysates were removed with a cell scraper, collected in 1.5 ml reaction tubes, sonified for 20 s and boiled for 4 min. Before loading onto the gel, samples were centrifuged at maximum speed at $20,000 \times g$ for one minute. Detection and immunoprecipitation of endogenous FAT10 was performed using a mouse monoclonal FAT10-reactive antibody (clone 4F1[29], 1:50.000) or a rabbit polyclonal FAT10-reactive antibody (1:1000)[33]. JunB was detected with a JunB-reactive rabbit monoclonal antibody (Abcam, ab128878, 1:1000). UBA6 was detected with a rabbit polyclonal antibody (1:1000, Enzo Lifesciences, BML-PW0525). Anti-β-actin (Abcam, ab6276, mouse monoclonal, 1:5000) was used as loading control. For the detection of HA- or FLAG-tagged proteins, directly peroxidase-coupled antibodies anti-FLAG-HRP (clone M2, Sigma, 1:3000) or anti-HA-HRP (clone HA-7, Sigma, 1:4000) were used. 6His-tagged proteins were detected using a directly peroxidase-coupled anti-6His anti-body (1:5000, SIGMA, A7058-1VL). SUMO was detected with anti-SUMO-1 (#4930, rabbit polyclonal, Cell Signaling, 1:1000), or anti-SUMO-2/3 (clone 18H8, rabbit monoclonal) (#4971, Cell Signaling, 1:1000). Human AOS1/UBA2 was visualized using antibodies anti-SAE2 (UBA2) (Abcam, ab185955. rabbit monoclonal, 1:1000) or anti-SAE1 (AOS1) (Abcam, ab185949) rabbit monoclonal, 1:5000). Human SUMO E2 (UBC9) was detected with a rabbit polyclonal anti-UBC9 antibody (1:1000, kindly provided by Andrea Pichler, Freiburg, Germany). Detection of human PML was performed with a rabbit polyclonal antibody (Abcam, ab179466, 1:2000). Uncropped western blot scans are shown in the Source Data file.

**Affinity measurements with the Octet system**. GST, GST-FAT10, GST-SUMO-1, and AOS1/UBA2 were dissolved in buffer TNT (50 mM Tris HCl pH 7.6, 100 mM NaCl, 0.1% Triton X-100). The same buffer was used for the measurement at the Octet (Forte Bio). Two sensors were always loaded with 0.1 mg/ml AOS1/UBA2. As ligands, GST-FAT10 was measured versus GST, and GST-SUMO-1 was measured versus GST. The ligands were used at concentrations of 0.5 mg/ml, 0.25 mg/ml. 0.1 mg/ml and 0.05 mg/ml. Instrument settings were as follows: Baseline1 in TNT for 100 s, loading of AOS1/UBA2 for 300 s. Baseline2 in TNT for 100 s, association for 600 s, dissociation for 600 s, regeneration in 0.1 M Glycine buffer pH 2.7 for 30 s, neutralization in TNT for 30 s, activation in 0.1 M $NiSO_4$ for 120 s. This sequence was repeated five times with the different amounts of the ligands.

**Densitometric analysis and statistical analysis**. For quantification of protein amounts on immunoblots, ECL signals were quantified with the program Quantity One (BioRad, Cressier, Switzerland). Signals from JunB-HA-FLAG-FAT10 conjugate, JunB-HA or FLAG-FAT10 from five independent experiments were calculated and normalized to the respective signal of the loading control β-actin in the lysate. The value of 0 h CHX was set to unity and the other values were calculated accordingly. Values in the figure are given as mean ± s.e.m. Analysis of the Pearson`s coefficient, representing the probability of colocalization of PML and SUMO-1 as well as counting of PML bodies was calculated with Fiji ImageJ. Significance was calculated using an unpaired, non-parametric Mann-Whitney test with Instat Statistics (Graph Pad Software, San Diego, CA, USA). A two-tailed P-value of <0.0001 was considered to be highly statistically significant.

**Real-Time RT-PCR**. RNA was isolated using the RNeasy kit (QIAGEN) and cDNA was reverse transcribed using the High-Capacity cDNA Reverse Transcription Kit (Applied Biosystems/Life Technologies). Real-time RT-PCR was performed on an Applied Biosystems 7900-HT Fast Real-Time PCR Cycler using Fast SYBR® Green Master Mix (Applied Biosystems/Life Technologies) with specific primers (QuantiTect Primer Assays, QIAGEN) for SUMO-1 (Hs_SUMO1_1_SG), SUMO-2 (Hs_SUMO2_1_SG), UBA2 (Hs_UBA2_1_SG), AOS1 (Hs_SAE1_1_SG), or UBC9 (Hs_UBE2l_1_SG). GAPDH (Hs_GAPDH_1_SG) served as housekeeping gene.

**Luminescence-based ATP assay**. To determine ATP consumption in in vitro SUMO and FAT10 activation assays, the Luminescence ATP Detection Assay Kit (Abcam, ab113849) was used, as described by the manufacturer. Briefly, in vitro SUMO or FAT10 activation assays were performed with 0.3 nM AOS1/UBA2 and 3.3 nM SUMO-1 or 3.3 nM FAT10 or FAT10-AV in 1x in vitro buffer (20 mM Tris-HCl, pH 7.6, 50 mM NaCl, 10 mM MgCl2, 4 μM ATP, and 0.1 mM DTT) for 30 min at 30 °C. Samples were subsequently treated as described by the manufacturer and luminescence was measured in a multiwell plate reader SPARK 10 M (TECAN). To be in a linear range, the minimal amount of ATP needed for in vitro SUMO activation had been determined to be 4 μM in a preceding titration experiment.

**Confocal microscopy**. Cells were seeded directly in presence or absence of IFN-γ/TNF-α on glass slices in 6-well plates to a confluency of approximately 50%. Twenty-four hour later, cells were washed in PBS and fixed for 10 min in 4% formaldehyde /PBS. After two washing steps in PB buffer (PBS, 3% BSA), cells were permeabilized for five minutes in PBGT buffer (PBS, 0.2% Triton X-100, 20 mM

Glycine and 3% BSA). After two subsequent washing steps with PBG buffer (PBS, 20 mM Glycine, 3% BSA), cells were incubated for 2 hours in the dark with the following primary antibodies, diluted in PBG: anti-SUMO-1-21C7 mouse monoclonal antibody[55] (1:50), anti-PML (Abcam, ab179466, rabbit mAb, 1:500) and Phalloidin Alexa Fluor® 647 (Invitrogen, A22287, 1:250). Nuclear staining was performed with Nuclear Green DCS1 (Abcam, ab138905, 1:2000). Slices were carefully washed four times with PB buffer and then incubated for one hour in the dark with the secondary antibodies anti-mouse-Alexa 488 (Life Technologies, 1:250) or anti-rabbit-Alexa 568 (Life Technologies, 1:250). After four wash steps with PBS, slices were imbedded with fluorescent mounting medium (Dako, E3023) and imaged on a Leica TCS SP5 II laser scanning microscope using a ×63/1.4 NA oil-immersion objective (Leica).

**FRET-based SUMOylation assay.** The FRET-based SUMOylation assay was performed as described previously[36]. Shortly, 2 nM AOS1/UBA2, 150 nM UBC9, 1.35 µM CFP-RanGAP1-tail, 1.35 µM YFP-SUMO1 were incubated in FRET buffer (TB buffer (110 mM KOAc, 20 mM HEPES, pH 7.3, 2 mM Mg(OAc)2, 1 mM EGTA, 1 mM DTT, 1 µg/ml each of leupeptin, pepstatin and aprotinin) supplemented with 0.2 mg/ml ovalbumin and 0.05% Tween20) without or with the indicated amounts of unlabeled SUMO-1, FAT10, FAT10-AV, or Ub for 10 min at 30 °C before the addition of 1 mM ATP (final concentration; ATP stock solution 5 mM in TB buffer). At the desired time points, samples were excited at 430 nm and fluorescence emission at 485 and 525 nm were recorded with an integration time of 20 ms. Data are represented as the emission rate of 525 nm/485 nm and directly represent the rate of SUMO-1 conjugation to RanGAP.

**Chemical crosslinking coupled to mass spectrometry (XL-MS).** Proteins were crosslinked and measured essentially as described[56]. In short, a molar ratio of AOS1/UBA2 : FAT10/or FAT10–AV of ~1 : 6 were incubated as follows: FAT10 or FAT10-AV (1.8 µg/µl or 0.54 µg/µl stored in 50 mM Tris-HCl, pH 7.5, 150 mM NaCl, 5% glycerol, 1 mM TCEP, as described[51]) and 100 µg of AOS1/UBA2 (1.8 µg/µl stored in 40 mM Tris-HCl, pH 7.5, 100 mM NaCl, 20 mM MgCl₂, 1 mM TCEP, as described[49]), were incubated for 10 min on ice. Tris was used as all the biochemical assays were performed in Tris containing buffer and in order to replicate these experimental conditions as close as possible for the crosslinking experiments. A comparison of AOS1/UBA2 plus FAT10 linked in Tris and after buffer exchange into 20 mM HEPES, pH 7.5 confirmed that both buffer conditions yielded a highly comparable number of crosslinks and that the presence of Tris in our buffer did not preclude the formation of lysine-lysine crosslinks (Supplementary Data 5). Proteins were crosslinked by addition of H12/D12 BS3 (Creative Molecules) at a final ratio of 1 nmol BS3/1 µg protein for 30 min at 37 °C while shaking at 650 rpm in a Thermomixer (Eppendorf). After quenching by addition of ammonium bicarbonate to a final concentration of 50 mM and incubation for 10 min at 37 °C, samples were dried, dissolved in 8 M urea to a final concentration of 1 mg/ml, reduced with TCEP at a final concentration of 2.5 mM, alkylated with iodoacetamid at a final concentration of 5 mM and digested overnight with trypsin (Promega V5113) in 1 M urea (diluted with 50 mM ammonium bicarbonate) at an enzyme-to-substrate ratio of 1:40. Digested peptides were separated from the solution and retained by a solid phase extraction system (SepPak, Waters) and then separated by size exclusion chromatography prior to liquid chromatography (LC)-MS/MS analysis on an Orbitrap Fusion Tribrid mass spectrometer (Thermo Scientific). Data were searched using *xQuest* in ion-tag mode with a precursor mass tolerance of 10 ppm. For matching of fragment ions, tolerances of 0.2 Da for common ions and 0.3 Da for crosslink ions were applied. Crosslinks which were identified with deltaS < 0.95 and lD-Score > 25 and an assigned FDR as calculated by xProphet below 0.05[57] were visualized by xiNET software[58] and are shown in Fig. 7d and e (see also Supplementary Data 1 and 2).

**Quantitative chemical crosslinking (q-XL-MS).** Quantitative chemical crosslinking coupled to mass spectrometry (q-XL-MS) was carried out essentially as described[59]. In short, approximately 100 µg of AOS1/UBA2 were incubated either on its own, together with SUMO-1 (molar ratio 1:4) or with SUMO-1 and FAT10 (molar ratio 1:4:4) for 15 min at 30 °C in 150 µl total volume of 1 x reaction buffer (20 mM Tris-HCl pH 7.6, 50 mM NaCl, 10 mM MgCl₂, 4 mM ATP, 0.1 mM DTT) prior to crosslinking by addition of H12/D12 BS3 (Creative Molecules) at a ratio of 1 nmol/1 µg protein for 30 min at 37 °C while shaking at 650 rpm in a Thermomixer (Eppendorf). After quenching by addition of ammonium bicarbonate to a final concentration of 50 mM and incubation for 10 min at 37 °C, samples were reduced, alkylated, and digested with trypsin (details see above). Digested peptides were separated from the solution and retained by a solid phase extraction system (SepPak, Waters), and then separated by size exclusion chromatography prior to liquid chromatography (LC)-MS/MS analysis on an Orbitrap Fusion Tribrid mass spectrometer (Thermo Scientific). Amounts of potential crosslinks were normalized prior to MS by measuring peptide bond absorption at 215 nm for each fraction. Crosslinked samples were prepared in triplicates (i.e., three separate crosslink batches for each condition) and each of these was measured as technical duplicates. Crosslinks which were identified with deltaS < 0.95 and ID-Score ≥ 20 were used as input for q-XL-MS analysis with *xTract*[60]. For q-XL-MS analysis with *xTract*, the

chromatographic peaks of identified crosslinks in the samples with AOS1/UBA2, AOS1/UBA2/SUMO-1 or AOS1/UBA2/SUMO-1/FAT10 (n = 3, each sample analyzed additionally as technical duplicate) were integrated and summed up over different peak groups (taking different charge states and different unique crosslinked peptides for one unique crosslinking site into account). Only high-confidence crosslinks that were identified consistently in both, light and heavy labeled states (*xTract* settings violations was set to 0), were selected for further quantitative analysis. If a peptide was detected in only one condition (e.g., only in the reference experiment or vice versa), the fold change was estimated on the basis of the minimum detectable signal intensity (1e3 for Orbitrap Fusion Tribrid mass spectrometer), and instead of the area, the intensity of the first isotope was used for the comparison. This is indicated in Supplementary Data 3 and 4 in the column 'imputed values'. Changes in crosslinking abundance are expressed as log₂ ratio (e.g., abundance state 1, AOS1/UBA2/SUMO-1 was quantified versus abundance state 2, AOS1/UBA2). The p value indicates the regression between the two conditions. In this study, only links with an lD-Score ≥ 28 and an assigned FDR as calculated by xProphet below 0.05[57] that showed a change of log2ratio ≥ ±1 and a p-value of ≤0.01 were considered significant changes in abundances and are shown in green and red in the 2D visualizations (Supplementary Fig. 8b–d, Supplementary Data 3 and 4), respectively. All other changes were considered insignificant and are shown in gray. Crosslinks were visualized by xiNET software[58] using additional in-house scripts for the analysis and representation of quantitative crosslink information.

**Enrichment of crosslinked peptides.** Crosslinked peptides were enriched by size exclusion chromatography (SEC) on an ÄKTAmicro chromatography system (GE Healthcare) using a Superdex™ Peptide 3.2/30 column (GE Healthcare) at a flow rate of 50 µl/min of the mobile phase (water/acetonitrile/trifluoroacetic acid 70%/30%/0.1%, vol/vol/vol). UV absorption at a wavelength of 215 nm was used for monitoring the separation. The eluent was collected in fractions of 100 µl in a 96-well plate. The two fractions 1.2–1.3 ml and 1.3–1.4 ml were collected, dried and further analyzed by LC-MS/MS. For the FAT10-AV mutant also the fraction 1.1–1.2 ml was analyzed by LC-MS/MS.

**LC-MS/MS analysis.** Samples fractionated by SEC were re-dissolved in an appropriate volume of MS buffer (acetonitrile/formic acid 5%/0.1%, vol/vol) according to their UV signal. Peptides were separated on an EASY-nLC 1200 (Thermo Scientific) system equipped with a C18 column (Acclaim PepMap 100 RSLC, length 15 cm, inner diameter 50 µm, particle size 2 µm, pore size 100 Å, Thermo Scientific). Peptides were eluted at a flow rate of 300 nl/min using a 60 min gradient starting at 94% solvent A (water/acetonitrile/formic acid 100%/0%/0.1%, vol/vol/vol) and 6% solvent B (water/acetonitrile/formic acid 20%/80%/0.1%, vol/vol/vol) for 4 min, then increasing the percentage of solvent B to 44% within 45 min followed by a 1 min step to 100% B for additional 10 min. The mass spectrometer was operated in data-dependent-mode with dynamic exclusion set to 60 s and a total cycle time of 3 s. Full scan MS spectra were acquired in the Orbitrap (120.000 resolution, 2e5 AGC target, 50 ms maximum injection time). Most intense precursor ions with charge states 3–8 and intensities greater than 5e3 were selected for fragmentation using CID with 35% collision energy. Monoisotopic peak determination was set to peptide and MS/MS spectra were acquired in the linear ion trap (rapid scan rate, 1e4 AGC target).

**Reporting summary.** Further information on research design is available in the Nature Research Reporting Summary linked to this article.

## Data availability

The MS raw files, databases containing protein fasta sequences for analysis with xQuest as well as xQuest result- / xTract input- files (xtract.csv) and the xTract result files (analyzer.quant.xls) have been deposited to the ProteomeXchange Consortium via the PRIDE[61] partner repository with the dataset identifier PXD012592. The source data underlying Figs. 1d, 3a, 3e, 3f, 6c, and Supplementary Figs. 1, and 3a, as well as all uncropped western blot scans are provided as a Source Data file. All other data are available from the corresponding author on reasonable request.

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

## Acknowledgements

We are grateful to Mark Piechaczyk for providing JunB expression plasmids, Annette Flotho for providing pET23a-HA-SUMO-1, and Edith Uetz von Allmen for single cell sorting. This work was supported by the Swiss Velux Foundation (projects 855 and 1029), the DFG Collaborative Research Center (SFB) 969, TP C01 (A.A. and M.G.) and the graduate school KoRS-CB (C.S. and S.R.) at the University of Konstanz. F.S. is funded by the German Science Foundation Emmy Noether Program (STE 2517/1-1) and the DFG Collaborative Research Center (SFB) 969, TP A06. F.M. acknowledges funding by the DFG (SFB1036, TP15 and TRR186, TP18).

## Author contributions

A.A. designed experiments, wrote the manuscript, acquired resources, and performed all experiments except Fig. 3a and Supplementary Fig. 1, which were performed by N.S.V, Figs. 7d and 7e, Supplementary Figs. 8b-d and Supplementary Data 1–5, which were performed by C.S.; S.R. identified JunB as FAT10 substrate and performed initial experiments for JunB FAT10ylation; N.C. purified recombinant proteins and provided technical help; G.S. performed the experiments shown in Supplementary Fig.7; F.M. and F.S. analyzed data and supervised N.S.V and C.S., respectively; M.G. conceived experiments, refined the manuscript, and acquired resources.

## Competing interests

The authors declare no competing interests.
