## [Peer Review File · Nature Communications]

Reviewers' comments:

Reviewer #1 (Remarks to the Author):

In the article *"Regulating the regulator: the ubiquitin-like modifier FAT10 thwarts SUMO activation"* the authors describe new functionality for the cytokine-inducible ubiquitin-like modifier protein FAT10 in that it hinders activation of the anti-inflammatory ubiquitin-like modifier SUMO. They also demonstrate that the transcription factor JunB is a substrate of FAT10 and FAT10ylation of JunB occurs preferentially to SUMOylation. The authors utilize a number of affinity tagged (FLAG-tagged and HA-tagged) proteins with immunoprecipitation and site specific mutants to explore the interaction of FAT10 with JunB. They demonstrate that FAT10 modifies the same Lys residues on JunB as SUMO, however FAT10 prevents SUMOylation even in the absence of the FAT10 conjugation enzyme UBA6. They go on to demonstrate that FLAG-FAT10 expression leads to a bulk decrease in SUMOylation in HEK293 cells. They then use IFN- γ /TNF- α treatment of HepG2 and HepG3 cells to induce endogenous FAT10 and CRISPR/Cas9 FAT10 knockout cell lines to show it has the same effect of reducing SUMOylation as the FLAG-tagged version. The authors then investigated the mechanism by which FAT10 interferes with SUMO, and show that it interacts directly with the SUMO E1 activating enzyme UBA2/AOS1. They go on to demonstrate that FAT10 is activated by UBA2/AOS1, forming a thioester conjugate. They also demonstrate that although FAT10 is activated by UBA2/AOS1, this form of activated FAT10 is not subsequently transferred to an E2 conjugating enzyme and therefore does not result in FAT10ylation of proteins. Finally, the authors explore the interaction of FAT10 with UBA2/AOS1 with chemical cross-linking and mass spectrometry (XL-MS). They identify cross-links between lysine residues of FAT10 and both AOS1 and UBA2, although a greater number of links with AOS1. They then perform quantitative cross-linking between UBA2/AOS1 and SUMO1 in the absence or presence of FAT10 and conclude that FAT10 had no discernable impact on the conformation of UBA2/AOS1 although a number of the same Lys residues on AOS1 and UBA2 were identified as linked to both FAT10 and SUMO1 suggesting FAT10 and SUMO1 compete for the same binding interface.

Overall the authors provide a wealth of experimental data on FAT10, its interference with SUMO and the FAT10, UBA2/AOS1 interaction. The manuscript is on the whole well written and follows a logical progression of the investigation into this newly reported function for FAT10. The authors conclusions based on the data appear sound for the most part. I do have concerns about the authors interpretation of the XL-MS results and about how the cross-linking experiments were carried out which are described in detail below. While these concerns do not impact the broader story of FAT10 functioning to block SUMO activation, they could impact the conclusions of how this occurs due to the interaction between FAT10 and UBA2/AOS1 and therefore in my opinion, need to be clarified before this manuscript can be published.

The authors perform two sets of XL-MS experiments. One set as a discovery effort to identify cross-linked Lys between FAT10 and the FAT10-AV mutant with UBA2/AOS1, and the other set as quantitative XL-MS experiments to investigate the conformational impacts of FAT10 binding on UBA2/AOS1. I have concerns about the interpretation of data obtained by both sets of XL-MS experiments and how these experiments were carried out. At the heart of this is the confidence in cross-linked peptide identifications. The authors use the xQuest pipeline for assignment of peptides to the MS data and report score thresholds for the assignments they deemed confident. Of concern is that there is no mention as to how these score thresholds were determined and what they mean? It is now commonplace in XL-MS experiments, as in all MS-based proteomics, to report an estimate of false discovery rate (FDR). The authors do not provide an estimate of FDR or adequately describe how identifications were deemed confident. xQuest scores by themselves do not mean much. Why use different score thresholds used between the 1st set of XL-MS experiments and the 2nd quantitative XL-MS set? Reporting FDR would provide a consistent assessment for a score threshold and there are many software tools out there to provide this if xQuest does not. In the first set of experiments the authors identify cross-links between FAT10 and UBA2 and AOS1 and conclude that FAT10 inhibits SUMO activation mainly by interacting non-

covalently with AOS1. While it does appear that more cross-linked pairs of peptide and cross-linked sites were identified between FAT10 and AOS1, this doesn't necessarily mean that FAT10 is preferentially interacting with AOS1 compared with UBA2. The formation of Lys-Lys cross-links depends on many factors, including relative proximity and orientation, solvent accessibility, and reactivity of the linked Lys. The authors appear to equate more identified non-redundant cross-links with a greater abundance which is not an accurate assessment. It could be that there is just as much interaction between FAT10 and UBA2 as there is between FAT10 and AOS1 but due to the position of Lys residues within the structures not as many unique cross-links can be formed. Furthermore, additional FAT10-UBA2 links could be forming but just not detected by the authors strategy. These important points must be considered.

In the quantitative XL-MS experiments the authors use isotope labeled BS3 to look for conformational changes in AOS1/UBA2 that are potentially induced by the binding of FAT10. The authors use the data from these experiments to conclude that FAT10 binding has no impact on the conformational state of AOS1/UBA2. As presented it is not clear to me that this is an accurate conclusion to make. The authors seemingly arbitrarily selected links displaying a 2-fold or greater change to be indicative of a significant change? How was this threshold determined? It is very concerning to me that the quantified cross-links in supplementary tables 3 & 4 span very large ranges of ratios (TableS3 log₂ -13.69 to +17.5 and TableS4 log₂ -11.86 to +20.31). Do the authors really believe they are capable of accurately quantifying cross-links varying over 10 orders of magnitude? I find this extremely doubtful and think the quantitative analysis needs to be carefully revisited and thoroughly revised. Of additional concern is that the distributions of log₂ ratios in TableS3 and TableS4 are not normal nor centered at a log₂ of zero as one would expect for a situation where the majority of links do not change. I suspect that the authors just passed the data through xTract and accepted the output without much additional thought or analysis. Regardless it is not clear to me how they arrive at the conclusion that there is not a significant conformational change when the majority of the links quantified appear to have log₂ changes with a magnitude of greater than 1. Beyond just reporting the average log₂ ratios and associated p-values the authors should provide some measure of variance, such as a confidence interval or standard deviation for each log₂ value. Even granting the authors the benefit of the doubt, by comparing the quantified links shown in supplementary Fig 3 b&c there still appears to be differences that could indicate a significant conformational change. For example, in FigS3c there appears to be a link somewhere between K250 and K300 of UBA2 that is red (significantly decreased) that was not present in FigS3b. Why is this not indicative of a conformational change induced by FAT10?

As a final concern regarding XL-MS, in both cross-linking reactions why was Tris-HCl used as a buffer? This is very unusual as Tris will react with the NHS-esters in BS3 using up the reagent before it can react with Lys in the proteins, and also form Tris-protein cross-links potentially altering the proteins structure. This must be addressed and the authors need to justify why Tris was used here and that it has no negative impact on the results, such as inducing conformational changes in the proteins. Being that the Tris concentration is so high I would expect it to have a significant impact on the identified and quantified cross-links. It could very well preclude formation of Lys-Lys links that could be important in the interaction of FAT10 with UBA2/AOS1.

Specific comments

For clarity the authors should employ consistent use of SUMO E1 and/or UBA2/AOS1 throughout the manuscript. Once UBA2/AOS1 is defined as the SUMO E1 activating enzyme it should be used exclusively to avoid confusion.

Page 9 - "A similar result was obtained when the same experiments were performed in additional cancer cells lines such as HCT116 or MCF-7 (data not shown)."
This data should be shown, at least as a supplementary figure.

Page 10 - "A co-immunoprecipitation of recombinant FAT10 and SUMO E1 revealed that FAT10 directly interacted with UBA2/AOS1 in absence (Fig. 5a, lane 5) or presence of ATP (data not

shown).“

The data with ATP should be shown, at least as a supplementary figure. Does the presence of ATP impact the amount of the FAT10-UBA2/AOS1 interaction?

Methods Section

Page 23 - Chemical crosslinking coupled to mass spectrometry (XL-MS)

Authors state: “Proteins were crosslinked and measured essentially as described in 52.”

While it is fine to refer to a published protocol that is used for the cross-linking method, key details pertaining to this experiment need to be fully described. Currently they are not. Missing details that require clarification are:

Its stated that SUMO E1 - FAT10(-AV) are mixed at ~ 1:6 molar ratio, but no mention of the concentration of FAT10 and FAT10-AV in the stock solution?

What is the final concentration of all protein components and buffer composition and volume during the cross-linking reaction?

It is stated that both proteins are in Tris-HCl buffers (50 mM Tris for FAT-10 and 40 mM Tris for SUMO E1). Tris-HCl is an unusual choice of buffer for these experiments as it is generally incompatible with NHS-ester cross-linking due to the fact it contains primary amines that will react with the cross-linker. This is clearly stated in the protocol referred to in Ref 52 “Any additives containing amino groups must be avoided.”. Was a buffer exchange performed prior to cross-linking to remove the Tris? Otherwise the Tris would undoubtedly react with the BS3, consuming a large amount of the cross-linking reagent as well as potentially cross-link to the proteins themselves. Was this accounted for? Having a relatively high concentration of Tris present during the cross-linking reaction could impact the structures and interactions of the proteins by forming Tris-Lys cross-links. If these formed they should be detectable by MS analysis.

“were incubated for 10 min on ice. Proteins were crosslinked by addition of H12/D12 BS3 (Creative Molecules) at a final ratio of 1 nmol BS3/1 µg protein for 30 min at 37 °C while shaking at 650 rpm in a Thermomixer (Eppendorf).“

Why incubate on ice but then cross-link at 37C? What effect does temperature have on the structures and interactions? How was 1 nmol BS3/ 1 µg protein chosen? Was the ratio of cross-linker to protein optimized somehow?

“After quenching by addition of ammonium bicarbonate to a final concentration of 50 mM and incubation for 10 min at 37°C, samples were dried, reduced, alkylated and digested with trypsin.“

Is quenching necessary here? What amount of unreacted BS3 is expected after 30 min? Particularly if there was already 40-50 mM Tris-HCl in the cross-linking reaction buffer. What compounds and concentrations of reducing/alkylating reagents and trypsin were used?

“Digested peptides were separated from the solution and retained by a solid phase extraction system (SepPak, Waters) and then separated by size exclusion chromatography prior to liquid chromatography (LC)-MS/MS analysis on an Orbitrap Fusion Tribrid mass spectrometer (Thermo Scientific).“

Details on SEC and LC-MS analysis are missing. What conditions for the SEC? How many fractions were collected? LC-MS conditions? LC type, column, gradient, mass spec methods?

“Data were searched using xQuest in ion-tag mode with a precursor mass tolerance of 10 ppm. For matching of fragment ions, tolerances of 0.2 Da for common ions and 0.3 Da for crosslink ions were applied. Crosslinks which were identified with deltaS < 0.95 and ID-Score > 25 were visualized by xiNET software“

What protein database was searched? Some explanation as to what these scores mean and how the thresholds were determined is necessary. In XL-MS experiments, as in general mass spec based proteomics, it is now routine to estimate false discovery and report a false discovery rate (FDR). I would expect the authors to provide some estimate of FDR for the reported cross-links. FDR for cross-linking can also be reported at multiple levels (PSM, non-redundant peptide pair,

Lys-Lys pair, protein pair, see Anal Chem. 2017 Apr 4; 89(7): 3829–3833.). Some analysis/discussion related to FDR needs to be included.

“In short, approximately 100 µg of SUMO E1 were incubated either on its own, together with SUMO-1 (molar ratio 1:4) or with SUMO-1 and FAT10 (molar ratio 1:4:4) for 15 min at 30 °C in 1 x reaction buffer (20 mM Tris-HCl pH 7.6, 50 mM NaCl, 10 mM MgCl₂, 4 mM ATP, 0.1 mM DTT) prior to crosslinking by addition of H12/D12 BS3 (Creative Molecules) at a ratio of 1 nmol / 1 µg protein for 30 min at 37 °C while shaking at 650 rpm in a Thermomixer (Eppendorf).”

Why are different molar ratios between SUMO E1 and FAT10 used here compared with cross-linking section above? Also why is the incubation here 15 min at 30C vs. 10 min on ice previously? Again please provide concentrations for all proteins and stock/ final buffer compositions. Tris-HCl is used here again but apparently at a different concentration. For quantitative cross-linking separate reactions with H12-BS3 and D12-BS3 must have been performed but this is not clear as described. How and when did mixing of the H12 and D12 samples occur?

“Amounts of potential crosslinks were normalized prior to MS by measuring peptide bond absorption at 215 nm for each fraction.”

More details are needed here. What fractions the SEC? How many are there and what is the reproducibility of SEC fractionation between different injections? Why not mix/ normalize samples prior to digestion to remove variability introduced in downstream processing steps? How does the absorption at 215nm normalize for cross-links when much more abundant non cross-linked peptides must also contribute to this absorbance?

“In this study, only links with an ID-Score ≥ 28 that showed a change of $\log_2\text{ratio} \geq \pm 1$ and a p-value of ≤ 0.01 were considered significant changes in abundances and are shown in green and red in the 2D visualizations (Supplementary Fig. 3b and 3c, Supplementary Table 3 and 4), respectively.”

Why is a different ID-Score threshold used here? What about FDR? I’m confused about how significant changes were determined. A change in $\log_2\text{ratio}$ or $\log_2\text{ratio} \geq \pm 1$? How was 1 selected as a threshold? The \log_2 ratios provided in Tables S3 and S4 span a huge dynamic range (~ 10 orders of magnitude). I’m skeptical cross-linked peptides or any peptides for that matter can be quantified by MS approaches accurately over such a range.

Figure 7 d&e and Figure S3b&c – These xiNET figures, while easy to generate, may not be the most effective way to display the data. It is difficult for readers to really get a sense of what the cross-linking data is. It is nearly impossible to distinguish or pick out any specific Lys residue pair and it is easy to miss or discount a link that could be important. Why can the cross-links not be used to generate structural models for these proteins that could then be docked? This type of information could prove much more valuable. I also do not like that in the quantitative data it has been reduced down to increasing, decreasing or no change. This masks the fact that the quantified values span a huge range. At the very least they should be color coded on a scale with accompanying histograms of the \log_2 values so readers can assess what is being displayed.

Reviewer #2 (Remarks to the Author):

The authors characterize an interplay between two posttranslational protein modification (PTM) pathways involving SUMO and FAT10. They provide evidence from assays using cultured cells and in vitro biochemistry that FAT10 inhibits sumoylation. They demonstrate that FAT10 is able to interact with the SUMO E1 activating enzyme and can form a thioester intermediate with the active site cysteine. Non-covalent interaction between the SUMO E1 enzyme and FAT10 is sufficient to block SUMO activation, as demonstrated by competition experiments with FAT10 c-terminal mutants that cannot be activated. Chemical protein crosslinking and mass spectrometry analysis confirmed non-covalent interactions between FAT10 and both subunits of the SUMO E1 enzyme, and also supported a model in which FAT10 and SUMO compete for binding to the adenylation site

of AOS1.

PTMs play critical roles in regulating nearly all essential cell functions. SUMO and FAT10 are both important in controlling many facets of immunological response pathways. Cross-talk between SUMO and FAT10 could potentially enhance the complexity of regulation that is possible by these PTMs, however, it remains largely unexplored. The findings reported here, demonstrating that FAT10 has the ability to suppress sumoylation of individual proteins and also globally, is therefore a significant finding. The physiological significance remains to be explored, but the implications are nonetheless important. A number of minor comments for improving the study include:

- 1) It is suggested in the introduction that SUMO-2/3 "are mostly present as a free SUMO pool in the cytoplasm and utilized under a variety of stresses." While it is true that SUMO-2/3 modification is enhanced under stress, it is an oversimplification to say that it exists largely as a free pool in the cytoplasm under normal conditions.
- 2) It is interesting that FAT10 is largely conjugated to consensus site lysines modified also by SUMO. It would be interesting to know the authors' thoughts on the nature of this overlap in specificity.
- 3) It appears that no FAT10 E2 enzyme (USE1) was used in the assays shown in Figure 2a. Is it known that UBA6 is sufficient to transfer activated FAT10 onto JunB? If so, the authors should comment on this. If not, the authors should also comment on the finding that no E2 is required.
- 4) It is challenging to interpret the immunofluorescence data in Figure 3e without DAPI staining. Are PML foci in the FAT10-expressing cells in the cytoplasm? Is SUMO-1 in the nucleus or cytoplasm? It would be helpful to include DAPI.
- 5) The loss of co-localization of PML and SUMO-1 in cells expressing FAT10 is only suggestive of suppression of PML modification. To conclude that FAT10 leads to downregulation "of a single endogenous SUMO substrate such as PML", IP and western blot data needs to be included.

Reviewer #3 (Remarks to the Author):

See attached file

Review Aichem et al.

Despite the fact that FAT10 represents one of the classical ubiquitin like modifiers, remarkably little is known about its molecular function. In this manuscript, Aichem et al provide evidence that FAT10 interferes with SUMO modification by binding to UBA2/AOS1 and impairing the function of SUMO E1. More precisely, authors claim that SUMO-E1 thioester formation is disturbed by FAT10 binding. As FAT10 is strongly induced by IFN γ /TNF this would constitute an interesting mechanistic link between SUMOylation and inflammation. Authors also describe the FAT10 modification of JunB (Figure1). These are interesting findings which shed a new light on the function of FAT10 and would have considerable impact in the field. However, the conclusions from the manuscript in its current form are largely based on overexpression experiments and only limited data about the physiological consequences are provided. Furthermore some technical issues need to be clarified.

Specific points to address

- Figure 1. lane 8/9 authors claim that the FAT10 modification mediates proteasome degradation of modified JunB. However, not only the modified but also the intensity of the unmodified band increase upon MG132 treatment. Likewise, the stability of JunB is increased when the FAT10 non conjugatable form is transfected in the presence of MG132. Please clarify this discrepancy which is in contrast to the conclusion that FAT10 modification mediates proteasomal degradation

- In Figure 1b. authors show that endogenous JunB is modified by FAT10. This experiment should also be performed in the presence of MG132 to show that Fatylation influences stability of JunB. Beside JunB and beta actin, lysates should also be probed with anti FAT10 to estimate level of FAT10 induction in this particular cell line. Furthermore an IP with respective Ig Control is missing. The reverse experiment (IP against endogenous JunB and probing for FAT10 would be supportive as well). Does the asterix indicate an unspecific band? Why is this band also enhanced upon cytokine treatment? Molecular weight marker missing in b for FAT10.

- Figure 1c. There is a double band in the blot (IP against HA (JunB) / probed with anti FLAG (FAT10). Do these bands represent mono and multi-Mono modified JunB? Why is this not seen when another Tag is used (IP against FLAG-Jun in Fig. 1a).

To support the argumentation that FATylation is somehow specific for JunB, the same experiment should be performed with a negative control (e.g Coexpression of FLAG FAT10 with HA tagged GFP, followed by respective IPs). This is particularly important as the same group has published a paper (Spinnenhirn et al. FEBS Lett. 2017) stating that FAT10 targets newly translated proteins (Which is the case in such an overexpression experiment, but would have limited relevance for the in vivo situation).

- Figure 1e. Based on the mutational analysis the authors argue that all sites are involved in FAT 10 modification with K237R having a more pronounced effect. It's a bit strange that even the K3R mutant shows a stronger FAT10 band than the K237R mutant. The authors should attempt to validate the FAT10 modification of particular sites of JunB by Mass Spec upon JunB IP.

- Figure 2A. In vitro sumoylation experiments. Concentrations of enzymes used for the in vitro assays appear strange (e.g only 200pM E1 (typo?) in combination with 12,5uM SUMO2/ 8.3uM SUMO1/ 6,2uM FAT10) E2 conc not specified (Same as in 7c? (1,4uM?). Clarify throughout the whole manuscript. Fat 10 Blot missing.

- Figure 3a. It is mentioned in the text that the RanGAP-SUMO FRET experiments were also performed with the FAT10 diglycine mutant (Data not shown). As the data presented in Figure 3a is very clearcut, also the experiment with the diGly FAT10 mutant should be included in the figure or Suppl. Information.

- Figure 3e. Authors show that in the cell line expressing FAT10 PML bodies are reduced. Given the fact that FAT10 is induced upon IFN γ /TNF treatment, it would strengthen the statement of the paper and show physiological relevance, if the same effect (reduction of endogenous PML levels) is seen in wt cells after TNF/IFN γ treatment.

- Figure 5a. As recombinant proteins are available, it would strongly support the authors argumentation if experimental data for binding affinities of FAT 10 to UBA2/AOS1 are provided (e.g by Biacore measurements or Microscale thermophoresis (Nanotemper)). This might also be a suitable approach to address/validate the crosslink results shown in figure 7 (Higher interaction with AOS1(Fig. 7d)).

- Figure 5b and c. Thioester formation might be better monitored upon a shorter incubation periode (e.g 10min) as after 30 min most likely dominantly conjugates are monitored. This might explain why only minor differences between red and non red conditions are observed. Likewise, band representing SUMO-E1 thioester would be expected to be stronger. It might be more conclusive to show Thioester formation with a titration of FAT10.

- Figure 5e and Figure 7a-c show important results but might be shifted to suppl data to facilitate reading

- Page 6 Referring to the paper by Deque et al (Nat Immunol. 2016) authors imply a competitive action of FAT10/SUMO modification in the inflammatory response. However the paper cited refers to a role of SUMO in type I IFN expression and Type I IFN ISG gene activation, whereas FAT10 is induced by IFN γ . The strong statement that "the pro-inflammatory cytokine induced UBL FAT10 inhibits the conjugation of the anti-inflammatory UBL (SUMO)" for sure is an oversimplification as the action of SUMO is not generally anti-inflammatory but more complex (e.g neg action on Nf κ B signaling, IL-8, PPAR etc.).

- Minor: Materials and methods: Rek proteins and in vitro assays.Fig5b: Concentration of FAT10AV, His-ISG15 and lin Di-Ub-Concentration not provided- please add.

Point-to-point responses to reviewers' comments on "Regulating the regulator: the ubiquitin-like modifier FAT10 thwarts SUMO activation" (NCOMMS-19-11396) by Aichem et al.

Reviewer #1 (Remarks to the Author):

In the article "Regulating the regulator: the ubiquitin-like modifier FAT10 thwarts SUMO activation" the authors describe new functionality for the cytokine-inducible ubiquitin-like modifier protein FAT10 in that it hinders activation of the anti-inflammatory ubiquitin-like modifier SUMO. They also demonstrate that the transcription factor JunB is a substrate of FAT10 and FAT10ylation of JunB occurs preferentially to SUMOylation. The authors utilize a number of affinity tagged (FLAG-tagged and HA-tagged) proteins with immunoprecipitation and site specific mutants to explore the interaction of FAT10 with JunB. They demonstrate that FAT10 modifies the same Lys residues on JunB as SUMO, however FAT10 prevents SUMOylation even in the absence of the FAT10 conjugation enzyme UBA6. They go on to demonstrate that FLAG-FAT10 expression leads to a bulk decrease in SUMOylation in HEK293 cells. They then use IFN- γ /TNF- α treatment of HepG2 and HepG3 cells to induce endogenous FAT10 and CRISPR/Cas9 FAT10 knockout cell lines to show it has the same effect of reducing SUMOylation as the FLAG-tagged version. The authors then investigated the mechanism by which FAT10 interferes with SUMO, and show that it interacts directly with the SUMO E1 activating enzyme UBA2/AOS1. They go on to demonstrate that FAT10 is activated by UBA2/AOS1, forming a thioester conjugate. They also demonstrate that although FAT10 is activated by UBA2/AOS1, this form of activated FAT10 is not subsequently transferred to an E2 conjugating enzyme and therefore does not result in FAT10ylation of proteins. Finally, the authors explore the interaction of FAT10 with UBA2/AOS1 with chemical cross-linking and mass spectrometry (XL-MS). They identify cross-links between lysine residues of FAT10 and both AOS1 and UBA2, although a greater number of links with AOS1. They then perform quantitative cross-linking between UBA2/AOS1 and SUMO1 in the absence or presence of FAT10 and conclude that FAT10 had no discernable impact on the conformation of UBA2/AOS1 although a number of the same Lys residues on AOS1 and UBA2 were identified as linked to both FAT10 and SUMO1 suggesting FAT10 and SUMO1 compete for the same binding interface. Overall the authors provide a wealth of experimental data on FAT10, its interference with SUMO and the FAT10, UBA2/AOS1 interaction. The manuscript is on the whole well written and follows a logical progression of the investigation into this newly reported function for FAT10. The authors conclusions based on the data appear sound for the most part. I do have concerns about the authors interpretation of the XL-MS results and about how the cross-linking experiments were carried out which are described in detail below. While these concerns do not impact the broader story of FAT10 functioning to block SUMO activation, they could impact the conclusions of how this occurs due to the interaction between FAT10 and UBA2/AOS1 and therefore in my opinion, need to be clarified before this manuscript can be published.

The authors perform two sets of XL-MS experiments. One set as a discovery effort to identify cross-linked Lys between FAT10 and the FAT10-AV mutant with UBA2/AOS1, and the other set as quantitative XL-MS experiments to investigate the conformational impacts of FAT10 binding on UBA2/AOS1. I have concerns about the interpretation of data obtained by both sets of XL-MS experiments and how these experiments were carried out. At the heart of this is the confidence in cross-linked peptide identifications. The authors use the xQuest pipeline for assignment of peptides to the MS data and report score thresholds for the assignments they deemed confident. Of concern is that there is no mention as to how these score thresholds were determined and what they mean? It is now commonplace in XL-MS experiments, as in all MS-based proteomics, to report an estimate of false discovery rate (FDR). The authors do not provide an estimate of FDR or adequately describe how identifications were deemed confident. xQuest scores by themselves do not mean much. Why use different score thresholds used between the 1st set of XL-MS experiments and the 2nd quantitative XL-

MS set? Reporting FDR would provide a consistent assessment for a score threshold and there are many software tools out there to provide this if xQuest does not.

Reply 1: The linear discriminant score, as calculated by *xQuest* and which we report, is a weighted sum of four subscores (xcorr_c, xcorr_x, match-odds and TIC) and is used to assess the quality of the composite MS² spectrum. It is explained in detail in the cited literature (Leitner et al, Nature Protoc, 2014 (PMID:24356771); referring to Rinner et al, Nature Methods, 2008 (PMID:18327264)). It is thus an excellent tool to assess the confidence of crosslink identifications and directly correlates with FDR scores, as shown numerous times (see for example Erzberger et al, Cell, 2014 (PMID:28898626)). Different score thresholds between the 1st and 2nd set of XL-MS experiments were used to provide similar and appropriate levels of confidence between experiments (see also below).

We also have already calculated false discovery rates (FDRs) with *xProphet* (please see again cited literature (Leitner et al, Nature Protoc, 2014 (PMID:24356771); Walzthoeni, Nature Methods, 2012 (PMID: 22772729)) and we did report those already in the current manuscript (please see provided uploaded data (xQuest result files) in PRIDE).

We have so far refrained from reporting the calculated FDR values directly in the methods description, as we do believe that 'proper FDR control is not trivial for small search spaces' - in particularly for non-cleavable crosslinkers, as used in this study - a notion that is as shared in the latest and first community-wide XL-MS study (Iacobucci et al., Anal. Chem., 2019 (PMID:31045356)).

Despite this, we have now amended the manuscript and have included the FDR values for every crosslink already in the Supplementary tables. The overall calculated FDR is < 0.05 for all datasets, with the vast majority of crosslinks having an FDR ≤ 0.01.

We have now also added the FDR to the method section for all crosslink experiments:

For the FAT10 or FAT10-AV with SUMO E1 dataset:

Page 23: "Crosslinks which were identified with deltaS < 0.95 and ID-Score > 25 *and an assigned FDR as calculated by xProphet below 0.05...*"

For the quantification crosslink experiment shown in supplementary figure S3B and S3C:

Page 24: "In this study, only links with an ID-Score ≥ 28 and *an assigned FDR as calculated by xProphet below 0.05* that showed a change of log₂ratio ≥ ±1 and a p-value of ≤ 0.01 were considered significant."

In the first set of experiments the authors identify cross-links between FAT10 and UBA2 and AOS1 and conclude that FAT10 inhibits SUMO activation mainly by interacting non-covalently with AOS1. While it does appear that more cross-linked pairs of peptide and cross-linked sites were identified between FAT10 and AOS1, this doesn't necessarily mean that FAT10 is preferentially interacting with AOS1 compared with UBA2. The formation of Lys-Lys cross-links depends on many factors, including relative proximity and orientation, solvent accessibility, and reactivity of the linked Lys. The authors appear to equate more identified non-redundant cross-links with a greater abundance which is not an accurate assessment. It could be that there is just as much interaction between FAT10 and UBA2 as there is between FAT10 and AOS1 but due to the position of Lys residues within the structures not as many unique cross-links can be formed. Furthermore, additional FAT10-UBA2 links could be forming but just not detected by the authors strategy. These important points must be considered.

Reply2: While we fully agree with the reviewer that lysine-lysine crosslink formation depends on many factors, we nevertheless quite strongly believe that crosslinking mass spectrometry is able to tell us something about protein-protein interactions – a notion that is in all likelihood shared by the vast majority of the crosslinking community and may be even considered as its central dogma (see for example recent reviews (Piotrowski & Sinz, Adv Exp Med Biol, 2018 (PMID:30617826); Yu and Huang, Anal Chem, 2018 (PMID: 29160693); O'Reilly & Rappsilber, Nat Struct Mol Bio, 2018 (PMID:30374081); Leitner, Faini, Stengel et al., Trends Biochem Sci. 2016 (PMID:26654279)).

We also do not make any statement on the abundance of identified crosslinks whatsoever in the manuscript at this stage. However, as shown in Figure 7d and 7e (and also Supplementary Fig. 8b and

Supplementary Fig. 8c) all three proteins - FAT10, AOS1 and UBA2 - contain lysine residues that are roughly evenly distributed over the full length of these proteins and, importantly, for basically all of these lysine residues we detect intra-protein crosslinks. Hence, we assume that these lysine residues are in principle accessible for crosslink formation and also detectable by our workflow. While AOS1 forms many unique inter-protein crosslinks with FAT10 and UBA2, UBA2 forms mainly inter-protein crosslinks with AOS1.

This experimental observations we do indeed then interpret in the way that the most likely explanation of this crosslinking pattern is, that FAT10 is interacting with both AOS1 and UBA2, but likely more with AOS1 than with UBA2.

In order to clarify these points even further, we have amended the manuscript on page 13.

“The analysis revealed a large number of high-confidence crosslinks both within the different subunits of SUMO E1 but also between FAT10 and AOS1 and, less pronounced, FAT10 and UBA2, **suggesting that FAT10 mainly interacts with AOS1**”.

And on page 14:

“The same experiment performed with the FAT10 diglycine mutant FAT10-AV resulted in the identification of approximately the same crosslinks as seen for wildtype FAT10, **suggesting** that the covalent interaction of FAT10 with the E1 was not mandatory to exert the inhibitory function.”

In the quantitative XL-MS experiments the authors use isotope labeled BS3 to look for conformational changes in AOS1/UBA2 that are potentially induced by the binding of FAT10. The authors use the data from these experiments to conclude that FAT10 binding has no impact on the conformational state of AOS1/UBA2. As presented it is not clear to me that this is an accurate conclusion to make. The authors seemingly arbitrarily selected links displaying a 2-fold or greater change to be indicative of a significant change? How was this threshold determined? It is very concerning to me that the quantified cross-links in supplementary tables 3 & 4 span very large ranges of ratios (TableS3 log₂ -13.69 to +17.5 and TableS4 log₂ -11.86 to +20.31). Do the authors really believe they are capable of accurately quantifying cross-links varying over 10 orders of magnitude? I find this extremely doubtful and think the quantitative analysis needs to be carefully revisited and thoroughly revised. Of additional concern is that the distributions of log₂ ratios in TableS3 and TableS4 are not normal nor centered at a log₂ of zero as one would expect for a situation where the majority of links do not change. I suspect that the authors just passed the data through xTract and accepted the output without much additional thought or analysis. Regardless it is not clear to me how they arrive at the conclusion that there is not a significant conformational change when the majority of the links quantified appear to have log₂ changes with a magnitude of greater than 1.

Reply 3: We suspect there are some misunderstanding with regard to the quantitative XL-MS experiments and we are happy to clarify these.

The reviewer correctly points out that it would be a very daunting undertaking to quantify crosslinks over 10 orders of magnitude. However, we have neither done so nor claimed that we did. The very large ranges of ratios come from imputed values – e.g. when a crosslink was identified in only one set of experiments and not in the other. This is a common practice in proteomics and also discussed at length in the cited literature (Leitner et al, Nature Protoc, 2014 (PMID:24356771); Walzthoeni et al., Nature Methods, 2015 (PMID:26501516); Sailer et al., Nat Commun, 2018 (PMID: 30361475)).

All imputed values were also already clearly indicated in the Supplementary Tables (column name: “imputed values”). For a detailed description please also refer to the Supplementary Information.

We also had plotted all measured intensities (**Figure 1** of this point to point reply), confirming that they are normally distributed since kde and normal distribution overlay almost perfectly.

Figure 1. Distribution of measured intensities from qXL-MS experiments

The histogram of the log₂ transformed intensities (only non-imputed values) are shown for the qXL-MS experiments comparing crosslink abundances of SUMO E1 in presence or absence of SUMO-1 (**left panel**) and in presence or absence of SUMO-1 and FAT10 (**right panel**). The blue line shows the kernel density estimator (kde) of the log₂ transformed measured intensities whereas the black line shows a normal distribution fitted to our data.

The biggest misunderstanding, however, seems to stem from the reviewer's understanding of our actual experimental set-up. We do neither claim that there is no conformational change if SUMO is present nor do we state this. It is published that the conformation of the SUMO E1 changes upon binding of SUMO (Olsen et al., Nature, 2010 (DOI 10.1038/nature08765)). Thus, one would also expect that a large proportion of crosslinks actually *does* change upon addition and binding of SUMO and this is also what we see, as upon addition of SUMO many links in the SUMO E1 are significantly up- or down-regulated (Supplementary Figure 8b). We only state in the manuscript that the overall **crosslinking pattern** (of significantly up- or down-regulated links) is not altered if FAT10 is present alongside SUMO in the sample.

And as an important side-note: we are also very cautious to causally link a change in a crosslinking pattern or even a change in a single crosslink to a conformational change. A change in crosslink abundance can be caused by many things – among them a conformational change of the protein or protein complex under investigation.

This is why we phrase very carefully on page 14:

“the addition of FAT10 to SUMO E1 and SUMO-1 resulted in *no apparent differences in the crosslinking pattern* within and between AOS1/UBA2

and SUMO-1 (Supplementary Fig. 8c), *indicating* that FAT10 has no discernible impact on the conformational state of the activated SUMO E1”.

Finally, we have further chosen a 2-fold change as significance criterion, as a 2-fold change is a widely used and accepted value in quantitative proteomics and biology.

However, even taking an even more conservative 3-fold or greater change as significance criterion, the overall picture stays the same and similar links are regulated (**Figure 2** of this point to point reply).

Figure 2. Quantification of crosslinks in presence or absence of SUMO-1 and FAT10/SUMO-1

Q-XL-MS was used to compare crosslink abundances of SUMO E1 in presence or absence of SUMO-1 (**left panel**) and in presence or absence of SUMO-1 and FAT10 (**right panel**). Only crosslinks that could be reproducibly quantified from the pool of identified high-confidence crosslinks in both samples ($n = 3$) are shown (violation = 0, p -value ≤ 0.01 , ID-Score ≥ 28 , see methods and Supplementary Figure 8b, 8c for details). Depicted in red are

crosslinks that were significantly downregulated (3-fold change or higher) in samples with SUMO-1 or SUMO-1 and FAT10, while green links indicate significant upregulation (3-fold change or higher).

More importantly, also in this case the overall **crosslinking pattern** (of significantly up- or down-regulated links) is not altered if FAT10 is present alongside SUMO in the sample, thus corroborating our original significance criterion.

Beyond just reporting the average log₂ ratios and associated p-values the authors should provide some measure of variance, such as a confidence interval or standard deviation for each log₂ value.

Reply 4: The log₂ ratio already contains two values which both have a standard deviation (the intensities in the different experiments). These standard deviations would have to be taken into account when calculating a standard deviation for the log₂ ratios. It is not common in the field of mass spectrometry to provide a confidence interval or a standard deviation for log₂ ratios. Usually p-values or corrected p-values (e.g. Benjamini-Hochberg FDR, Permutation based FDR or the Bonferroni correction as a family wise error rate) are determined for log₂ ratios.

The p-value sufficiently describes the quality of a log₂ ratio as it contains both the standard deviation of the two experiments as well as their difference.

Even granting the authors the benefit of the doubt, by comparing the quantified links shown in supplementary Fig 3 b&c there still appears to be differences that could indicate a significant conformational change. For example, in FigS3c there appears to be a link somewhere between K250 and K300 of UBA2 that is red (significantly decreased) that was not present in FigS3b. Why is this not indicative of a conformational change induced by FAT10?

Reply 5: Of note, Supplementary Fig. 3 has changed to new Supplementary Fig. 8. First, we want to state that the overwhelming majority of crosslinks that were reliably quantified shows an **identical** crosslinking pattern between the two datasets – e.g. basically *all crosslinks* in both datasets are either (significantly) up – or downregulated or show no change in their crosslinking abundance. Such there is clear evidence, that the overall crosslinking pattern does indeed NOT change, if FAT10 is present alongside SUMO in the sample.

The reviewer has (by undisclosed criteria) selected one single link that he found to be differentially regulated in its abundance between these datasets. This, in our opinion, is already problematic, as it assumes an absolute error-free world, which is far from both the biological and also technical reality.

We assume that the uxID which the reviewer refers to is UBA2:260:x:UBA2:316. This uxID shows a log₂ ratio of -1.24 in the set of experiments where FAT10 is present and is thus shown in Supplementary Fig.8c in red (significantly down regulated). In the set of experiments in which FAT10 is **not** present, this uxID was not reliably quantified (p-value of 0.07; all links are part of the tables uploaded to PRIDE). This is why it is shown in Supplementary Fig. 3b as a dashed line. However, the log₂ ratio of this uxID is -0.59, so the tendency is the same as in the experiment with FAT10. Still we would not draw any conclusions from this link as it was not reliably quantified.

As a final concern regarding XL-MS, in both cross-linking reactions why was Tris-HCl used as a buffer? This is very unusual as Tris will react with the NHS-esters in BS3 using up the reagent before it can react with Lys in the proteins, and also form Tris-protein cross-links potentially altering the proteins structure. This must be addressed and the authors need to justify why Tris was used here and that it has no negative impact on the results, such as inducing conformational changes in the proteins. Being that the Tris concentration is so high I would expect it to have a significant impact on the identified and quantified cross-links. It could very well preclude formation of Lys-Lys links that could be important in the interaction of FAT10 with UBA2/AOS1.

Reply 6: Our reviewer raises an important point here. We usually indeed recommend not to use Tris containing buffers and we are aware of the fact that some groups even suggest to quench the crosslinking reactions for NHS-esters with Tris.

However, in this particular case the underlying biology necessitated the use of Tris in our buffer system, as all the biochemical assays were performed in this Tris containing buffer and we wanted to both replicate these experimental conditions as close as possible for our crosslinking experiments and also avoid any potential negative influences of a buffer exchange on complex stability.

We therefore carefully had examined if our crosslink reaction was still working under these conditions; the fact that we identify both a comparable number of identified crosslinks and at a similar ratio in terms of distribution of inter- to intra- and monolinks to protein complexes of comparable size and function that we have investigated in our lab over the last years, was a strong support of our assumption that the crosslinking reaction was still working fine. Even more, the fact that we obtain a highly reproducible crosslink pattern over multiple experiments (overall, we have performed 8 different crosslinking experiments) using biological replicates together with the fact that we do detect intralinks to basically all lysines, also clearly demonstrate that the presence of Tris in our buffer did not preclude the formation of lysine-lysine crosslinks.

These findings are also in line with reports from the literature that claim that Tris, while containing a primary amine, is *sterically* too confined to react with NHS-esters and also with reports that show that a Tris containing buffer system can still be used to quench NHS-esters (Wang et al., Proteomics, 2010 (PMID:20127692)). This is also the reason why we in our suggested workflow always advise to rather quench with ammonium bicarbonate than with Tris.

Despite this, we have undertaken additional experiments, where we have exchanged FAT10 and AOS1/UBA2 into 20 mM HEPES pH 7.5 before crosslinking. **Figure 3** of this point to point reply shows that the crosslinking pattern does not undergo any major changes in the presence of Tris, thus clearly demonstrating that the presence of Tris in the buffer did not significantly alter the outcome of the crosslinking reaction

Figure 3. Comparison of Sumo E1 and Fat10 crosslinked in Tris-buffer or HEPES buffer

100 μ g of SUMO E1 and FAT10 (molar ratio 1:6) were crosslinked with BS3 in 45 mM Tris-HCl, pH 7.5, 100 mM NaCl, 20 mM MgCl₂, 1 mM TCEP (**left panel**, see also Figure 7D in manuscript) or in 20 mM HEPES, pH 7.5 (**right panel**).

Specific comments

For clarity the authors should employ consistent use of SUMO E1 and/or UBA2/AOS1 throughout the manuscript. Once UBA2/AOS1 is defined as the SUMO E1 activating enzyme it should be used exclusively to avoid confusion.

Reply 7: We agree with our reviewer and have changed the manuscript accordingly.

Page 9 - “A similar result was obtained when the same experiments were performed in additional cancer cells lines such as HCT116 or MCF-7 (data not shown).” This data should be shown, at least as a supplementary figure.

Reply 8: We have now included these data as new Supplementary Fig. 4.

Page 10 – “A co-immunoprecipitation of recombinant FAT10 and SUMO E1 revealed that FAT10 directly interacted with UBA2/AOS1 in absence (Fig. 5a, lane 5) or presence of ATP (data not shown).” The data with ATP should be shown, at least as a supplementary figure. Does the presence of ATP impact the amount of the FAT10-UBA2/AOS1 interaction?

Reply 9: We did not show these data in the first submission, because it looked exactly the same way. Nevertheless, we have now included these data as new Supplementary Figure 5.

Methods Section Page 23 - Chemical crosslinking coupled to mass spectrometry (XL-MS) Authors state: “Proteins were crosslinked and measured essentially as described in 52.” While it is fine to refer to a published protocol that is used for the cross-linking method, key details pertaining to this experiment need to be fully described. Currently they are not. Missing details that require clarification are: Its stated that SUMO E1 - FAT10(-AV) are mixed at ~ 1:6 molar ratio, but no mention of the concentration of FAT10 and FAT10-AV in the stock solution? What is the final concentration of all protein components and buffer composition and volume during the cross-linking reaction?

Reply 10: To clarify these points we have amended our methods section on page 25:

“FAT10 or FAT10-AV (1.8 µg/µl or 0.54 µg/µl stored in 50 mM Tris-HCl, pH 7.5, 150 mM NaCl, 5% Glycerol, 1 mM TCEP, as described in ⁴⁹) and 100 µg of AOS1/UBA2 (1.8 µg/µl stored in 40 mM Tris-HCl, pH 7.5, 100 mM NaCl, 20 mM MgCl₂, 1 mM TCEP, as described in ⁴⁷), were incubated for 10 min on ice.”

For the description of the quantitative crosslinking we also amended our methods section on page 25:

“In short, approximately 100 µg of AOS1/UBA2 were incubated either on its own, together with SUMO-1 (molar ratio 1:4) or with SUMO-1 and FAT10 (molar ratio 1:4:4) for 15 min at 30 °C in 150 µl total volume of 1 x reaction buffer (20 mM Tris-HCl pH 7.6, 50 mM NaCl, 10 mM MgCl₂, 4 mM ATP, 0.1 mM DTT) prior to crosslinking by addition of H12/D12 BS3 (Creative Molecules) at a ratio of 1 nmol / 1 µg protein for 30 min at 37 °C while shaking at 650 rpm in a Thermomixer (Eppendorf).”

It is stated that both proteins are in Tris-HCl buffers (50 mM Tris for FAT-10 and 40 mM Tris for SUMO E1). Tris-HCl is an unusual choice of buffer for these experiments as it is generally incompatible with NHS-ester cross-linking due to the fact it contains primary amines that will react with the cross-linker. This is clearly stated in the protocol referred to in Ref 52 “Any additives containing amino groups must be avoided.”. Was a buffer exchange performed prior to cross-linking to remove the Tris? Otherwise the Tris would undoubtedly react with the BS3, consuming a large amount of the cross-linking reagent

as well as potentially cross-link to the proteins themselves. Was this accounted for? Having a relatively high concentration of Tris present during the cross-linking reaction could impact the structures and interactions of the proteins by forming Tris-Lys cross-links. If these formed they should be detectable by MS analysis.

Reply 11: These issues have been taken care of as outlined in Reply 6 above.

“were incubated for 10 min on ice. Proteins were crosslinked by addition of H12/D12 BS3 (Creative Molecules) at a final ratio of 1 nmol BS3/1 µg protein for 30 min at 37 °C while shaking at 650 rpm in a Thermomixer (Eppendorf)”. Why incubate on ice but then cross-link at 37C? What effect does temperature have on the structures and interactions? How was 1 nmol BS3/ 1 µg protein chosen? Was the ratio of cross-linker to protein optimized somehow?

Reply 12: We incubated on ice in order to allow thorough mixing of SUMO E1 and FAT10 or FAT10-AV prior to crosslinking. The crosslinker to protein ratio has been optimized by us and other groups over years (Leitner et al, Nature Protoc, 2014 (PMID:24356771); Walzthoeni, Nature Methods, 2012 (PMID: 22772729)). Also, for the quantification experiments either the final protein concentration or the total protein amount has to differ because additional proteins (SUMO or SUMO and FAT10) are added in one condition. We wanted to keep the ratio of crosslinker-to-lysine-residues at a similar level, so we added 1 nmol BS3 per µg protein.

Furthermore, the complex is stable under the used conditions and as we wanted to ensure that we are looking at end-points in all crosslinking reactions, we chose 37 °C for the crosslinking reactions.

“After quenching by addition of ammonium bicarbonate to a final concentration of 50 mM and incubation for 10 min at 37°C, samples were dried, reduced, alkylated and digested with trypsin.” Is quenching necessary here? What amount of unreacted BS3 is expected after 30 min? Particularly if there was already 40-50 mM Tris-HCl in the cross-linking reaction buffer.

Reply 13: Yes, see also Reply 6 above. There are reports in literature that indicate that Tris is *sterically* too confined to react with NHS-esters and will therefore not suffice to adequately quench NHS-ester containing crosslinkers (Wang et al., Proteomics, 2010 (PMID:20127692)). This is why we did quench with 50 mM ammonium bicarbonate.

What compounds and concentrations of reducing/alkylating reagents and trypsin were used?

Reply 14: To clarify this point we altered the following sentence on page 25:

“After quenching by addition of ammonium bicarbonate to a final concentration of 50 mM and incubation for 10 min at 37 °C, samples were dried, dissolved in 8M urea to a final concentration of 1 mg/ml, reduced with TCEP at a final concentration of 2.5 mM, alkylated with iodacetamid at a final concentration of 5 mM and digested over night with trypsin (Promega V5113) in 1 M Urea (diluted with 50 mM ammonium bicarbonate) at an enzyme-to-substrate ratio of 1:40.”

We also amended the description in the methods section for the quantitative crosslinking experiments on page 24

“After quenching by addition of ammonium bicarbonate to a final concentration of 50 mM and incubation for 10 min at 37 °C, samples were reduced, alkylated, and digested with trypsin (details see above).”

“Digested peptides were separated from the solution and retained by a solid phase extraction system (SepPak, Waters) and then separated by size exclusion chromatography prior to liquid chromatography (LC)-MS/MS analysis on an Orbitrap Fusion Tribrid mass spectrometer (Thermo Scientific).” Details on

SEC and LC-MS analysis are missing. What conditions for the SEC? How many fractions were collected? LC-MS conditions? LC type, column, gradient, mass spec methods?

Reply 15: To clarify this point we added the following subsections to the methods section on page 27:

“Enrichment of crosslinked peptides by size exclusion chromatography (SEC)

Crosslinked peptides were enriched by size exclusion chromatography on an ÄKTAmicro chromatography system (GE Healthcare) using a Superdex™ Peptide 3.2/30 column (GE Healthcare) at a flow rate of 50 µl/min of the mobile phase (water/acetonitrile/trifluoroacetic acid 70/30/0.1, vol/vol/vol). UV absorption at a wavelength of 215 nm was used for monitoring the separation. The eluent was collected in fractions of 100 µl in a 96-well plate. The two fractions 1.2 - 1.3 ml and 1.3 – 1.4 ml were collected, dried and further analyzed by LC-MS/MS. For the FAT10-AV mutant also the fraction 1.1 -1.2 ml was analyzed by LC-MS/MS.

“LC-MS/MS analysis

Samples fractionated by SEC were re-dissolved in an appropriate volume of MS buffer (acetonitrile/formic acid 5/0.1, vol/vol) according to their UV signal. Peptides were separated on an EASY-nLC 1200 (Thermo Scientific) system equipped with a C18 column (Acclaim PepMap 100 RSLC, length 15 cm, inner diameter 50 µm, particle size 2 µm, pore size 100 Å, Thermo Scientific). Peptides were eluted at a flow rate of 300 nl/min using a 60 min gradient starting at 94% solvent A (water/acetonitrile/formic acid 100/0/0.1, vol/vol/vol) and 6 % solvent B (water/acetonitrile/formic acid 20/80/0.1, vol/vol/vol) for 4 min, then increasing the percentage of solvent B to 44 % within 45 min followed by a 1 min step to 100 % B for additional 10 min. The mass spectrometer was operated in data-dependent-mode with dynamic exclusion set to 60 s and a total cycle time of 3 s. Full scan MS spectra were acquired in the Orbitrap (120.000 resolution, 2e5 AGC target, 50 ms maximum injection time). Most intense precursor ions with charge states 3-8 and intensities greater than 5e3 were selected for fragmentation using CID with 35 % collision energy. Monoisotopic peak determination was set to peptide and MS/MS spectra were acquired in the linear ion trap (rapid scan rate, 1e4 AGC target).”

“Data were searched using xQuest in ion-tag mode with a precursor mass tolerance of 10 ppm. For matching of fragment ions, tolerances of 0.2 Da for common ions and 0.3 Da for crosslink ions were applied. Crosslinks which were identified with $\Delta S < 0.95$ and ID-Score > 25 were visualized by xiNET software” What protein database was searched? Some explanation as to what these scores mean and how the thresholds were determined is necessary.

Reply 16: The protein databases have already been uploaded to PRIDE as stated in the data availability statement in the original manuscript:“The MS raw files, databases containing protein fasta sequences for analysis with xQuest as well as xQuest result- / xTract input- files (xtract.csv) and the xTract result files (analyzer.quant.xls) have been deposited to the ProteomeXchange Consortium via the PRIDE ⁵⁶ partner repository with the dataset identifier PXD012592.”

Please also see Reply 1 of this rebuttal for an explanation to the xQuest scores.

In XL-MS experiments, as in general mass spec based proteomics, it is now routine to estimate false discovery and report a false discovery rate (FDR). I would expect the authors to provide some estimate of FDR for the reported cross-links. FDR for cross-linking can also be reported at multiple levels (PSM, non-redundant peptide pair, Lys-Lys pair, protein pair, see Anal Chem. 2017 Apr 4; 89(7): 3829–3833.). Some analysis/discussion related to FDR needs to be included.

Reply 17: These points have already been addressed in Reply 1 above.

“In short, approximately 100 µg of SUMO E1 were incubated either on its own, together with SUMO-1 (molar ratio 1:4) or with SUMO-1 and FAT10 (molar ratio 1:4:4) for 15 min at 30 °C in 1 x reaction buffer (20 mM Tris-HCl pH 7.6, 50 mM NaCl, 10 mM MgCl₂, 4 mM ATP, 0.1 mM DTT) prior to crosslinking by addition of H12/D12 BS3 (Creative Molecules) at a ratio of 1 nmol / 1 µg protein for 30 min at 37 °C while shaking at 650 rpm in a Thermomixer (Eppendorf).” Why are different molar ratios between SUMO E1 and FAT10 used here compared with cross-linking section above? Also why is the incubation here 15 min at 30C vs. 10 min on ice previously? Again please provide concentrations for all proteins and stock/ final buffer compositions. Tris-HCl is used here again but apparently at a different concentration.

Reply 18: For the quantification experiments we wanted to start with the minimal excess of SUMO or FAT10 for which we still see the effect of FAT10 inhibiting SUMO activation. We tested for this and at a molar ratio of SUMO E1 : SUMO-1 : FAT10 of 1:4:4 this effect is still visible (see Supplementary Fig. 8a). The proteins were incubated for 15 min at 30 °C to allow SUMO activation by the SUMO E1 to take place prior to crosslinking.

In the section above we only wanted to see if and how FAT10 and SUMO E1 interact, so we incubated for 10 min on ice to allow a thorough mixing of the two proteins before the samples were crosslinked.

For quantitative cross-linking separate reactions with H12-BS3 and D12-BS3 must have been performed but this is not clear as described. How and when did mixing of the H12 and D12 samples occur?

Reply 19: We use isotope labelled crosslink reagent (H12/D12 BS3) to provide an additional layer of evidence for identification, not quantification, as the xQuest algorithm actually makes use of this mass shift already during the identification of the crosslinks. This is explained in detail in the cited literature (Leitner et al, Nature Protoc, 2014 (PMID:24356771)).

“Amounts of potential crosslinks were normalized prior to MS by measuring peptide bond absorption at 215 nm for each fraction.” More details are needed here. What fractions the SEC? How many are there and what is the reproducibility of SEC fractionation between different injections? Why not mix/ normalize samples prior to digestion to remove variability introduced in downstream processing steps? How does the absorption at 215nm normalize for cross-links when much more abundant non cross-linked peptides must also contribute to this absorbance?

Reply 20: We again want to point the reviewer to the cited literature where on page 125 the relevant procedure is explained in quite some detail (Leitner et al, Nature Protoc, 2014 (PMID:24356771)). SEC is used to deconvolute and to enrich for crosslinked peptides, not to fully separate them from their non-crosslinked counterparts. This also prohibits mixing, at least without any additional labelling steps (Leitner et al, Mol Cell Proteomics. 2012 (PMID: 22286754)).

However, to clarify this point even further we added the following subsections to the methods section on page 27:

Enrichment of crosslinked peptides by size exclusion chromatography (SEC)

Crosslinked peptides were enriched by size exclusion chromatography on an ÄKTAmicro chromatography system (GE Healthcare) using a Superdex™ Peptide 3.2/30 column (GE Healthcare) using as mobile phase (water/acetonitrile/trifluoroacetic acid 70/30/0.1, vol/vol,vol) at a flow rate of 50 µl/min. UV absorption at a wavelength of 215 nm was used for monitoring the separation. The eluent was collected in fractions of 100 µl in a 96-well plate. The two fractions 1.2 -1.3 ml and 1.3 – 1.4 ml were collected, dried and further analyzed by LC-MS/MS. For the FAT10-AV mutant also the fraction 1.1 -1.2 ml was analyzed by LC-MS/MS.

LC-MS/MS analysis

Samples fractionated by SEC were re-dissolved in an appropriate volume of MS buffer (acetonitrile/formic acid 5/0.1, vol/vol) according to their UV signal. Peptides were separated on an EASY-nLC 1200 (Thermo Scientific) system equipped with a C18 column (Acclaim PepMap 100 RSLC, length 15 cm, inner diameter 50 μ m, particle size 2 μ m, pore size 100 Å, Thermo Scientific). Peptides were eluted on a flow rate of 300 nl/min using a 60 min gradient starting at 94% solvent A (water/acetonitrile/formic acid 100/0/0.1, vol/vol/vol) and 6 % solvent B (water/acetonitrile/formic acid 20/80/0.1, vol/vol/vol) for 4 min, then increasing the percentage of solvent B to 44 % within 45 min followed by a 1 min step to 100 % B for additional 10 min. The mass spectrometer was operated in data-dependent-mode with dynamic exclusion set to 60 s and a total cycle time of 3 s. Full scan MS spectra were acquired in the Orbitrap (120,000 resolution, 2e5 AGC target, 50 ms maximum injection time). Most intense precursor ions with charge states 3-8 and intensities greater than 5e3 were selected for fragmentation using CID with 35 % collision energy. Monoisotopic peak determination was set to peptide and MS/MS spectra were acquired in the linear ion trap (rapid scan rate, 1e4 AGC target).

“In this study, only links with an ID-Score ≥ 28 that showed a change of $\log_2\text{ratio} \geq \pm 1$ and a p-value of ≤ 0.01 were considered significant changes in abundances and are shown in green and red in the 2D visualizations (Supplementary Fig. 3b and 3c, Supplementary Table 3 and 4), respectively.” Why is a different ID-Score threshold used here? What about FDR? I’m confused about how significant changes were determined. A change in $\log_2\text{ratio}$ or $\log_2\text{ratio} \geq \pm 1$? How was 1 selected as a threshold? The \log_2 ratios provided in Tables S3 and S4 span a huge dynamic range (~ 10 orders of magnitude). I’m skeptical cross-linked peptides or any peptides for that matter can be quantified by MS approaches accurately over such a range. Figure 7 d&e and Figure S3b&c –

Reply 21: All these points have already been addressed above (Reply 1).

These xiNET figures, while easy to generate, may not be the most effective way to display the data. It is difficult for readers to really get a sense of what the cross-linking data is. It is nearly impossible to distinguish or pick out any specific Lys residue pair and it is easy to miss or discount a link that could be important. Why can the cross-links not be used to generate structural models for these proteins that could then be docked? This type of information could prove much more valuable. I also do not like that in the quantitative data it has been reduced down to increasing, decreasing or no change. This masks the fact that the quantified values span a huge range. At the very least they should be color coded on a scale with accompanying histograms of the \log_2 values so readers can assess what is being displayed.

Reply 22: While we agree with the reviewer that it is difficult to find an optimal graphical representation of crosslinking data, we believe that these 2-D plots (as plotted for example by xiNET), are actually a pretty good representation of the actual experimental data. As these plots are used by many groups, we are also not alone in our opinion here. Finally, as the complete list of crosslinks is also part of the provided data, it will hopefully be hard to miss any important link.

As we have introduced Bayesian crosslinking guided integrative structural modelling to the crosslinking field (Erzberger et al, Cell, 2014 (PMID:28898626)), we are aware of the fact that crosslinks can be used to generate structural models. However, one needs to be very cautious in doing so, in order to avoid any over-interpretation of the data. In this particular case, we had started to generate structural models, but have eventually decided against such an approach, as the visual model would not add any real biological insights and could draw the attention away from the actual experimental data.

Regarding the representation of the quantitative data, we do disagree with the reviewer. We have introduced and used this representation before, up to now without any complaints (for example: Sailer et al, Nat Commun, 2018 (PMID:30361475); Walker-Gray et al., Proc Natl Acad Sci U S A. 2017 (PMID: 28893983)). More importantly, we also think that it is scientifically difficult to discriminate within a data range that is statistically non-significant. Finally, as the complete list of crosslinks and also their relative

abundance-changes are also part of the provided data, every reader can already now precisely assess what is being displayed.

Reviewer #2 (Remarks to the Author):

1) It is suggested in the introduction that SUMO-2/3 “are mostly present as a free SUMO pool in the cytoplasm and utilized under a variety of stresses.” While it is true that SUMO-2/3 modification is enhanced under stress, it is an oversimplification to say that it exists largely as a free pool in the cytoplasm under normal conditions.

Reply 23: We agree with our referee and have changed the text in the following way (page 3):

“Reversible modification with SUMO is a highly dynamic process. Both, in the absence or presence of cellular stress, SUMO-1 is mostly found conjugated to substrates. The almost identical isoforms SUMO-2 and -3, named hereafter SUMO-2/3, are found mostly unconjugated in normal growth conditions and shift to predominantly conjugated form under a variety of cellular stresses.”

2) It is interesting that FAT10 is largely conjugated to consensus site lysines modified also by SUMO. It would be interesting to know the authors’ thoughts on the nature of this overlap in specificity.

Reply 24: We agree that this is a very interesting finding, however, we have identified conjugation of FAT10 to a lysine within a SUMOylation consensus site only in case of JunB. Unfortunately, we were not able to identify a general FAT10ylation consensus until now and so far only very little lysines within substrates have been identified which become FAT10ylated. We have compared known amino acid sequences containing FAT10ylated lysines as e.g. around Lys-323 in its cognate E2 conjugating enzyme USE1 (Aichele et al., FEBS Journal 2014) with the SUMOylation consensus motif. However, the sequence around Lys-323 does not represent a SUMOylation consensus site. Moreover, in case of USE1, a mutation of Lys-323 did not abrogate FAT10ylation but led to a FAT10ylation of one of the other lysines within USE1 (Aichele et al., FEBS Journal 2014). Thus, it might be possible, that the choice of the lysine to be modified by FAT10 might not strictly be dependent on a specific consensus site. In any case, this is a very interesting question which hopefully will be clarified when more FAT10-modified lysines will be identified.

3) It appears that no FAT10 E2 enzyme (USE1) was used in the assays shown in Figure 2a. Is it known that UBA6 is sufficient to transfer activated FAT10 onto JunB? If so, the authors should comment on this. If not, the authors should also comment on the finding that no E2 is required.

Reply 25: It is indeed the case that for some of the FAT10 substrates we have tested so far in our *in vitro* FAT10ylation assays, no further need for the addition of the E2 conjugating enzyme USE1 to the reaction was necessary. In addition, no increase in FAT10ylated substrate was observed in presence of UBA6 AND USE1. We have published this finding already in two previous papers, namely Bialas et al., PLOS One 2015 and Bialas et al., JBC 2019 and have added the following sentence to the main manuscript (page 6):

“In accordance with what we have published recently for two other FAT10 substrates, namely UBE1 and OTUB1^{30, 36}, only the FAT10 E1 activating enzyme UBA6 was necessary to achieve *in vitro* FAT10ylation of JunB.”

4) It is challenging to interpret the immunofluorescence data in Figure 3e without DAPI staining. Are PML foci in the FAT10-expressing cells in the cytoplasm? Is SUMO-1 in the nucleus or cytoplasm? It would be helpful to include DAPI.

Reply 26: We agree that DAPI staining would be helpful for interpretation of the confocal data. Unfortunately, our microscope has no UV laser to visualize DAPI staining. What we did instead of DAPI staining was to perform another nuclear staining with a dye called “Nuclear Green DCS1” (Abcam, ab138905). As can be seen in the new Supplementary Figure 3b and 3c, a co-staining of PML or SUMO-1 with Nuclear Green DCS1 clearly showed that both proteins are located in the nucleus.

5) The loss of co-localization of PML and SUMO-1 in cells expressing FAT10 is only suggestive of suppression of PML modification. To conclude that FAT10 leads to downregulation “of a single endogenous SUMO substrate such as PML”, IP and western blot data needs to be included.

Reply 27: We are thankful for this recommendation and have now performed Western Blot analyses of HEK293 wildtype and HEK293 FAT10 knockout cells, in presence or absence of IFN- γ /TNF- α treatment to induce endogenous FAT10 expression. As can be seen in the new Figure 3g, an increase in SUMOylated PML was detectable in As₂O₃ and cytokine treated FAT10 knockout cells as compared to HEK293 wildtype cells, expressing FAT10 upon cytokine treatment. This Western blot clearly confirms that in the presence of endogenous FAT10 expression, SUMOylation of PML is diminished. Moreover, inspired by this comment, we have moved old Figures 3e and f, showing the co-localization of PML and SUMO-1 as well as the calculation of the Pearson’s coefficient in stable FLAG-FAT10 expressing cells, to the Supplementary data (new Supplementary Figure 3a). Instead of this, we have included now as new Figure 3e and f the calculation of the numbers of PML bodies per cell, since PML-SUMOylation is required for the formation of PML bodies, whereas the co-localization of the two proteins does not necessarily mean that PML is covalently modified with SUMO-1, as our referee has mentioned.

Reviewer #3 (Remarks to the Author):

Specific points to address

- Figure 1. lane 8/9 authors claim that the FAT10 modification mediates proteasome degradation of modified JunB. However, not only the modified but also the intensity of the unmodified band increase upon MG132 treatment. Likewise, the stability of JunB is increased when the FAT10 non conjugatable form is transfected in the presence of MG132. Please clarify this discrepancy which is in contrast to the conclusion that FAT10 modification mediates proteasomal degradation.

Reply 28: We show in Figure 1 a co-immunoprecipitation experiment with overexpressed FLAG-tagged JunB and wildtype HA-tagged FAT10 or its conjugation-deficient mutant HA-FAT10-AV. Next to the covalent modification of JunB-FLAG with a single HA-FAT10 moiety, we also observe a non-covalent interaction of JunB with FAT10. Treatment of the cells with proteasome inhibitor MG132 causes an

accumulation not only of FAT10, but also of FAT10-AV, since also monomeric, unconjugated FAT10 and FAT10-AV is degraded by the proteasome. Thus, an accumulation of FAT10 or FAT10-AV upon proteasomal inhibition leads to a higher amount of non-covalently interacting JunB-FLAG as seen in the upper panel (IB:HA, IB: FLAG lanes 9 and 11), beyond the unspecific binding of JunB-FLAG to the HA-agarose (seen in in lane 7). To make this more clear, we have now included the following sentence in the main text (page 5):

“Next to this covalent modification, a non-covalent interaction of JunB-FLAG and HA-FAT10 or HA-FAT10-AV was observed, which was more pronounced under proteasome inhibition with MG132, since here, both, wildtype HA-FAT10 as well as HA-FAT10-AV accumulated.”

- In Figure 1b. authors show that endogenous JunB is modified by FAT10. This experiment should also be performed in the presence of MG132 to show that Fatylation influences stability of JunB. Beside JunB and beta actin, lysates should also be probed with anti FAT10 to estimate level of FAT10 induction in this particular cell line. Furthermore an IP with respective Ig Control is missing. The reverse experiment (IP against endogenous JunB and probing for FAT10 would be supportive as well). Does the asterisk indicate an unspecific band? Why is this band also enhanced upon cytokine treatment? Molecular weight marker missing in b for FAT10.

Reply 29: In this experiment we did not use a cell line, as mentioned by our reviewer, but primary peripheral blood lymphocytes (PBLs) from volunteer blood donors. For each of the experiments, PBLs from a 400 ml blood donation were isolated and treated or not with IFN- γ /TNF- α . Thus, since a large number of cells were necessary to perform these experiments, we had decided to exclude MG132 treatment since this was already clearly shown in Figure 1a. The immunoprecipitation (IP) was also performed in the other direction (IP: JunB, IB: FAT10) but did unfortunately give only unsatisfying results. Since we are able to detect the endogenous JunB-FAT10 very clearly in Fig. 1b besides showing the conjugate formed out of the tagged JunB and FAT10 versions in Fig. 1a, c and e, we think that this is convincing enough.

Moreover, we have now repeated the experiment three times including an unspecific isotype control for IP as shown in new Figure 1b.

Since in most experiments, we detected the JunB-FAT10 conjugate already in non-cytokine stimulated cells, we have now additionally included real-time PCR data showing the relative FAT10 expression in PBLs with or without IFN- γ /TNF- α treatment. As can clearly be seen in new Figure 1b (lysate IB:FAT10 and bar graph showing real-time PCR data), cells were already pre-stimulated and expressed decent amounts of endogenous FAT10 already before additional cytokine treatment (Ct levels of 22.4 for untreated PBLs and 21.7 for cytokine treated cells). We have added this observation now also to the main text (page 5):

“Also under endogenous conditions, a stable JunB-FAT10 conjugate was detectable upon immunoprecipitation using a FAT10-reactive antibody but not with an unspecific isotype control antibody. The conjugate was even already visible without cytokine treatment since the cells expressed decent amounts of FAT10 mRNA as well as FAT10 protein already before cytokine treatment (Fig. 1b, lysate IB: FAT10 and real-time PCR data in the bar graph on the right).”

We apologize that we have forgotten to explain the asterisk that marks an unspecific background band in the Figure legend. We have added the following sentence to the legend of Figure 1b: “The asterisk marks an unspecific background band. Mouse IgG1 was used as isotype control for the IP. Bar graph represents relative expression levels of FAT10 mRNA, normalized to the housekeeping gene GAPDH as measured by real-time PCR. Ct-levels of untreated cells were 22.4, and 21.7 for IFN- γ /TNF- α treated cells.”

• Figure 1c. There is a double band in the blot (IP against HA (JunB) / probed with anti FLAG (FAT10). Do these bands represent mono and multi-Mono modified JunB? Why is this not seen when another Tag is used (IP against FLAG-Jun in Fig. 1a).

Reply 30: The double band represents JunB, modified with one single FAT10 moiety. FAT10 has a molecular weight of 18 kDa. In case of a multi-mono-FAT10ylation, the second, upper band should rather be detected at a molecular weight of approximately 75-80 kDa. Since JunB also heavily gets phosphorylated, we suggest, that we detect here mono-FAT10ylated, and putatively phosphorylated (higher band) or non-phosphorylated JunB (lower band). When we are using the other tag-combination, (HA-FAT10 and JunB-FLAG) as seen in Fig. 1a, we do not see this double band as clearly as in Figs. 1c and 1e. Unfortunately, we do not know why this is the case.

To support the argumentation that FATylation is somehow specific for JunB, the same experiment should be performed with a negative control (e.g Coexpression of FLAG FAT10 with HA tagged GFP, followed by respective IPs). This is particularly important as the same group has published a paper (Spinnenhirn et al. FEBS Lett. 2017) stating that FAT10 targets newly translated proteins (Which is the case in such an overexpression experiment, but would have limited relevance for the in vivo situation).

Reply 31: We don't agree with the concern of our reviewer suggesting that JunB-FAT10ylation might be unspecific, namely that FAT10ylation of JunB might only happen under conditions, when the substrates are overexpressed. FAT10ylation of newly synthesized proteins, as we have published recently (Spinnenhirn et al. FEBS Lett. 2017), is for sure one function of FAT10 and we of course cannot exclude, that also newly synthesized JunB becomes modified with FAT10. However, since the proteome of FAT10 conjugation substrates has the smallest overlap with that of ubiquitin when compared to other UBL modifiers (Merbl et al., Cell 2013) a high specificity of FAT10 in choosing its substrates is suggested. Thus, it seems that not randomly all newly synthesized proteins are modified with FAT10. Additionally, we detect the JunB-FAT10 conjugate under completely endogenous conditions in human PBLs. If FAT10 would get conjugated unspecifically to every newly synthesized protein, basically every protein should be a substrate of FAT10ylation and this is something we do not see in our previously reported proteomic analysis of FAT10-linked proteins from IFN- γ /TNF- α stimulated HEK293 cells (Aichele et al. J. Cell Sci. 2010).

• Figure 1e. Based on the mutational analysis the authors argue that all sites are involved in FAT 10 modification with K237R having a more pronounced effect. It's a bit strange that even the K3R mutant shows a stronger FAT10 band than the K237R mutant. The authors should attempt to validate the FAT10 modification of particular sites of JunB by Mass Spec upon JunB IP.

Reply 32: It is very difficult to identify the FAT10-modified lysines by mass spectrometry. Upon trypsin digest of a FAT10ylated substrate, a 13 residue C-terminal FAT10 fragment remains on the substrate, which can't be analyzed by a normal MS approach. We think that our Western Blot data are convincing enough to show, that FAT10 indeed gets conjugated to one of these three SUMOylation sites. Since a JunB-FAT10 conjugate is still formed with the JunB-K3R, FAT10ylation certainly occurs also at additional lysine residues to a minor extent. In case of the particular experiment shown in Figure 1e, the differences in the JunB-FAT10 conjugate certainly also arise from slightly different amounts of JunB variants in the lysates (see lysate IB : HA). Our conclusion that FAT10 gets mainly conjugated to Lys-237 derived from comparing the results of all experiments performed with these mutants. However, we agree that the statement that FAT10 mainly gets conjugated to Lys-237 is difficult to prove without further analysis. Therefore we have deleted the following sentence on page 6: **“However, the strongest**

effect was observed when expressing the JunB-K237R-HA mutant together with FLAG-FAT10 (Fig. 1e, lane 4), suggesting that this lysine could be the main FAT10ylation site within JunB.”

- Figure 2A. In vitro sumoylation experiments. Concentrations of enzymes used for the in vitro assays appear strange (e.g only 200pM E1 (typo?) in combination with 12,5uM SUMO2/ 8.3uM SUMO1/ 6,2uM FAT10) E2 conc not specified (Same as in 7c? (1,4uM?). Clarify throughout the whole manuscript. Fat 10 Blot missing.

Reply 33: Indeed, only very low amounts of FLAG-UBA6 are necessary to achieve FAT10 activation in our experiments. We apologize for having forgotten to mention the concentration of UBC9 which was indeed 1.4 μ M and have now added this information to the Materials and Methods part. We have now also included the missing FAT10 Western blot in Figure 2a.

- Figure 3a. It is mentioned in the text that the RanGAP-SUMO FRET experiments were also performed with the FAT10 diglycine mutant (Data not shown). As the data presented in Figure 3a is very clearcut, also the experiment with the diGly FAT10 mutant should be included in the figure or Suppl. Information.

Reply 34: We have included the result of the FRET experiment using FAT10-AV as new Supplementary Figure 1 and have also mentioned this result in the Results section.

“As already observed for JunB SUMOylation in Fig. 2, the FAT10 diglycine mutant FAT10-AV also inhibited RanGAP SUMOylation, yet even more efficiently (Supplementary Fig. 1).”

- Figure 3e. Authors show that in the cell line expressing FAT10 PML bodies are reduced. Given the fact that FAT10 is induced upon IFN γ /TNF treatment, it would strengthen the statement of the paper and show physiological relevance, if the same effect (reduction of endogenous PML levels) is seen in wt cells after TNF/IFN γ treatment.

Reply 35: We completely agree with this suggestion. Inspired by this comment as well as by the comment of our second reviewer who pointed to the fact that co-localization of SUMO-1 and PML does not necessarily reflect covalent modification of PML with SUMO-1, we have now completely rearranged this part of Fig. 3. In detail we have moved old Fig. 3e and f, showing the co-localization of SUMO-1 and PML to new Supplementary Fig. 3a. As new data, since covalent SUMOylation of PML is a prerequisite for PML body formation, we have added the calculation of the numbers of PML bodies in stable FLAG-FAT10 expressing cells (new Fig. 3e), and in HEK293 wildtype or FAT10 knockout cells, treated or not with IFN- γ /TNF- α to induce expression of endogenous FAT10 (new Fig. 3f). Here it can clearly be seen, that in presence of FAT10, the number of PML bodies is significantly reduced. Under endogenous conditions in HEK293 cells, more PML bodies are formed in cytokine treated FAT10 knockout cells than in wildtype cells, due to the missing inhibition of PML SUMOylation when FAT10 is knocked out. Moreover, we included a new Western blot showing an increase in PML SUMOylation in HEK293 FAT10 knockout cells as compared to HEK293 wildtype cells, expressing FAT10 upon cytokine treatment (new Fig. 3g), further confirming the new data shown in the new Fig. 3f.

- Figure 5a. As recombinant proteins are available, it would strongly support the authors argumentation if experimental data for binding affinities of FAT 10 to UBA2/AOS1 are provided (e.g by Biacore

measurements or Microscale thermophoresis (Nanotemper)). This might also be a suitable approach to address/validate the crosslink results shown in figure 7 (Higher interaction with AOS1(Fig. 7d)).

Reply 36: We thank our reviewer for this pertinent suggestion and have now performed experiments using the Octet system to determine K_d levels. Using recombinant proteins we were able to measure a K_d value of 1.8 μM for binding of GST-FAT10 to 6His-AOS1/UBA2 (new Supplementary Fig. 7). Under the same conditions, binding of GST-SUMO-1 to AOS1/UBA2 could not be measured for unclear reasons. However, a K_d value of 1.8 μM for binding of FAT10 to AOS1/UBA2 is similar to the previously published K_m value for the interaction of SUMO-1 with AOS1/UBA2 of $0.74508 \pm 0.1105 \mu\text{M}$ (Wiryawan et al., *Biotechnology and Bioengineering*, Vol. 112, 2015). Even though K_d and K_m values are not always directly comparable, these similar binding affinities of FAT10 and SUMO-1 for AOS1/UBA2 are in good agreement with the experiment shown in Fig. 3a. We have therefore added the following sentence to the main manuscript (page: 13).

“This finding was further confirmed by determination of the affinity constant of the FAT10-AOS1/UBA2 interaction using the Octet system (Supplementary Fig. 7). The measured K_d value of 1.8 μM for the interaction of FAT10 and AOS1/UBA2 is similar to the previously published K_m value for SUMO-1 and AOS1/UBA2 of $0.74508 \pm 0.1105 \mu\text{M}^{41}$ which is in good agreement with the FRET-based inhibition assay shown in Fig. 3a.”

• Figure 5b and c. Thioester formation might be better monitored upon a shorter incubation period (e.g 10min) as after 30 min most likely dominantly conjugates are monitored. This might explain why only minor differences between red and non red conditions are observed. Likewise, band representing SUMO-E1 thioester would be expected to be stronger. It might be more conclusive to show Thioester formation with a titration of FAT10.

Reply 37: We have performed the *in vitro* experiments also for shorter time periods as can be seen in an example in Fig. 4 of this point to point letter. In contrast to the expectation of our reviewer, we did not observe a major difference when changing the incubation time or the temperature. Therefore, we have decided to maintain the conditions of our *in vitro* experiments as shown in the paper. Concerning a titration of FAT10 to show the inhibition of SUMO activation in a clearer way, we already had such a titration in old Supplementary Fig. 3a. We have now exchanged this Figure and have included additional molar ratios as well as a FAT10 Western blot in the new Supplementary Fig. 8a.

Figure 4: in vitro SUMO activation at 30°C or 37°C for 10, 15, or 30 minutes. The same protein amounts were used as in Figure 5b of the main manuscript.

• Figure 5e and Figure 7a-c show important results but might be shifted to suppl data to facilitate reading

Reply 38: We agree that Figure 7a-c and also Fig. 5e show important results and hence prefer to leave them in the main manuscript.

• Page 6 Referring to the paper by Deque et al (Nat Immunol. 2016) authors imply a competitive action of FAT10/SUMO modification in the inflammatory response. However the paper cited refers to a role of SUMO in type I IFN expression and Type I IFN ISG gene activation, whereas FAT10 is induced by IFN γ . The strong statement that “the pro-inflammatory cytokine induced UBL FAT10 inhibits the conjugation of the anti-inflammatory UBL (SUMO)” for sure is an oversimplification as the action of SUMO is not generally anti-inflammatory but more complex (e.g neg action on Nf κ B signaling, IL-8, PPAR etc.).

Reply 39: We agree with this statement of our referee. We are aware that SUMO is not generally anti-inflammatory and that the paper of Decque refers to a role of SUMO in innate signaling. We have recently published that FAT10 fine-tunes the balance of interferons produced during an LCMV infection by lowering the production of type I and enhancing the production of type II interferons (Mah et al., Mol Immunol 2019, The ubiquitin-like modifier FAT10 is required for normal IFN- γ production by activated CD8(+) T cells”). Since both type I and type II interferons are considered to be pro-inflammatory, it is also not so obvious that FAT10 is exclusively pro-inflammatory. Since space is limited and an enhanced discussion about the FAT10 and SUMO interplay in inflammation would be out of the scope of this manuscript, we have decided to remove completely this statement from the paper.

We have made the following changes in the text:

Abstract: Hence, we report the unprecedented paradigm that one ubiquitin-like modifier (FAT10) inhibits the conjugation and function of another ubiquitin-like modifier (SUMO) by impairing its activation.

Results (page 6): Since our data revealed that both modifiers were conjugated to the same lysines within JunB, we were wondering if FAT10ylation and SUMOylation of JunB was regulated by a putative competitive mechanism.

• Minor: Materials and methods: Rek proteins and in vitro assays.Fig5b: Concentration of FAT10AV, His-ISG15 and lin Di-Ub-Concentration not provided- please add.

Reply 40: We already had these concentrations listed in the Material and Methods sections, but maybe in a way, that was not clear enough. We have therefore now changed the text to:

“Fig. 5b: 0.2 nM AOS1/UBA2, 8.3 μM SUMO-1, 6.2 μM FAT10, 6.2 μM FAT10-AV, 6.2 μM His-ISG15, 6.2 μM linear di-Ub, 15 μM His-Ub;”

REVIEWERS' COMMENTS:

Reviewer #1 (Remarks to the Author):

I appreciate the authors thorough response and feel the revised version of the manuscript is much improved and they have addressed the majority of my concerns in their response. That said, based on their response I do have a few suggestions for clarifying the presentation of the data so that future readers can clearly interpret the results. Once these issues are addressed I feel the manuscript would be acceptable for publication.

I appreciate in Reply 3 of the authors rebuttal, the clarification of use of imputed ratios. While the authors claim they did not intend to quantify cross-links over 10 orders of magnitude, the imputed values they report do span this large range and I feel they need to justify their reported results. This brings into question the underlying value of reporting these imputed log₂ ratios versus just indicating they are only detected in one sample and not the other. The authors need to make clear in the methods section of the text that they are utilizing imputed ratios and how the imputed ratios were determined. Contrary to the authors statement in the literature cited on imputed values (Leitner et al, Nature Protoc, 2014 (PMID:24356771); Walzthoeni et al., Nature Methods, 2015 (PMID:26501516); Sailer et al., Nat Commun, 2018 (PMID: 30361475), only Walzthoeni et al. briefly mentions how xTract performs the imputation: "If a peptide was not detected in one experiment but was detected in the reference experiment (or vice versa), the fold change is estimated on the basis of the minimum detectable signal intensity (e.g., 1×10^3 for Orbitrap Elite), and instead of the area, the intensity of the first isotope is used for the comparison. This is documented in the output file by the sign ">=" if the I.D. has not been detected in the reference experiment but was in the comparison experiment and by the sign "<=" for the opposite situation." Leitner et al. perform no quantitation of cross-links so have no discussion of imputed quantitative values and Sailer et al. use xTract imputation but have no discussion or explanation about it. I would not consider this a lengthy discussion, but I do feel the authors should include a similar statement as Walzthoeni et al. in their methods and specify what the minimum detectable signal intensity value they used for the imputation since the Orbitrap Fusion Tribrid mass spectrometer may have a different minimal detectable signal intensity than an Orbitrap Elite mass spectrometer.

Gauging the sense of the overall pattern of changing cross-links seems like a subjective process and it is not clear how the authors do this. To clarify this for future readers I would suggest that instead they include a figure with a graphical direct head to head comparison of the quantitative cross-linking results from supplementary tables 3 and 4. For the 60 cross-links that are shared between the two experiments this makes the picture much more clear (see Figure 1 below or attached). Along with this the authors should include some discussion and a Venn diagram with the overlap of cross-links that were detected and quantified in the presence and/or absence of FAT10. My own cursory analysis shows over half (62/122 non-redundant cross-links) of the data presented in supplementary table S4 is not shared in supplementary table S3. Similarly, there are 12 unique cross-links to supplementary table S3. Some of this is to be expected with the addition of FAT10, however 36 of the 62 links that are unique to the experiment with FAT10 added do not involve FAT10 directly. That is, they are cross-links detected and quantified within AOS1, UBA2 and Sumo that were not found in the absence of FAT10. There should be some mention and discussion of these links in the text as there is no way to rule out that they represent some change in the complex conformation as a result of FAT10 addition.

I appreciate the clarification on the use of Tris during cross-linking in Reply 6 in the authors rebuttal. I could not find in the reference supplied by authors (Wang et al., Proteomics, 2010 (PMID:20127692)) any data or discussion that Tris is sterically too confined to react with NHS esters, only that indeed it can be used as a quenching agent. I do agree with the authors that ammonium bicarbonate is a better quenching reagent. That said I think they should clarify their reasoning for using Tris in the text to make clear to readers that they considered the potential effects of Tris on the cross-linking results. I especially appreciate the additional experiments

undertaken with HEPES, but yet again am not entirely convinced how the authors go about determining significant changes in the cross-linking pattern between the Tris and HEPES buffer conditions? I can clearly see what appears to be many more cross-links between FAT10 and AOS1 and between AOS1 and UBA2 in the HEPES buffer, than the Tris buffer. How many differing cross-links do the authors consider to be significant? I suggest to include a table with the full list of cross-links identified in the HEPES experiment so readers can make up their own minds about the significance.

In response to reply 22 of the authors rebuttal, I appreciate the explanation on the reasons to avoid structural modeling in these experiments. I still think it would be helpful to include a graphical representation of the quantitative values and statistical significance to accompany Supplementary Fig 8 b & c, such as volcano plots as shown in Supplementary Fig 4 of Sailer et al, Nat Commun, 2018 (PMID:30361475). This would allow readers to more clearly and easily interpret the changing links shown in Supplementary Fig 8 b & c without having to attempt to compare with the supplementary tables 3 and 4.

Grammatical errors:

Page 25 – change iodacetamid to iodoacetamide.

Page 27 – LC-MS/MS section add % to the various solvent compositions (eg. acetonitrile/formic acid 5/0.1, vol/vol).

Reviewer #3 (Remarks to the Author):

The authors have thoroughly adressed all of the questions and concerns which I raised and performed new experiments, which further increased the quality of this manuscript. In particular, measurement of the Kd values of the FAT10 - AOS/UBA2 interaction and the revised figure 3 adressing PML formation are strong datasets further supporting the argumentation of the authors. I do not have any more concerns and would strongly support publication in the revised form.

Point-by-point reply REVIEWERS' COMMENTS for manuscript NCOMMS-19-11396A:

Reviewer #1 (Remarks to the Author):

I appreciate the authors thorough response and feel the revised version of the manuscript is much improved and they have addressed the majority of my concerns in their response. That said, based on their response I do have a few suggestions for clarifying the presentation of the data so that future readers can clearly interpret the results. Once these issues are addressed I feel the manuscript would be acceptable for publication.

We are happy to hear that the reviewer judges the manuscript to be ready for publication after we have followed the additional suggestions for improvement.

1.) I appreciate in Reply 3 of the authors rebuttal, the clarification of use of imputed ratios. While the authors claim they did not intend to quantify cross-links over 10 orders of magnitude, the imputed values they report do span this large range and I feel they need to justify their reported results. This brings into question the underlying value of reporting these imputed log₂ ratios versus just indicating they are only detected in one sample and not the other. The authors need to make clear in the methods section of the text that they are utilizing imputed ratios and how the imputed ratios were determined. Contrary to the authors statement in the literature cited on imputed values (Leitner et al, Nature Protoc, 2014 (PMID:24356771); Walzthoeni et al., Nature Methods, 2015 (PMID:26501516); Sailer et al., Nat Commun, 2018 (PMID: 30361475), only Walzthoeni et al. briefly mentions how xTract performs the imputation: “If a peptide was not detected in one experiment but was detected in the reference experiment (or vice versa), the fold change is estimated on the basis of the minimum detectable signal intensity (e.g., 1×10^3 for Orbitrap Elite), and instead of the area, the intensity of the first isotope is used for the comparison. This is documented in the output file by the sign “>=” if the I.D. has not been detected in the reference experiment but was in the comparison experiment and by the sign “<=” for the opposite situation.” Leitner et al. perform no quantitation of cross-links so have no discussion of imputed quantitative values and Sailer et al. use xTract imputation but have no discussion or explanation about it. I would not consider this a lengthy discussion, but I do feel the authors should include a similar statement as Walzthoeni et al. in their methods and specify what the minimum detectable signal intensity value they used for the imputation since the Orbitrap Fusion Tribrid mass spectrometer may have a different minimal detectable signal intensity than an Orbitrap Elite mass spectrometer.

We agree with the reviewer and in order to clarify the reviewers concerns about imputed values we added the following sentence to the methods section on page 30:

„If a peptide was detected in only one condition (e.g. only in the reference experiment or vice versa), the fold change was estimated on the basis of the minimum detectable signal intensity ($1e^3$ for Orbitrap Fusion Tribrid mass spectrometer), and instead of the area, the intensity of the first isotope was used for the comparison. This is indicated in Supplementary Data 3 and 4 in the column ‘imputed values’.“

2.) Gauging the sense of the overall pattern of changing cross-links seems like a subjective process and it is not clear how the authors do this. To clarify this for future readers I would suggest that instead they include a figure with a graphical direct head to head comparison of the quantitative cross-linking results from supplementary tables 3 and 4. For the 60 cross-links that are shared between the two experiments this makes the picture much more clear (see Figure 1 below or attached). Along with this the authors should include some discussion

and a Venn diagram with the overlap of cross-links that were detected and quantified in the presence and/or absence of FAT10. My own cursory analysis shows over half (62/122 non-redundant cross-links) of the data presented in supplementary table S4 is not shared in supplementary table S3. Similarly, there are 12 unique cross-links to supplementary table S3. Some of this is to be expected with the addition of FAT10, however 36 of the 62 links that are unique to the experiment with FAT10 added do not involve FAT10 directly. That is, they are cross-links detected and quantified within AOS1, UBA2 and Sumo that were not found in the absence of FAT10. There should be some mention and discussion of these links in the text as there is no way to rule out that they represent some change in the complex conformation as a result of FAT10 addition.

In total, 134 unique uxIDs were reliably quantified in at least one of the two quantification experiments (quantification of crosslinks in presence or absence of SUMO-1 or SUMO-1/FAT10, see Supplementary Figures 8b and c and Supplementary Data 3 and 4). Figure 1 of this point-by-point reply shows that 60/134 uxIDs were reliably quantified in both experiments, so in both experiments a log₂ ratio could be determined.

Figure 1: Venn-diagram showing the overlap of uxIDs that were reliably quantified in presence or absence of SUMO-1 or SUMO-1/FAT10.

For the comparison of these 60 log₂ ratios, a linear regression analysis was performed which is shown in Figure 2 of this point-by-point reply. The resulting regression coefficient is $r=0.99$. This data strongly indicates, that there is no major difference in crosslink abundance if FAT10 is present in the sample or not because all uxIDs are regulated in the same direction in both conditions.

Figure 2: Linear regression analysis of the log₂ ratios determined in the two quantification experiments.

If a log₂ ratio could not be determined for an uxID in one of the experiments it is difficult to make a statement whether this uxID is involved in a significant change in abundance if FAT10 is present or not, because these uxIDs could be regulated either way.

Of those, 12 uxIDs were reliably quantified only in the sample in presence or absence of SUMO-1. 6 of those 12 uxIDs were identified in both experiments but not reliably quantified due to one of the following parameters: p-value > 0.01, ID-Score < 28 or violations > 0. 6/12 uxIDs were not identified in the sample with FAT10.

A further 62 uxIDs were reliably quantified only in the sample in presence or absence of SUMO-1/FAT10. As the reviewer pointed out correctly, 26/62 uxIDs involve FAT10 and can therefore not be identified in samples without FAT10. Most of the other uxIDs (80 %) were identified in crosslinking experiments with SUMO-1 and SUMO-1/FAT10. These uxIDs were not reliably quantified due to one of the following parameters: p-value > 0.01, ID-Score < 28 or violations > 0.

However, uxIDs which are downregulated in one experiment and not identified in the other experiment are probably regulated in the same direction (if we consider the total absence of a uxID as the strongest form of downregulation).

To the contrary, uxIDs which are upregulated in one experiment and not identified in the other condition are potential candidates for a differential regulation. These uxIDs are listed in Table 1. 3/5 uxIDs which show this behaviour involve SUMO-1 and do not allow a conclusion on SUMO E1 structure. The uxID 6His-AOS1:55:x:UBA2:336 seems to be downregulated in one condition and upregulated in the other, however the adjacent uxIDs 6His-AOS1:55:x:UBA2:324 and 6His-AOS1:55:x:UBA2:371 behave similarly in both conditions. In addition, if the Id-Score conditions are lowered for the quantification experiment in presence or absence of Sumo-1 to 22, this uxIDs is found upregulated with a log₂ ratio of 1.36 (which is the exact same value as in the quantification experiment in presence or absence of SUMO-1/FAT10).

Table 1: uxIDs which were upregulated in one experiment and not identified in the other experiment.

No.	uxID	log2ratio table S3 Sumo-1	log2ratio table S4 Sumo-1/Fat10
1	UBA2:308:x:UBA2:65	1,14	not identified
2	Sumo:45:x:UBA2:505	12,14	not identified
3	Sumo:37:x:UBA2:623	12,58	not identified
4	6His-AOS1:55:x:UBA2:336	not identified	1,36
5	Sumo:37:x:Sumo:45	not identified	12,69

This leaves only one uxID UBA2:308:x:UBA2:65 which is clearly not similarly regulated between the different datasets.

We think however that including this detailed explanation is beyond the scope of this manuscript and the importance that the quantitative XL-MS data has on the conclusions of this manuscript.

However, in order for the readers of the manuscript to better compare the changes in the crosslinking pattern between the different datasets we have added Figure 2 from above as Supplementary Figure 8 d and have also included the following text on page 10 of the Supplementary Information.

“d. Linear regression analysis of the log₂ ratios of uxIDs which have been quantified in both experimental datasets (e.g. in presence or absence of SUMO-1 (S3) and in presence or absence of FAT10/SUMO-1 (S4)) results in a regression coefficient of $r=0.99$, indicating a strong correlation.”

3.) *I appreciate the clarification on the use of Tris during cross-linking in Reply 6 in the authors rebuttal. I could not find in the reference supplied by authors (Wang et al., Proteomics, 2010 (PMID:20127692)) any data or discussion that Tris is sterically too confined to react with NHS esters, only that indeed it can be used as a quenching agent. I do agree with the authors that ammonium bicarbonate is a better quenching reagent. That said I think they should clarify their reasoning for using Tris in the text to make clear to readers that they considered the potential effects of Tris on the cross-linking results. I especially appreciate the additional experiments undertaken with HEPES, but yet again am not entirely convinced how the authors go about determining significant changes in the cross-linking pattern between the Tris and HEPES buffer conditions? I can clearly see what appears to be many more cross-links between FAT10 and AOS1 and between AOS1 and UBA2 in the HEPES buffer, than the Tris buffer. How many differing cross-links do the authors consider to be significant? I suggest to include a table with the full list of cross-links identified in the HEPES experiment so readers can make up their own minds about the significance.*

Very similar amounts of crosslinks were identified under both buffer conditions; in the crosslinking experiment using TRIS buffer 234 unique crosslinks compared to 222 unique crosslinks in HEPES buffer were identified (both at an Id-Score > 25 and an FDR ≤ 0.05). We therefore believe that the data continues to speak for itself.

However, in order for every reader to make-up its own mind, we have now added the following sentences to the manuscript on page 28.

“Tris was used as all the biochemical assays were performed in Tris containing buffer and in order to replicate these experimental conditions as close as possible for the crosslinking experiments. A comparison of AOS1/UBA2 plus FAT10 linked in Tris and after buffer

exchange into 20 mM HEPES, pH 7.5 confirmed that both buffer conditions yielded a highly comparable number of crosslinks and that the presence of Tris in our buffer did not preclude the formation of lysine-lysine crosslinks (Supplementary Data 5)".

Also, all the crosslinks identified in the additional crosslinking experiment using HEPES buffer are now included in an additional Supplementary Data 5.

4.) *In response to reply 22 of the authors rebuttal, I appreciate the explanation on the reasons to avoid structural modeling in these experiments. I still think it would be helpful to include a graphical representation of the quantitative values and statistical significance to accompany Supplementary Fig 8 b & c, such as volcano plots as shown in Supplementary Fig 4 of Sailer et al, Nat Commun, 2018 (PMID:30361475). This would allow readers to more clearly and easily interpret the changing links shown in Supplementary Fig 8 b & c without having to attempt to compare with the supplementary tables 3 and 4.*

We appreciate the suggestion of the reviewer, but unfortunately volcano plots are not suitable for a direct comparison of the two quantification experiments in this case, as one would need to plot 2 separate volcano plots and would need to label individual data points for comparison. We feel that this would complicate the interpretation in comparison to the 2-D graphical representation that we currently have already included in Supplementary Figure 8 b and c.

5.) *Grammatical errors: Page 25 – change iodacetamid to iodoacetamide. Page 27 – LC-MS/MS section add % to the various solvent compositions (eg. acetonitrile/formic acid 5/0.1, vol/vol).*

The grammatical errors have been changed as suggested by the reviewer.